# ADMM for Nonsmooth Composite Optimization under Orthogonality Constraints

## Abstract

We consider a class of structured, nonconvex, nonsmooth optimization problems under orthogonality constraints, where the objectives combine a smooth function, a nonsmooth concave function, and a nonsmooth weakly convex function. This class of problems finds diverse applications in statistical learning and data science. Existing methods for addressing these problems often fail to exploit the specific structure of orthogonality constraints, struggle with nonsmooth functions, or result in suboptimal oracle complexity. We propose OADMM, an Alternating Direction Method of Multipliers (ADMM) designed to solve this class of problems using efficient proximal linearized strategies. Two specific variants of OADMM are explored: one based on Euclidean Projection (OADMM-EP) and the other on Riemannian Retraction (OADMM-RR). Under mild assumptions, we prove that OADMM converges to a critical point of the problem with an ergodic convergence rate of $\mathcal{O}(1/\epsilon^3)$. Additionally, we establish a super-exponential convergence rate or polynomial convergence rate for OADMM, depending on the specific setting, under the Kurdyka-Lojasiewicz (KL) inequality. To the best of our knowledge, this is the first non-ergodic convergence result for this class of nonconvex nonsmooth optimization problems. Numerical experiments demonstrate that the proposed algorithm achieves state-of-the-art performance.

**Keywords:** Orthogonality Constraints; Nonconvex Optimization; Nonsmooth Composite Optimization; ADMM; Convergence Analysis

## 1 Introduction

This paper focuses on the following nonsmooth composite optimization problem under orthogonality constraints ('$\triangleq$' means define):

$$\min_{\mathbf{X} \in \mathbb{R}^{n \times r}} F(\mathbf{X}) \triangleq f(\mathbf{X}) - g(\mathbf{X}) + h(\mathcal{A}(\mathbf{X})), \ s.t. \ \mathbf{X}^\mathsf{T}\mathbf{X} = \mathbf{I}_r. \tag{1}$$

Here, $n \geq r$, $\mathcal{A}(\mathbf{X}) \in \mathbb{R}^m$ is a linear mapping of $\mathbf{X}$, and $\mathbf{I}_r$ is a $r \times r$ identity matrix. For conciseness, the orthogonality constraints $\mathbf{X}^\mathsf{T}\mathbf{X} = \mathbf{I}_r$ in Problem (1) is rewritten as $\mathbf{X} \in \mathcal{M} \in \mathbb{R}^{n \times r}$, with $\mathcal{M}$ representing the Stiefel manifold in the literature (Edelman et al., 1998; Absil et al., 2008b).

We impose the following assumptions on Problem (1) throughout this paper. ($\mathbb{A}$-i) $f(\mathbf{X})$ is $L_f$-smooth, satisfying $\|\nabla f(\mathbf{X}) - \nabla f(\mathbf{X}')\|_\mathsf{F} \leq L_f \|\mathbf{X} - \mathbf{X}'\|_\mathsf{F}$ holds for all $\mathbf{X}, \mathbf{X}' \in \mathbb{R}^{n \times r}$. This implies: $|f(\mathbf{X}) - f(\mathbf{X}') - \langle \nabla f(\mathbf{X}'), \mathbf{X} - \mathbf{X}' \rangle| \leq \frac{L_f}{2}\|\mathbf{X} - \mathbf{X}'\|_\mathsf{F}^2$ (cf. Lemma 1.2.3 in (Nesterov, 2003)). We also assume that $f(\mathbf{X})$ demonstrates $C_f$-Lipschitz continuity, with $\|\nabla f(\mathbf{X})\|_\mathsf{F} \leq C_f$ for all $\mathbf{X} \in \mathcal{M}$. The convexity of $f(\mathbf{X})$ is not assumed. ($\mathbb{A}$-ii) The function $g(\cdot)$ is convex, proper, and $C_g$-Lipschitz continuous, though it is not necessarily smooth. ($\mathbb{A}$-iii) The function $h(\cdot)$ is proper, lower semicontinuous, $C_h$-Lipschitz continuous, and potentially nonsmooth. Also, it is weakly convexity with constant $W_h \geq 0$, which implies that the function $h(\mathbf{y}) + \frac{W_h}{2}\|\mathbf{y}\|_2^2$ is convex for all $\mathbf{y} \in \mathbb{R}^m$. ($\mathbb{A}$-iv) The proximal operator, $\mathbb{P}_\mu(\mathbf{y}') \triangleq \arg\min_\mathbf{y} \frac{1}{2\mu}\|\mathbf{y} - \mathbf{y}'\|_2^2 + h(\mathbf{y})$, can be computed efficiently and exactly for any given $\mu > 0$ and $\mathbf{y}' \in \mathbb{R}^m$.

Problem (1) represents an optimization framework that plays a crucial role in a variety of statistical learning and data science models. These models include sparse Principal Component Analysis (PCA) (Journée et al., 2010; Lu & Zhang, 2012), deep neural networks (Cho & Lee, 2017; Xie et al.,

2017; Bansal et al., 2018; Cogswell et al., 2016; Huang & Gao, 2023), orthogonal nonnegative matrix factorization (Jiang et al., 2022), range-based independent component analysis (Selvan et al., 2015), and dictionary learning (Zhai et al., 2020).

## 1.1 RELATED WORK

▶ **Optimization under Orthogonality Constraints**. Solving Problem (1) is challenging due to the computationally expensive and non-convex orthogonality constraints. Existing methods can be divided into three classes. *(i)* Geodesic-like methods (Edelman et al., 1998; Abrudan et al., 2008; Absil et al., 2008b; Jiang & Dai, 2015). These methods involve calculating geodesics by solving ordinary differential equations, which can introduce significant computational complexity. To mitigate this, geodesic-like methods iteratively compute the geodesic logarithm using simple linear algebra calculations. Efficient constraint-preserving update schemes have been integrated with the Barzilai-Borwein (BB) stepsize strategy (Wen & Yin, 2013; Jiang & Dai, 2015) for minimizing smooth functions under orthogonality constraints. *(ii)* Projection and retractions methods (Absil et al., 2008b; Golub & Van Loan, 2013). These methods maintain orthogonality constraints through projection or retraction. They reduce the objective value by using its current Euclidean gradient direction or Riemannian tangent direction, followed by an orthogonal projection operation. This projection can be computed using polar decomposition or singular value decomposition, or approximated with QR factorization. *(iii)* Multiplier correction methods (Gao et al., 2018; 2019; Xiao et al., 2022). Leveraging the insight that the Lagrangian multiplier associated with the orthogonality constraint is symmetric and has an explicit closed-form expression at the first-order optimality condition, these methods tackle an alternative unconstrained nonlinear objective minimization problem, rather than the original smooth function under orthogonality constraints.

▶ **Optimization with Nonsmooth Objectives**. Another challenge in addressing Problem (1) stems from the nonsmooth nature of the objective function. Existing methods for tackling this challenge fall into three main categories. *(i)* Subgradient methods (Ferreira & Oliveira, 1998; Hwang et al., 2015; Li et al., 2021). Subgradient methods, analogous to gradient descent methods, can incorporate various geodesic-like and projection-like techniques. However, they often exhibit slower convergence rates compared to other approaches. *(ii)* Proximal gradient methods (Chen et al., 2020). These methods use a semi-smooth Newton approach to solve a strongly convex minimization problem over the tangent space, finding a descent direction while preserving the orthogonality constraint through a retraction operation. *(iii)* Operator splitting methods (Lai & Osher, 2014; Chen et al., 2016; Zhang et al., 2020b). These methods introduce linear constraints to break down the original problem into simpler subproblems that can be solved separately and exactly. Among these, ADMM is a promising solution for Problem (1) due to its capability to handle nonsmooth objectives and nonconvex constraints separately and alternately. Several ADMM-like algorithms have been proposed for solving nonconvex problems (Boţ & Nguyen, 2020; Boţ et al., 2019; Wang et al., 2019; Li & Pong, 2015; He & Yuan, 2012; Yuan, 2024; Zhang et al., 2020b), but these methods fail to exploit the specific structure of orthogonality constraints or cannot be adapted to solve Problem (1). *(iv)* Other methods. OBCD (Yuan, 2023) has been proposed to solve a specific class of our problems, while the exact augmented Lagrangian method ManIAL was introduced in (Deng et al., 2024).

▶ **Detailed Discussions on Operator Splitting Methods**. We list some popular variants of operator splitting methods for tackling Problem (1). Initially, two natural splitting strategies are used in the literature:

$$\min_{\mathbf{X},\mathbf{y}} F_1(\mathbf{X},\mathbf{y}) \triangleq f(\mathbf{X}) - g(\mathbf{X}) + h(\mathbf{y}) + \mathcal{I}_\mathcal{M}(\mathbf{X}), \ s.t. \ \mathcal{A}(\mathbf{X}) = \mathbf{y} \tag{2}$$

$$\min_{\mathbf{X},\mathbf{Y}} F_2(\mathbf{X},\mathbf{Y}) \triangleq f(\mathbf{X}) - g(\mathbf{X}) + h(\mathcal{A}(\mathbf{X})) + \mathcal{I}_\mathcal{M}(\mathbf{Y}), \ s.t. \ \mathbf{X} = \mathbf{Y}. \tag{3}$$

(*a*) Smoothing Proximal Gradient Methods (SPGM, (Beck & Rosset, 2023; Böhm & Wright, 2021)) incorporate a penalty (or smoothing) parameter $\mu \to 0$ to penalize the squared error in the constraints, resulting in the subsequent minimization problem (Beck & Rosset, 2023; Böhm & Wright, 2021; Chen, 2012): $\min_{\mathbf{X},\mathbf{y}} \ F_1(\mathbf{X},\mathbf{y}) + \frac{1}{2\mu}\|\mathcal{A}(\mathbf{X}) - \mathbf{y}\|_2^2$. During each iteration, SPGM employs proximal gradient strategies to alternatively minimize *w.r.t.* $\mathbf{X}$ and $\mathbf{y}$. (*b*) Splitting Orthogonality Constraints Methods (SOCM, (Lai & Osher, 2014)) use the following iteration scheme: $\mathbf{X}^{t+1} \approx \arg\min_{\mathbf{X}} F_2(\mathbf{X},\mathbf{Y}^t) + \langle \mathbf{Z}^t, \mathbf{X} - \mathbf{Y}^t \rangle + \frac{\beta}{2}\|\mathbf{X} - \mathbf{Y}\|_\mathsf{F}^2$, $\mathbf{Y}^{t+1} \in \arg\min_{\mathbf{Y}} F_2(\mathbf{X}^{t+1},\mathbf{Y}) + \langle \mathbf{Z}^t, \mathbf{X}^{t+1} - \mathbf{Y} \rangle + \frac{\beta}{2}\|\mathbf{X}^{t+1} - \mathbf{Y}\|_\mathsf{F}^2$, and $\mathbf{Z}^{t+1} = \mathbf{Z}^t + \beta(\mathbf{X}^{t+1} - \mathbf{Y}^{t+1})$, where $\beta$ is a fixed penalty constant, and $\mathbf{Z}^t$ is the multiplier associated with the constraint $\mathbf{X} = \mathbf{Y}$ at

Table 1: Comparison of existing methods for solving Problem (1).

| Reference | $h(\mathcal{A}(\mathbf{X}))$ | $g(\mathbf{X})$ | Notable Features | Complexity | Conv. Rate |
|---|---|---|---|---|---|
| SOCM (Lai & Osher, 2014) | convex $h(\cdot)$ | empty | $\sigma = 1, \alpha = 0$ | unknown | unknown |
| MADMM (Kovnatsky et al., 2016) | convex $h(\cdot)$ | empty | $\sigma = 1, \alpha = 0$ | unknown | unknown |
| RSG (Li et al., 2021) | weakly convex $h(\cdot)$ | empty | – | $\mathcal{O}(\epsilon^{-4})$ | unknown |
| ManPG (Chen et al., 2020) | $h(\mathcal{A}(\mathbf{X})) = \|\mathbf{X}\|_1$ | empty | hard subproblem | $\mathcal{O}(\epsilon^{-2})$ | unknown |
| OBCD (Yuan, 2023) | separable $h(\cdot)$ | empty | hard subproblem | $\mathcal{O}(\epsilon^{-2})$ | unknown |
| RADMM (Li et al., 2022) | convex $h(\cdot)$ | empty | $\sigma = 1, \alpha = 0$ | $\mathcal{O}(\epsilon^{-4})$ | unknown |
| ManIAL (Deng et al., 2024) | convex $h(\cdot)$ | empty | inexact subproblem | $\mathcal{O}(\epsilon^{-3})$ | unknown |
| SPGM (Beck & Rosset, 2023) | convex $h(\cdot)$ | empty | – | $\mathcal{O}(\epsilon^{-3})$ | unknown |
| OADMM-EP[ours] | weakly convex $h(\cdot)$ | convex | $\sigma \in [1,2), \alpha > 0$ | $\mathcal{O}(\epsilon^{-3})$ | $\mathcal{O}(1/\exp(T^{\dot{u}})), \dot{u} \in (0, \frac{2}{3}]$ $\star$ |
| OADMM-RR[ours] | weakly convex $h(\cdot)$ | convex | $\sigma \in [1,2)$, MBB | $\mathcal{O}(\epsilon^{-3})$ | or $\mathcal{O}(1/T^{\ddot{u}}), \ddot{u} \in (0, +\infty)$ $\ddagger$ |

Note $\star$: This is known as super-exponential convergence, please refer to Theorem 5.9(*a*) for more details.

Note $\ddagger$: This is known as polynomial convergence, please refer to Theorem 5.9(*b*) for more details.

iteration $t$. (*c*) Similarly, Manifold ADMM (MADMM, (Kovnatsky et al., 2016)) iterates as follows: $\mathbf{X}^{t+1} \approx \arg\min_{\mathbf{X}} F_1(\mathbf{X}, \mathbf{y}^t) + \langle \mathbf{z}^t, \mathcal{A}(\mathbf{X}) - \mathbf{y}^t \rangle + \frac{\beta}{2}\|\mathcal{A}(\mathbf{X}) - \mathbf{y}^t\|_{\mathsf{F}}^2$, $\mathbf{y}^{t+1} \in \arg\min_{\mathbf{y}} F_1(\mathbf{X}^{t+1}, \mathbf{y}) + \langle \mathbf{z}^t, \mathcal{A}(\mathbf{X}^{t+1}) - \mathbf{y} \rangle + \frac{\beta}{2}\|\mathcal{A}(\mathbf{X}^{t+1}) - \mathbf{y}\|_{\mathsf{F}}^2$, and $\mathbf{z}^{t+1} = \mathbf{z}^t + \beta(A(\mathbf{X}^{t+1}) - \mathbf{y}^{t+1})$, where $\mathbf{z}^t$ is the multiplier associated with the constraint $A(\mathbf{X}) - \mathbf{y} = \mathbf{0}$ at iteration $t$. (*d*) Like MADMM, Riemannian ADMM (RADMM, (Li et al., 2022)) operates using the first splitting strategy in Equation (2). In contrast, it employs a Riemannian retraction strategy to solve the $\mathbf{X}$-subproblem and a Moreau envelope smoothing strategy to solve the $\mathbf{y}$-subproblem.

**Contributions.** We compare existing methods for solving Problem (1) in Table 1, and our main contributions are summarized as follows. (*i*) We introduce OADMM, a specialized ADMM designed for structured nonsmooth composite optimization problems under orthogonality constraints in Problem (1). Two specific variants of OADMM are explored: one based on Euclidean Projection (OADMM-EP) and the other on Riemannian Retraction (OADMM-EP). Notably, while many existing works primarily address cases where $g(\mathbf{X}) = 0$ and $h(\cdot)$ is convex, our approach considers a more general setting where $h(\cdot)$ is weakly convex and $g(\mathbf{X})$ is convex. (ii) OADMM could demonstrate fast convergence by incorporating Nesterov's extrapolation (Nesterov, 2003) into OADMM-EP and a Monotone Barzilai-Borwein (MBB) stepsize strategy (Wen & Yin, 2013) into OADMM-RR to potentially accelerate primal convergence. Both variants also employ an over-relaxation strategy to enhance dual convergence (Gonçalves et al., 2017; Yang et al., 2017; Li et al., 2016). (*iii*) By introducing a novel Lyapunov function, we establish the convergence of OADMM to critical points of Problem (1) within an oracle complexity of $\mathcal{O}(1/\epsilon^3)$, matching the best-known results to date (Beck & Rosset, 2023; Böhm & Wright, 2021). This is achieved through a decreasing step size for updating primal and dual variables. In contrast, RADMM employs a small constant step size for such updates, resulting in a sub-optimal oracle complexity of $\mathcal{O}(\epsilon^{-4})$ (Li et al., 2022). (*iv*) We establish a super-exponential convergence rate or polynomial convergence rate for OADMM, depending on the specific setting, under the Kurdyka-Lojasiewicz (KL) inequality, providing *the first non-ergodic convergence result* for this class of non-convex nonsmooth optimization problems.

## 2 TECHNICAL PRELIMINARIES

This section provides some technical preliminaries on Moreau envelopes for weakly convex functions and manifold optimization.

**Notations.** We define $[n] \triangleq \{1, 2, \ldots, n\}$. We use $\mathcal{A}^{\mathsf{T}}(\cdot)$ to denote the adjoin operator of $\mathcal{A}(\cdot)$ with $\langle \mathcal{A}(\mathbf{X}), \mathbf{z} \rangle = \langle \mathbf{X}, \mathcal{A}^{\mathsf{T}}(\mathbf{z}) \rangle$ for all $\mathbf{X} \in \mathbb{R}^{n \times r}$ and $\mathbf{z} \in \mathbb{R}^m$. We define $\overline{\mathrm{A}} \triangleq \max_{\mathbf{V}} \|\mathcal{A}(\mathbf{V})\|_{\mathsf{F}}/\|\mathbf{V}\|_{\mathsf{F}}$. We use $\mathcal{I}_{\mathcal{M}}(\mathbf{X})$ to denote the indicator function of orthogonality constants. Further notations, technical preliminaries, and relevant lemmas are detailed in Appendix Section A.

### 2.1 MOREAU ENVELOPES FOR WEAKLY CONVEX FUNCTIONS

We provide the following useful definition.

**Definition 2.1.** *For a proper convex, and Lipschitz continuous function $h(\mathbf{y}): \mathbb{R}^m \mapsto \mathbb{R}$, the Moreau envelope of $h(\mathbf{y})$ with the parameter $\mu > 0$ is given by $h_\mu(\mathbf{y}) \triangleq \min_{\breve{\mathbf{y}}} h(\breve{\mathbf{y}}) + \frac{1}{2\mu}\|\breve{\mathbf{y}} - \mathbf{y}\|_2^2$.*

We show some useful properties of Moreau envelope for weakly convex functions.

**Lemma 2.2.** *Let $h : \mathbb{R}^m \mapsto \mathbb{R}$ to be a proper, $W_h$-weakly convex, and lower semicontinuous function. Assume $\mu \in (0, W_h^{-1})$. We have the following results (Böhm & Wright, 2021). (a) The function $h_\mu(\cdot)$ is $C_h$-Lipschitz continuous. (b) The function $h_\mu(\cdot)$ is continuously differentiable with gradient $\nabla h_\mu(\mathbf{y}) = \frac{1}{\mu}(\mathbf{y} - \mathbb{P}_\mu(\mathbf{y}))$ for all $\mathbf{y}$, where $\mathbb{P}_\mu(\mathbf{y}) \triangleq \arg\min_{\breve{\mathbf{y}}} h(\breve{\mathbf{y}}) + \frac{1}{2\mu}\|\breve{\mathbf{y}} - \mathbf{y}\|_2^2$. This gradient is $\max(\mu^{-1}, \frac{W_h}{1-\mu W_h})$-Lipschitz continuous. In particular, when $\mu \in (0, \frac{1}{2W_h}]$, the condition $\mu^{-1} \geq \frac{W_h}{1-\mu W_h}$ ensures that $h_\mu(\mathbf{y})$ is $(\mu^{-1})$-smooth and $(\mu^{-1})$-weakly convex.*

**Lemma 2.3.** *(Proof in Appendix B.1) Assume $0 < \mu_2 < \mu_1 < \frac{1}{W_h}$, and fixing $\mathbf{y} \in \mathbb{R}^m$. We have: $0 \leq h_{\mu_2}(\mathbf{y}) - h_{\mu_1}(\mathbf{y}) \leq \min\{\frac{\mu_1}{2\mu_2}, 1\} \cdot (\mu_1 - \mu_2)C_h^2$.*

**Lemma 2.4.** *(Proof in Appendix B.2) Assume $0 < \mu_2 < \mu_1 \leq \frac{1}{2W_h}$, and fixing $\mathbf{y} \in \mathbb{R}^m$. We have: $\|\nabla h_{\mu_1}(\mathbf{y}) - \nabla h_{\mu_2}(\mathbf{y})\| \leq (\frac{\mu_1}{\mu_2} - 1)C_h$.*

**Lemma 2.5.** *(Proof in Appendix B.3) Assume that $h(\mathbf{y})$ is $W_h$-weakly convex, $\mu \in (0, \frac{1}{2W_h}]$, $\beta > \mu^{-1}$. Consider the following strongly convex optimization problem: $\bar{\mathbf{y}} = \arg\min_{\mathbf{y}} h_\mu(\mathbf{y}) + \frac{\beta}{2}\|\mathbf{y} - \mathbf{b}\|_2^2$, which is equivalent to: $(\bar{\mathbf{y}}, \breve{\mathbf{y}}) = \arg\min_{\mathbf{y},\mathbf{y}'} h(\mathbf{y}') + \frac{1}{2\mu}\|\mathbf{y}' - \mathbf{y}\|_2^2 + \frac{\beta}{2}\|\mathbf{y} - \mathbf{b}\|_2^2$. We have: (a) $\bar{\mathbf{y}} = \frac{(\breve{\mathbf{y}} + \mu\beta\mathbf{b})}{1+\mu\beta}$, where $\breve{\mathbf{y}} = \arg\min_{\mathbf{y}} h(\mathbf{y}) + \frac{\beta}{2(1+\mu\beta)}\|\mathbf{y} - \mathbf{b}\|_2^2 = \mathbb{P}_{[\mu+1/\beta]}(\mathbf{b})$. (b) $\beta(\mathbf{b} - \bar{\mathbf{y}}) \in \partial h(\breve{\mathbf{y}})$. (c) $\|\bar{\mathbf{y}} - \breve{\mathbf{y}}\| \leq \mu C_h$.*

**Remark 2.6.** *(i) Lemmas 2.3 and 2.4 presented in this paper are novel. (ii) The upper bound in Lemma 2.3 is slightly better than the bound established in Lemma 4.1 of (Böhm & Wright, 2021). (iii) Lemma 2.5 is very critical in our algorithm development and theoretical analysis.*

## 2.2 MANIFOLD OPTIMIZATION

We define the $\epsilon$-stationary point of Problem (1) as follows.

**Definition 2.7.** *(First-Order Optimality Conditions, (Chen et al., 2020; Li et al., 2022; Beck & Rosset, 2023)) The solution $(\ddot{\mathbf{X}}, \ddot{\mathbf{y}}, \ddot{\mathbf{z}})$ with $\ddot{\mathbf{X}} \in \mathcal{M}$ is called an $\epsilon$-stationary point of Problem (1) if: $\mathrm{Crit}(\ddot{\mathbf{X}}, \ddot{\mathbf{y}}, \ddot{\mathbf{z}}) \leq \epsilon$, where $\mathrm{Crit}(\mathbf{X}, \mathbf{y}, \mathbf{z}) \triangleq \|\mathcal{A}(\mathbf{X}) - \mathbf{y}\| + \|\partial h(\mathbf{y}) - \mathbf{z}\| + \|\mathrm{Proj}_{\mathbf{T}_{\mathbf{X}}\mathcal{M}}(\nabla f(\mathbf{X}) - \partial g(\mathbf{X}) + \mathcal{A}^\mathsf{T}(\mathbf{z}))\|_\mathsf{F}$. Here, according to (Absil et al., 2008a), for all $\mathbf{X} \in \mathcal{M}$ and $\mathbf{\Delta} \in \mathbb{R}^{n \times r}$, we have: $\mathrm{Proj}_{\mathbf{T}_{\mathbf{X}}\mathcal{M}}(\mathbf{\Delta}) = \mathbf{\Delta} - \frac{1}{2}\mathbf{X}(\mathbf{\Delta}^\mathsf{T}\mathbf{X} + \mathbf{X}^\mathsf{T}\mathbf{\Delta})$.*

The proposed algorithm is an iterative procedure. After shifting the current iterate $\mathbf{X} \in \mathcal{M}$ in the search direction, it may no longer reside on $\mathcal{M}$. Therefore, we must retract the point onto $\mathcal{M}$ to form the next iterate. The following definition is useful in this context.

**Definition 2.8.** *A retraction on $\mathcal{M}$ is a smooth map (Absil et al., 2008a): $\mathrm{Retr}_{\mathbf{X}}(\mathbf{\Delta}) \in \mathcal{M}$ with $\mathbf{X} \in \mathcal{M}$ and $\mathbf{\Delta} \in \mathbb{R}^{n \times r}$ satisfying $\mathrm{Retr}_{\mathbf{X}}(\mathbf{0}) = \mathbf{X}$, and $\lim_{\mathbf{T}_{\mathbf{X}}\mathcal{M} \ni \mathbf{\Delta} \to \mathbf{0}} \frac{\|\mathrm{Retr}_{\mathbf{X}}(\mathbf{\Delta}) - \mathbf{X} - \mathbf{\Delta}\|_\mathsf{F}}{\|\mathbf{\Delta}\|_\mathsf{F}} = 0$ for any $\mathbf{X} \in \mathcal{M}$.*

**Remark 2.9.** *Several retractions on the Stiefel manifold have been explored in literature (Absil & Malick, 2012; Absil et al., 2008b). We present two examples below. (i) Polar Decomposition-Based Retraction: $\mathrm{Retr}_{\mathbf{X}}(\mathbf{\Delta}) = (\mathbf{X} + \mathbf{\Delta})(\mathbf{I}_r + \mathbf{\Delta}^\mathsf{T}\mathbf{\Delta})^{-1/2}$. (ii) QR-Decomposition-Based Retraction: $\mathrm{Retr}_{\mathbf{X}}(\mathbf{\Delta}) = \mathrm{qf}(\mathbf{X} + \mathbf{\Delta})$, where $\mathrm{qf}(\mathbf{X})$ is the $\mathbf{Q}$-factor in the thin QR-decomposition of $\mathbf{X}$.*

The following lemma concerning the retraction operator is useful for our subsequent analysis.

**Lemma 2.10.** *((Boumal et al., 2019)) Let $\mathbf{X} \in \mathcal{M}$ and $\mathbf{\Delta} \in \mathbf{T}_{\mathbf{X}}\mathcal{M}$. There exists positive constants $\{\dot{k}, \ddot{k}\}$ such that $\|\mathrm{Retr}_{\mathbf{X}}(\mathbf{\Delta}) - \mathbf{X}\|_\mathsf{F} \leq \dot{k}\|\mathbf{\Delta}\|_\mathsf{F}$, and $\|\mathrm{Retr}_{\mathbf{X}}(\mathbf{\Delta}) - \mathbf{X} - \mathbf{\Delta}\|_\mathsf{F} \leq \frac{1}{2}\ddot{k}\|\mathbf{\Delta}\|_\mathsf{F}^2$.*

Furthermore, we present the following three insightful lemmas.

**Lemma 2.11.** *(Proof in Appendix B.4) Let $\mathbf{X} \in \mathcal{M}$ and $\mathbf{\Delta} \in \mathbb{R}^{n \times r}$, we have $\|\mathrm{Proj}_{\mathbf{T}_{\mathbf{X}}\mathcal{M}}(\mathbf{\Delta})\|_\mathsf{F} \leq \|\mathbf{\Delta}\|_\mathsf{F}$.*

**Lemma 2.12.** *(Proof in Appendix B.5) Let $\rho > 0$, $\mathbf{G} \in \mathbb{R}^{n \times r}$, and $\mathbf{X} \in \mathcal{M}$. We define $\mathbb{G}_\rho \triangleq \mathbf{G} - \rho\mathbf{X}\mathbf{G}^\mathsf{T}\mathbf{X} - (1-\rho)\mathbf{X}\mathbf{X}^\mathsf{T}\mathbf{G}$. It follows that: (a) $\max(1, 2\rho) \cdot \langle \mathbf{G}, \mathbb{G}_\rho \rangle \geq \|\mathbb{G}_\rho\|_\mathsf{F}^2 \geq \min(1, \rho^2)\|\mathbb{G}_1\|_\mathsf{F}^2$. (b) $\min(1, 2\rho)\|\mathbb{G}_{1/2}\|_\mathsf{F} \leq \|\mathbb{G}_\rho\|_\mathsf{F} \leq \max(1, 2\rho)\|\mathbb{G}_{1/2}\|_\mathsf{F}$.*

**Lemma 2.13.** *(Proof in Appendix B.6) Consider the following optimization problem:* $\min_{\mathbf{X} \in \mathcal{M}} f(\mathbf{X})$, *where* $f(\mathbf{X})$ *is differentiable. For all* $\mathbf{X} \in \mathcal{M}$, *we have:* $\mathrm{dist}(\mathbf{0}, \partial I_{\mathcal{M}}(\mathbf{X}) + \nabla f(\mathbf{X})) \leq \|\nabla f(\mathbf{X}) - \mathbf{X} \nabla f(\mathbf{X})^{\mathsf{T}} \mathbf{X}\|_{\mathsf{F}}$.

**Remark 2.14.** *The matrix* $\mathbb{G}_{\rho} \in \mathbb{R}^{n \times r}$ *in Lemma 2.12 is closely related to the search descent direction of the proposed* OADMM-RR *algorithm. While one can set* $\rho$ *to typical values such as* $1$ *or* $1/2$, *we consider the setting* $\rho \in (0, \infty)$ *to enhance the versatility of* OADMM-RR, *aligning with (Liu et al., 2016; Jiang & Dai, 2015).*

## 3  THE PROPOSED OADMM ALGORITHM

This section provides the proposed OADMM algorithm for solving Problem (1), featuring two variants, one is based on Euclidean Projection (OADMM-EP) and the other on Riemannian Retraction (OADMM-RR).

Using the Moreau envelope smoothing technique, we consider the following optimization problem:

$$\min_{\mathbf{X}, \mathbf{y}} f(\mathbf{X}) - g(\mathbf{X}) + h_{\mu}(\mathbf{y}) + \mathcal{I}_{\mathcal{M}}(\mathbf{X}), \ s.t. \ \mathcal{A}(\mathbf{X}) = \mathbf{y}, \tag{4}$$

where $\mu \to 0$, and $h_{\mu}(\mathbf{y})$ is the Moreau Envelope of $h(\mathbf{y})$. Importantly, $h_{\mu}(\mathbf{y})$ is $(\mu^{-1})$-smooth when $\mu \leq \frac{1}{2W_h}$, according to Lemma 2.2. It is worth noting that similar smoothing techniques have been used in the design of augmented Lagrangian methods (Zeng et al., 2022), and minimax optimization (Zhang et al., 2020a), and ADMMs (Li et al., 2022). We define the augmented Lagrangian function of Problem (4) as follows:

$$\mathcal{L}(\mathbf{X}, \mathbf{y}; \mathbf{z}; \beta, \mu) = \underbrace{f(\mathbf{X}) + \langle \mathbf{z}, \mathcal{A}(\mathbf{X}) - \mathbf{y} \rangle + \tfrac{\beta}{2} \|\mathcal{A}(\mathbf{X}) - \mathbf{y}\|_2^2}_{\triangleq \mathcal{S}(\mathbf{X}, \mathbf{y}; \mathbf{z}; \beta)} - g(\mathbf{X}) + h_{\mu}(\mathbf{y}) + \mathcal{I}_{\mathcal{M}}(\mathbf{X}). \tag{5}$$

Here, $\mathbf{z}$ is the dual variable for the equality constraint, $\mu$ is the smoothing parameter linked to the function $h(\mathbf{y})$, $\beta$ is the penalty parameter associated with the equality constraint, and $\mathcal{I}_{\mathcal{M}}(\mathbf{X})$ is the indicator function of the set $\mathcal{M}$.

In simple terms, OADMM updates are performed by minimizing the augmented Lagrangian function $\mathcal{L}(\mathbf{X}, \mathbf{y}, \mathbf{z}; \beta, \mu)$ over the primal variables $\{\mathbf{X}^t, \mathbf{y}^t\}$ at each iteration, while keeping all other primal and dual variables fixed. The dual variables are updated using gradient ascent on the dual problem.

For updating the primal variable $\mathbf{X}$, we use different strategies, resulting in distinct variants of OADMM. We first observe that the function $\mathcal{S}(\mathbf{X}, \mathbf{y}^t; \mathbf{z}^t; \beta^t)$ is $\ell(\beta^t)$-smooth *w.r.t.* $\mathbf{X}$, where $\ell(\beta^t) \triangleq \beta^t \overline{\mathrm{A}}^2 + L_f$. In OADMM-EP, we adopt a proximal linearized method based on Euclidean projection (Lai & Osher, 2014), while in OADMM-RR, we apply line-search methods on the Stiefel manifold (Liu et al., 2016).

We detail iteration steps of OADMM in Algorithm 1, and have the following remarks.

**(a)** To achieve possible faster dual convergence, we apply an over-relaxation step size with $\sigma \in (1, 2)$ for updating the dual variable $\mathbf{z}$, as suggested by previous studies (Gonçalves et al., 2017; Yang et al., 2017; Li et al., 2016; 2023).

**(b)** To accelerate primal convergence in OADMM-EP, we incorporate a Nesterov extrapolation strategy with parameter $\alpha \in (0, 1)$.

**(c)** To enhance primal convergence in OADMM-RR, we use a Monotone Barzilai-Borwein (MBB) strategy (Wen & Yin, 2013) with a dynamically adjusted parameter $b^t$ to capture the problem's curvature [1]. The parameters $\{\gamma, \delta\}$ represent the decay rate and sufficient decrease parameter, commonly used in line search procedures (Chen et al., 2020).

**(d)** The $\mathbf{X}$-subproblem is solved as: $\mathbf{X}^{t+1} = \arg\min_{\mathbf{X} \in \mathcal{M}} \|\mathbf{X} - \mathbf{X}'\|_{\mathsf{F}}^2 = \dot{\mathbf{U}} \dot{\mathbf{V}}^{\mathsf{T}}$, where $\mathbf{X}' = \mathbf{X}_{\mathsf{c}}^t - \mathbf{G}^t / (\theta \ell(\beta^t))$, and $\dot{\mathbf{U}} \mathrm{diag}(\dot{\mathbf{x}}) \dot{\mathbf{V}}^{\mathsf{T}} = \mathbf{X}'$ is the using singular value decomposition of $\mathbf{X}'$.

**(e)** The $\mathbf{y}$-subproblem can be solved using the result from Lemma 2.5.

**(f)** For practical implementation, we recommend the following default parameters: $p = 1/3$, $\theta = 1.01$, $\sigma = 1.1$, $\rho = 1$, $\gamma = 1/2$, $\delta = 10^{-3}$, $\xi = 1$, $\alpha = \frac{\theta - 1}{(\theta + 1)(\xi + 2)} - 10^{-12}$.

---

[1]Following (Wen & Yin, 2013), one can set $b^t = \langle \mathbf{S}^t, \mathbf{S}^t \rangle / \langle \mathbf{S}^t, \mathbf{Z}^t \rangle$ or $b^t = \langle \mathbf{S}^t, \mathbf{Z}^t \rangle / \langle \mathbf{Z}^t, \mathbf{Z}^t \rangle$, where $\mathbf{S}^t = \mathbf{X}^t - \mathbf{X}^{t-1}$ and $\mathbf{Z}^t = \mathbb{G}_1^{t-1} - \mathbb{G}_1^t$, with $\mathbb{G}_1^t$ being the Riemannian gradient.

---

**Algorithm 1:** OADMM: The Proposed ADMM for Solving Problem (1).

---

**Initialization:**

Choose $\{\mathbf{X}^0, \mathbf{y}^0, \mathbf{z}^0\}$. Choose $p \in (0,1)$, $\xi \in (0, \infty)$, $\theta \in (1, \infty)$, $\sigma \in [1, 2)$.

Choose $\chi \in (1 + 4\omega\ddot{\sigma}, \infty)$, where $\omega \triangleq \frac{1}{\sigma} + \frac{\xi}{2\sigma^2} + \frac{\varepsilon_z}{\sigma^2}$, $\ddot{\sigma} \triangleq (\sigma/(2-\sigma))^2$, $\varepsilon_z = \xi$.

Choose $\beta^0$ sufficiently large such that $\beta^0 \geq 2\chi W_h$.

For OADMM-EP, choose $\alpha \in [0, \frac{\theta-1}{(\theta+1)(\xi+2)})$.

For OADMM-RR, choose $\alpha = 0$, $\rho \in (0, \infty)$, $\gamma \in (0, 1)$, $\delta \in (0, \frac{1}{\max(1,2\rho)})$.

**for** $t$ from $0$ to $T$ **do**

  S1) Set $\beta^t = \beta^0(1 + \xi t^p)$, $\mu^t = \chi/\beta^t$.

  S2) Update the primal variable $\mathbf{X}$: **if** OADMM-EP **then**

    Set $\mathbf{X}_c^t = \mathbf{X}^t + \alpha(\mathbf{X}^t - \mathbf{X}^{t-1})$, $\mathbf{G}^t \in \nabla_{\mathbf{X}} \mathcal{S}(\mathbf{X}_c^t, \mathbf{y}^t; \mathbf{z}^t; \beta^t) - \partial g(\mathbf{X}^t)$.

    $\mathbf{X}^{t+1} \in \arg\min_{\mathbf{X} \in \mathcal{M}} \langle \mathbf{X} - \mathbf{X}^t, \mathbf{G}^t \rangle + \frac{\theta \ell(\beta^t)}{2} \|\mathbf{X} - \mathbf{X}_c^t\|_{\mathsf{F}}^2$, where $\ell(\beta^t) \triangleq \beta^t \overline{\mathbf{A}}^2 + L_f$.

  **end**

  **if** OADMM-RR **then**

    Set $\mathbf{G}^t \in \nabla_{\mathbf{X}} \mathcal{S}(\mathbf{X}^t, \mathbf{y}^t; \mathbf{z}^t; \beta^t) - \partial g(\mathbf{X}^t)$, $\dot{\mathcal{L}}(\mathbf{X}) \triangleq L(\mathbf{X}, \mathbf{y}^t; \mathbf{z}^t; \beta^t, \mu^t)$. Set

    $\mathbb{G}_\rho^t \triangleq \mathbf{G}^t - \rho\mathbf{X}^t[\mathbf{G}^t]^{\mathsf{T}}\mathbf{X}^t - (1-\rho)\mathbf{X}^t[\mathbf{X}^t]^{\mathsf{T}}\mathbf{G}^t$. Set $b^t \in (\underline{b}, \overline{b})$ as the BB step size,

    where $\underline{b}, \overline{b} \in (0, \infty)$. Set $\mathbf{X}^{t+1} = \mathrm{Retr}_{\mathbf{X}^t}(-\eta^t \mathbb{G}_\rho^t)$, where $\eta^t \triangleq \frac{b^t \gamma^j}{\beta^t}$, and

    $j \in \{0, 1, 2, \ldots\}$ is the smallest integer that:

    $\dot{\mathcal{L}}(\mathrm{Retr}_{\mathbf{X}^t}(-\eta^t \mathbb{G}_\rho^t)) - \dot{\mathcal{L}}(\mathbf{X}^t) \leq -\delta\eta^t \|\mathbb{G}_\rho^t\|_{\mathsf{F}}^2$.

  **end**

  S3) Update the primal variable $\mathbf{y}$: $\mathbf{y}^{t+1} = \arg\min_{\mathbf{y}} h_{\mu^t}(\mathbf{y}) + \frac{\beta^t}{2}\|\mathbf{y} - \mathbf{b}\|_2^2$, where

  $\mathbf{b} \triangleq \mathbf{y}^t - \frac{1}{\beta^t} \nabla_{\mathbf{y}} \mathcal{S}(\mathbf{X}^{t+1}, \mathbf{y}^t, \mathbf{z}^t; \beta^t)$. It can be solved as: $\mathbf{y}^{t+1} = \frac{\breve{\mathbf{y}}^{t+1} + \mu^t \beta^t \mathbf{b}}{1 + \mu^t \beta^t}$, where

  $\breve{\mathbf{y}}^{t+1} = \mathbb{P}_{[\mu^t + 1/\beta^t]}(\mathbf{b})$.

  S4) Update the dual variable $\mathbf{z}$: $\mathbf{z}^{t+1} = \mathbf{z}^t + \sigma\beta^t(\mathcal{A}(\mathbf{X}^{t+1}) - \mathbf{y}^{t+1})$

**end**

---

## 4   ORACLE COMPLEXITY

This section details the oracle complexity of Algorithm 1.

We define $\varepsilon_z = \xi$, $\varepsilon_y \triangleq \frac{1}{2}(1 - \frac{1+4\omega\ddot{\sigma}}{\chi})$, $\dot{\sigma} \triangleq (\sigma-1)/(2-\sigma)$, $\ddot{\sigma} \triangleq (\sigma/(2-\sigma))^2$, $\omega \triangleq \frac{1}{\sigma} + \frac{\xi}{2\sigma^2} + \frac{\varepsilon_z}{\sigma^2}$.

We define the potential function (or Lyapunov function) for all $t \geq 1$, as follows:

$$
\begin{aligned}
\Theta^t &\triangleq \Theta(\mathbf{X}^t, \mathbf{X}^{t-1}, \mathbf{y}^t, \mathbf{z}^t; \beta^t, \beta^{t-1}, \mu^{t-1}, t) \\
&\triangleq L(\mathbf{X}^t, \mathbf{y}^t, \mathbf{z}^t; \beta^t, \mu^{t-1}) + \mu^{t-1}C_h^2 + \mathbb{T}^t + \mathbb{Z}^t + \mathbb{X}^t,
\end{aligned} \tag{6}
$$

where $\mathbb{T}^t \triangleq \frac{4\omega\ddot{\sigma}}{\beta^0} C_h^2 \frac{1}{t}$, $\mathbb{Z}^t \triangleq \omega\dot{\sigma}\sigma^2 \beta^{t-1} \|\mathcal{A}(\mathbf{X}^t) - \mathbf{y}^t\|_2^2$, and $\mathbb{X}^t \triangleq \frac{\alpha(\theta+1)\ell(\beta^t)}{2}\|\mathbf{X}^t - \mathbf{X}^{t-1}\|_{\mathsf{F}}^2$.

Additionally, we define:

$$
e^t \triangleq \begin{cases} \|\mathbf{y}^t - \mathbf{y}^{t-1}\| + \|\mathcal{A}(\mathbf{X}^t) - \mathbf{y}^t\| + \|\mathbf{X}^t - \mathbf{X}^{t-1}\|_{\mathsf{F}}, & \text{OADMM-EP}; \\ \|\mathbf{y}^t - \mathbf{y}^{t-1}\| + \|\mathcal{A}(\mathbf{X}^t) - \mathbf{y}^t\| + \|\frac{1}{\beta^t}\mathbb{G}_{1/2}^{t-1}\|_{\mathsf{F}}, & \text{OADMM-RR}. \end{cases} \tag{7}
$$

We have the following useful lemma, derived using the first-order optimality condition of $\mathbf{y}^{t+1}$.

**Lemma 4.1.** *(Proof in Section C.1,* Bounding Dual using Primal*) We have:* **(a)** $\forall t \geq 0$, $\mathbf{z}^t - \frac{1}{\sigma}(\mathbf{z}^t - \mathbf{z}^{t+1}) = \nabla h_{\mu^t}(\mathbf{y}^{t+1}) \in \partial h(\breve{\mathbf{y}}^{t+1})$. **(b)** $\forall t \geq 1$, $\|\mathbf{z}^{t+1} - \mathbf{z}^t\|_2^2 \leq \dot{\sigma}(\|\mathbf{z}^t - \mathbf{z}^{t-1}\|_2^2 - \|\mathbf{z}^{t+1} - \mathbf{z}^t\|_2^2) + 2\ddot{\sigma}(\beta^t/\chi)^2\|\mathbf{y}^{t+1} - \mathbf{y}^t\|_2^2 + 2\ddot{\sigma}C_h^2(\frac{2}{t} - \frac{2}{t+1})$.

**Remark 4.2.** *Here, for* OADMM-RR*, we set* $\alpha = 0$*, resulting in* $\mathbb{X}^t = 0$ *for all* $t$*.* **(i)** *With the choice* $\sigma = 1$*, we have:* $\nabla h_{\mu^{t-1}}(\mathbf{y}^t) = \mathbf{z}^t$*, and* $\|\mathbf{z}^{t+1} - \mathbf{z}^t\| \leq \|\nabla h_{\mu^t}(\mathbf{y}^{t+1}) - \nabla h_{\mu^{t-1}}(\mathbf{y}^t)\|$*.*

**Lemma 4.3.** *(Proof in Appendix C.2)* **(a)** *It holds that* $\beta^{t+1} \leq \beta^t(1 + \xi)$*.* **(b)** *There exists constant* $\{\overline{\ell}, \underline{\ell}\}$ *such that* $\beta^t \underline{\ell} \leq \ell(\beta^t) \leq \beta^t \overline{\ell}$*.*

The subsequent lemma demonstrates that the sequence $\{\Theta^t\}_{t=1}^{\infty}$ is always lower bounded.

**Lemma 4.4.** *(Proof in Section C.3) For all $t \geq 1$, there exists constants $\{\overline{\mathbb{X}}, \overline{z}, \overline{y}, \underline{\Theta}\}$ such that* $\|\mathbf{X}^t\|_{\mathsf{F}} \leq \overline{\mathbb{X}}$, $\|\mathbf{z}^t\| \leq \overline{z}$, $\|\mathbf{y}^t\| \leq \overline{y}$, and $\Theta^t \geq \underline{\Theta}$.

The following lemma is useful for our subsequent analysis, applicable to both OADMM-EP and OADMM-RR.

**Lemma 4.5.** *(Proof in Appendix C.4,* Sufficient Decrease for Variables $\{\mathbf{y}, \mathbf{z}, \beta, \mu\}$*) We have* $L(\mathbf{X}^{t+1}, \mathbf{y}^{t+1}, \mathbf{z}^{t+1}; \beta^{t+1}, \mu^t) - L(\mathbf{X}^{t+1}, \mathbf{y}^t, \mathbf{z}^t; \beta^t, \mu^{t-1}) + (\mu^t - \mu^{t-1})C_h^2 + \mathbb{T}^{t+1} - \mathbb{T}^t + \mathbb{Z}^{t+1} - \mathbb{Z}^t + \varepsilon_z\beta^t\|\mathcal{A}(\mathbf{X}^{t+1}) - \mathbf{y}^{t+1}\|_2^2 \leq -\varepsilon_y\beta^t\|\mathbf{y}^{t+1} - \mathbf{y}^t\|_2^2$.

In the remaining content of this section, we provide separate analyses for OADMM-EP and OADMM-RR.

### 4.1 ANALYSIS FOR OADMM-EP

Using the optimality condition of $\mathbf{X}^{t+1}$, we derive the following lemma.

**Lemma 4.6.** *(Proof in Appendix C.5,* Sufficient Decrease for Variable $\mathbf{X}$*) We define $\varepsilon_x \triangleq \frac{1}{2}\varepsilon_x'\ell$, where $\varepsilon_x' \triangleq \theta - 1 - \alpha(2 + \xi)(1 + \theta) > 0$. We have $L(\mathbf{X}^{t+1}, \mathbf{y}^t, \mathbf{z}^t; \beta^t, \mu^{t-1}) - L(\mathbf{X}^t, \mathbf{y}^t, \mathbf{z}^t; \beta^t, \mu^{t-1}) \leq -\varepsilon_x\beta^t\|\mathbf{X}^{t+1} - \mathbf{X}^t\|_{\mathsf{F}}^2 + \mathbb{X}^t - \mathbb{X}^{t+1}$.*

Combining the results from Lemmas 4.5, and 4.6, we arrive at the following lemma.

**Lemma 4.7.** *(Proof in Appendix C.6) We have: (a)* $\beta^t\{\varepsilon_z\|\mathcal{A}(\mathbf{X}^{t+1}) - \mathbf{y}^{t+1}\|_2^2 + \varepsilon_y\|\mathbf{y}^{t+1} - \mathbf{y}^t\|_2^2 + \varepsilon_x\|\mathbf{X}^{t+1} - \mathbf{X}^t\|_{\mathsf{F}}^2\} \leq \Theta^t - \Theta^{t+1}$. *(b)* $\frac{1}{T}\sum_{t=1}^T \beta^t e^{t+1} \leq \mathcal{O}(T^{(p-1)/2})$.

Finally, we have the following theorem regarding the oracle complexity of OADMM-EP.

**Theorem 4.8.** *(Proof in Appendix C.7) Let $p = 1/3$. We have:* $\frac{1}{T}\sum_{t=1}^T \mathrm{Crit}(\mathbf{X}^{t+1}, \breve{\mathbf{y}}^{t+1}, \mathbf{z}^{t+1}) \leq \mathcal{O}(T^{-1/3})$. *In other words, there exists $\bar{t} \leq T$ such that:* $\frac{1}{T}\sum_{t=1}^T \mathrm{Crit}(\mathbf{X}^{t+1}, \breve{\mathbf{y}}^{t+1}, \mathbf{z}^{t+1}) \leq \epsilon$, *provided that $T \geq \mathcal{O}(1/\epsilon^3)$.*

**Remark 4.9.** *The oracle complexity of* OADMM-EP *matches the best-known complexities currently available to date (Beck & Rosset, 2023; Böhm & Wright, 2021).*

### 4.2 ANALYSIS FOR OADMM-RR

Using the properties of the line search procedure for updating the variable $\mathbf{X}^{t+1}$, we deduce the following lemma.

**Lemma 4.10.** *(Proof in Appendix C.8,* Sufficient Decrease for Variable $\mathbf{X}$*) We define $\varepsilon_x \triangleq \delta\overline{\gamma}\gamma\underline{b}\min(1, 2\rho)^2 > 0$, where $\overline{\gamma} \triangleq 2(1/\max(1, 2\rho) - \delta)/(\bar{\ell}\dot{k}\bar{k}\bar{b} + \overline{g}\ddot{k}\bar{b}/\beta^0) > 0$. We have: (a) For any $t \geq 0$, if $j$ is large enough such that $\gamma^j \in (0, \overline{\gamma})$, then the condition of the line search procedure is satisfied. (b) It follows that:* $L(\mathbf{X}^{t+1}, \mathbf{y}^t, \mathbf{z}^t; \beta^t, \mu^t) - L(\mathbf{X}^t, \mathbf{y}^t, \mathbf{z}^t; \beta^t, \mu^t) \leq -\frac{\varepsilon_x}{\beta^t}\|\mathbb{G}_{1/2}^t\|_{\mathsf{F}}^2$.
Here, $\overline{g}$ is a constant that $\|\mathbf{G}^t\|_{\mathsf{F}} \leq \overline{g}$, $\{\dot{k}, \ddot{k}\}$ are defined in Lemma 2.10, and $\{\rho, \gamma, \delta, \bar{b}, \underline{b}\}$ are defined in Algorithm 1.

**Remark 4.11.** *By Lemma 4.10(a), since $\overline{\gamma}$ is a universal constant and $\gamma^j$ decreases exponentially, the line search procedure of* OADMM-RR *will terminate in $\log(\overline{\gamma})/\log(\gamma) + 1 = \mathcal{O}(1)$ time.*

Combining the results from Lemmas 4.5, and 4.10, we obtain the following lemma.

**Lemma 4.12.** *(Proof in Appendix C.9) We have: (a)* $\beta^t\{\varepsilon_z\|\mathcal{A}(\mathbf{X}^{t+1}) - \mathbf{y}^{t+1}\|_2^2 + \varepsilon_y\|\mathbf{y}^{t+1} - \mathbf{y}^t\|_2^2 + \varepsilon_x\|\frac{1}{\beta^t}\mathbb{G}_{1/2}^t\|_{\mathsf{F}}^2\} \leq \Theta^t - \Theta^{t+1}$. *(b)* $\frac{1}{T}\sum_{t=1}^T \beta^t e^{t+1} \leq \mathcal{O}(T^{(p-1)/2})$.

Finally, we derive the following theorem on the oracle complexity of OADMM-RR.

**Theorem 4.13.** *(Proof in Appendix C.10) Let $p = 1/3$. We have:* $\frac{1}{T}\sum_{t=1}^T \mathrm{Crit}(\mathbf{X}^{t+1}, \breve{\mathbf{y}}^{t+1}, \mathbf{z}^{t+1}) \leq \mathcal{O}(T^{-1/3})$. *In other words, there exists $\bar{t} \leq T$ such that:* $\frac{1}{T}\sum_{t=1}^T \mathrm{Crit}(\mathbf{X}^{t+1}, \breve{\mathbf{y}}^{t+1}, \mathbf{z}^{t+1}) \leq \epsilon$, *provided that $T \geq \mathcal{O}(1/\epsilon^3)$.*

**Remark 4.14.** *Theorem 4.13 mirrors Theorem 4.8, and* OADMM-RR *shares the same oracle complexity as* OADMM-EP.

## 5 CONVERGENCE RATE

This section provides convergence rate of OADMM-EP and OADMM-RR. Our analyses are based on a non-convex analysis tool called KL inequality (Attouch et al., 2010; Bolte et al., 2014; Li & Lin, 2015; Li et al., 2023).

We define the Lyapunov function as: $\Theta(\mathbf{X}, \mathbf{X}^-, \mathbf{y}, \mathbf{z}; \beta, \beta^-, \mu^-, t) \triangleq L(\mathbf{X}, \mathbf{y}, \mathbf{z}; \beta, \mu^-) + \omega\ddot{\sigma}\sigma^2\beta^-\|\mathcal{A}(\mathbf{X}) - \mathbf{y}\|_2^2 + \frac{\alpha(\theta+1)\ell(\beta)}{2}\|\mathbf{X} - \mathbf{X}^-\|_\mathsf{F}^2 + \frac{4\omega\ddot{\sigma}}{\beta^0}C_h^2\frac{1}{t} + C_h^2\mu^-$, where we let $\alpha = 0$ for OADMM-RR. We define $\mathbb{w} \triangleq \{\mathbf{X}, \mathbf{X}^-, \mathbf{y}, \mathbf{z}\}$, $\mathbb{w}^t \triangleq \{\mathbf{X}^t, \mathbf{X}^{t-1}, \mathbf{y}^t, \mathbf{z}^t\}$, $\mathbb{u} \triangleq \{\beta, \beta^-, \mu^-, t\}$, and $\mathbb{u}^t \triangleq \{\beta^t, \beta^{t-1}, \mu^{t-1}, t\}$. Thus, we have $\Theta^t = \Theta(\mathbb{w}^t; \mathbb{u}^t)$. We denote $\mathbb{w}^\infty$ as a limiting point of Algorithm 1.

We make the following additional assumptions.

**Assumption 5.1.** *(Kurdyka-Łojasiewicz Inequality (Attouch et al., 2010)). Consider a semi-algebraic function $\Theta(\mathbb{w}^t; \mathbb{u}^t)$ w.r.t. $\mathbb{w}^t$ for all $t$, where $\mathbb{w}^t$ is in the effective domain of $\Theta(\mathbb{w}^t; \mathbb{u}^t)$. There exist $\tilde{\eta} \in (0, +\infty)$, $\tilde{\sigma} \in [0, 1)$, a neighborhood $\Upsilon$ of $\mathbb{w}^\infty$, and a continuous and concave desingularization function $\varphi(s) \triangleq \tilde{c}s^{1-\tilde{\sigma}}$ with $\tilde{c} > 0$ and $s \in [0, \tilde{\eta})$ such that, for all $\mathbb{w}^t \in \Upsilon$ satisfying $\Theta(\mathbb{w}^t, \mathbb{u}^t) - \Theta(\mathbb{w}^\infty, \mathbb{u}^\infty) \in (0, \tilde{\eta})$, it holds that: $\varphi'(\Theta(\mathbb{w}^t; \mathbb{u}^t) - \Theta(\mathbb{w}^\infty; \mathbb{u}^\infty)) \cdot \mathrm{dist}(\mathbf{0}, \partial\Theta(\mathbb{w}^t; \mathbb{u}^t)) \geq 1$. Here, $\mathrm{dist}(\mathbf{0}, \partial\Theta(\mathbb{w}^t; \mathbb{u}^t)) \triangleq \{\mathrm{dist}^2(\mathbf{0}, \partial_{\mathbf{X}}\Theta(\mathbb{w}^t; \mathbb{u}^t)) + \mathrm{dist}^2(\mathbf{0}, \partial_{\mathbf{X}^-}\Theta(\mathbb{w}^t; \mathbb{u}^t)) + \mathrm{dist}^2(\mathbf{0}, \partial_{\mathbf{y}}\Theta(\mathbb{w}^t; \mathbb{u}^t)) + \mathrm{dist}^2(\mathbf{0}, \partial_{\mathbf{z}}\Theta(\mathbb{w}^t; \mathbb{u}^t))\}^{1/2}$.*

**Assumption 5.2.** *The function $g(\mathbf{X})$ is $L_g$-smooth such that $\|\nabla g(\mathbf{X}) - \nabla g(\mathbf{X}')\|_\mathsf{F} \leq L_g\|\mathbf{X} - \mathbf{X}'\|_\mathsf{F}$ holds for all $\mathbf{X} \in \mathcal{M}$ and $\mathbf{X}' \in \mathcal{M}$.*

**Remark 5.3.** *Semi-algebraic functions, including real polynomial functions, finite combinations, and indicator functions of semi-algebraic sets, commonly exhibit the KL property and find extensive use in applications (Attouch et al., 2010).*

We present the following lemma regarding subgradient bounds for each iteration.

**Lemma 5.4.** *(Proof in Section D.1, Subgradient Bounds) (a) For OADMM-EP, there exists a constant $K > 0$ such that: $\mathrm{dist}(\mathbf{0}, \partial\Theta(\mathbb{w}^t; \mathbb{u}^t)) \leq \beta^t K(e^t + e^{t-1})$. (b) For OADMM-RR, there exists a constant $K > 0$ such that: $\mathrm{dist}(\mathbf{0}, \partial\Theta(\mathbb{w}^t; \mathbb{u}^t)) \leq \beta^t Ke^t$.*

**Remark 5.5.** *Lemma 5.4 significantly differs from prior work that used a constant penalty due to the crucial role played by the increasing penalty.*

The following theorem establishes a finite length property of OADMM.

**Theorem 5.6.** *(Proof in Section D.2, A Finite Length Property) We define $d^t \triangleq \sum_{i=t}^\infty e^{i+1}$. We define $\varphi^t \triangleq \varphi(\Theta(\mathbb{w}^t; \mathbb{u}^t) - \Theta(\mathbb{w}^\infty; \mathbb{u}^\infty))$, where $\varphi(\cdot)$ is the desingularization function defined in Assumption 5.1. (a) We have the following recursive inequality for both OADMM-EP and OADMM-RR: $(e^{t+1})^2 \leq (e^t + e^{t-1}) \cdot \dot{K}(\varphi^t - \varphi^{t+1})$, where $\dot{K} = \frac{3K}{\min(\varepsilon_z, \varepsilon_y, \varepsilon_x)}$, and $K$ is defined in Lemma 5.4. (b) It holds that $\forall t \geq 1$, $d^t \leq e^t + e^{t-1} + 4\dot{K}\varphi^t$. The sequence $\{\mathbb{w}^t\}_{t=1}^\infty$ has the finite length property that $d^1 \leq e^1 + e^0 + 4\dot{K}\varphi^1 < +\infty$.*

**Remark 5.7.** *The finite length property in Theorem 5.6 represents much stronger convergence results compared to those outlined in Theorems 4.8 and 4.13.*

We prove a lemma demonstrating that the convergence of $d^t \triangleq \sum_{i=t}^\infty e^{i+1}$ is sufficient to establish the convergence of $\|\mathbf{X}^t - \mathbf{X}^\infty\|_\mathsf{F}$.

**Lemma 5.8.** *(Proof in Section D.3) We define $d^t \triangleq \sum_{i=t}^\infty e^{i+1}$. For both OADMM-EP and OADMM-RR, we have: (a) There exists a constant $\ddot{c}$ such that $\|\mathbf{X}^t - \mathbf{X}^\infty\|_\mathsf{F} \leq \ddot{c} \cdot d^t$. (b) We have $d^t \leq d^{t-2} - d^t + \ddot{K}[\beta^t(d^{t-2} - d^t)]^{\frac{1-\tilde{\sigma}}{\tilde{\sigma}}}$, where $\ddot{K} \triangleq 4\dot{K}\tilde{c} \cdot [\tilde{c}(1 - \tilde{\sigma})K]^{\frac{1-\tilde{\sigma}}{\tilde{\sigma}}}$.*

Finally, we establish the convergence rate of OADMM with exploiting the KL exponent $\tilde{\sigma}$.

**Theorem 5.9.** *(Proof in Section D.4, Convergence Rate) We fix $p = 1/3$. There exists $t'$ such that for all $t \geq t'$, we have:*

*(a) If $\tilde{\sigma} \in (\frac{1}{4}, \frac{1}{2}]$, then we have $\|\mathbf{X}^t - \mathbf{X}^\infty\|_\mathsf{F} \leq \mathcal{O}(1/\exp(t^{1-u}))$, where $u = \frac{p(1-\tilde{\sigma})}{\tilde{\sigma}} \in [\frac{1}{3}, 1)$.*

*(b) If $\tilde{\sigma} \in (\frac{1}{2}, 1)$, then we have: $\|\mathbf{X}^t - \mathbf{X}^\infty\|_\mathsf{F} \leq \mathcal{O}(1/(t^{(1-p)/\tau}))$, where $\tau = \frac{\tilde{\sigma}}{1-\tilde{\sigma}} - 1 \in (0, \infty)$.*

**Remark 5.10.** *(i) To the best of our knowledge, Theorem 5.9 represents the first non-ergodic convergence rate for solving this class of nonconvex and nonsmooth problem in Problem (1). It is worth noting that the work of (Li et al., 2023) establishes a non-ergodic convergence rate for subgradient methods with diminishing stepsizes by further exploring the KL exponent. (ii) Under the KL inequality assumption, with the desingularizing function chosen in the form of $\varphi(s) \triangleq \tilde{c}s^{1-\tilde{\sigma}}$ with $\tilde{\sigma} \in (0,1)$,* OADMM *converges with a super-exponential rate when $\tilde{\sigma} \in (\frac{1}{4}, \frac{1}{2}]$, and converges with a polynomial convergence rate when $\tilde{\sigma} \in (\frac{1}{2}, 1)$ for the gap $\|\mathbf{X}^t - \mathbf{X}^\infty\|_\mathsf{F}$. Notably, super-exponential convergence is faster than polynomial convergence. (iii) Our result generalizes the classical findings of (Attouch et al., 2010; Bolte et al., 2014), which characterize the convergence rate of proximal gradient methods for a specific class of nonconvex composite optimization problems.*

## 6    APPLICATIONS AND NUMERICAL EXPERIMENTS

In this section, we assess the effectiveness of the proposed algorithm OADMM on the sparse PCA problem by comparing it against existing non-convex, non-smooth optimization algorithms.

▶ **Application to Sparse PCA**. Sparse PCA is a method to produce modified principal components with sparse loadings, which helps reduce model complexity and increase model interpretation (Chen et al., 2016). It can be formulated as:

$$\min_{\mathbf{X} \in \mathbb{R}^{n \times r}} \frac{1}{2\dot{m}}\|\mathbf{X}\mathbf{X}^\mathsf{T}\mathbf{D} - \mathbf{D}\|_\mathsf{F}^2 + \dot{\rho}(\|\mathbf{X}\|_1 - \|\mathbf{X}\|_{[k]}), \ s.t. \ \mathbf{X}^\mathsf{T}\mathbf{X} = \mathbf{I}_r,$$

where $\mathbf{D} \in \mathbb{R}^{n \times \dot{m}}$ is the data matrix, $\dot{m}$ is the number of data points, and $\|\mathbf{X}\|_{[k]}$ is the $\ell_1$ norm the the $k$ largest (in magnitude) elements of the matrix $\mathbf{X}$. Here, we consider the DC $\ell_1$-largest-$k$ function (Gotoh et al., 2018) to induce sparsity in the solution. One advantage of this model is that when $\dot{\rho}$ is sufficient large, we have $\|\mathbf{X}\|_1 \approx \|\mathbf{X}\|_{[k]}$, leading to a $k$-sparsity solution $\mathbf{X}$.

▶ **Compared Methods**. We compare OADMM-EP and OADMM-RR against four state-of-the-art optimization algorithms: (*i*) RADMM: ADMM using Riemannian retraction with fixed and small stepsizes (Li et al., 2022), tested with two different penalty parameters $\forall t, \beta^t \in \{100, 10000\}$, leading to two variants: RADMM-I and RADMM-II. (*ii*) SPGM-EP: Smoothing Proximal Gradient Method using Euclidean projection (Böhm & Wright, 2021). (*iii*) SPGM-EP: SPGM utilizing Riemannian retraction (Beck & Rosset, 2023). (*iv*) Sub-Grad: Subgradient methods with Euclidean projection (Davis & Drusvyatskiy, 2019; Li et al., 2021).

▶ **Experiment Settings**. All methods are implemented in MATLAB on an Intel 2.6 GHz CPU with 64 GB RAM. For all retraction-based methods, we use only polar decomposition-based retraction. We evaluate different regularization parameters $\dot{\rho} \in \{10, 50, 100, 500, 1000\}$. For OADMM, default parameters are used, with $\beta^0 = 10\dot{\rho}$ and corresponding values $\xi = \{1, 2, 5, 8, 10\}$ for each $\dot{\rho}$. For simplicity, we omit the Barzilai-Borwein strategy and instead use a fixed constant $b^t = 1$ for all iterations. All algorithms start with a common initial solution $\mathbf{x}^0$, generated from a standard normal distribution. Our code for reproducing the experiments is available in the **supplemental material**.

▶ **Experiment Results**. We report the objective values for different methods with varying parameters $\dot{\rho}$. The experimental results presented in Figures 1 and 2 reveal the following insights: (*i*) Sub-Grad essentially fails to solve this problem, as the subgradient is inaccurately estimated when the solution is sparse. (*ii*) SPGM-EP and SPGM-RR, which rely on a variable smoothing strategy, exhibit slower performance than the multiplier-based variable splitting method. This observation aligns with the commonly accepted notion that primal-dual methods are generally more robust and faster than primal-only methods. (*iii*) The proposed OADMM-EP and OADMM-RR demonstrate similar results and generally achieve lower objective function values than the other methods.

## 7    CONCLUSIONS

This paper introduces OADMM, an Alternating Direction Method of Multipliers (ADMM) tailored for solving structured nonsmooth composite optimization problems under orthogonality constraints. OADMM integrates either a Nesterov extrapolation strategy or a Monotone Barzilai-Borwein (MBB) stepsize strategy to potentially accelerate primal convergence, complemented by an over-relaxation stepsize strategy for rapid dual convergence. We adjust the penalty and smoothing parameters at

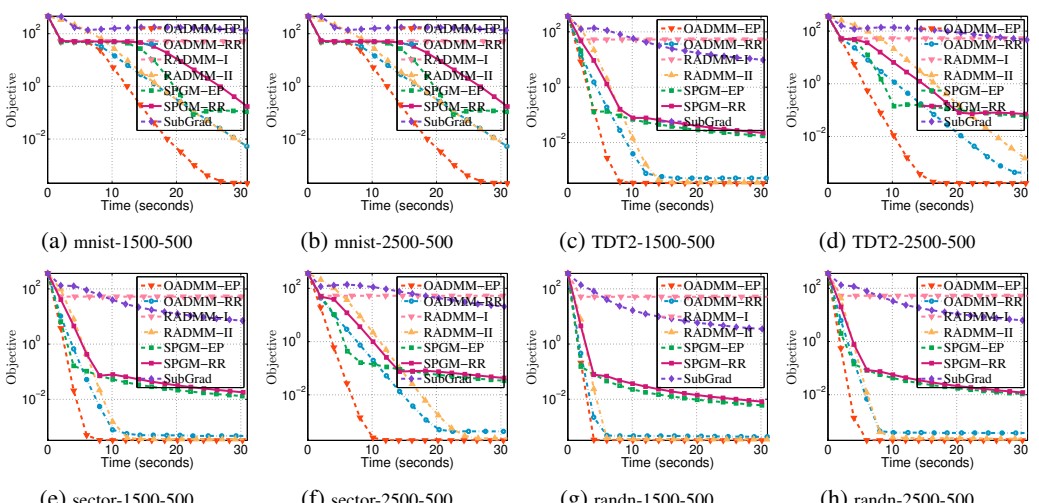

Figure 1: The convergence curve of the compared methods with $\dot{\rho} = 50$.

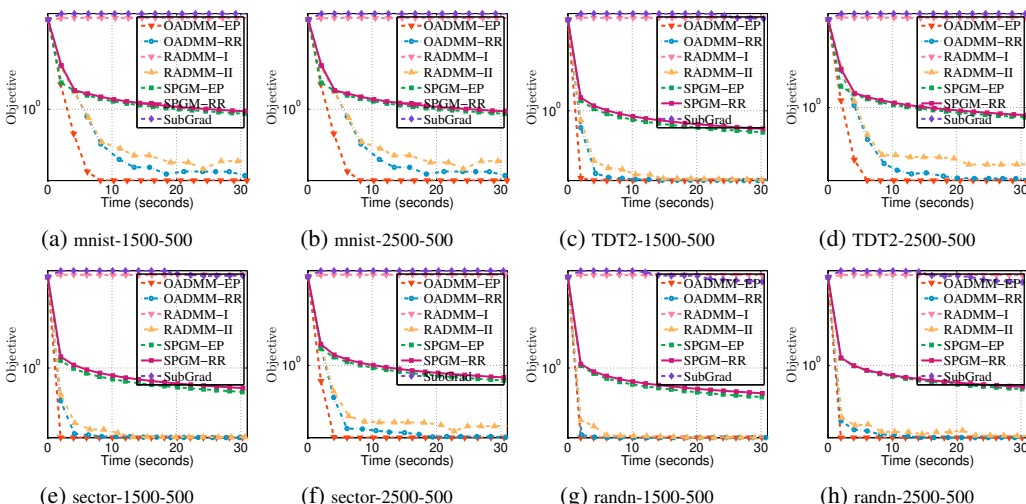

Figure 2: The convergence curve of the compared methods with $\dot{\rho} = 500$.

a controlled rate. Additionally, we develop a novel Lyapunov function to rigorously analyze the oracle complexity of OADMM and establish the first non-ergodic convergence rate for this method. Finally, numerical experiments show that our OADMM achieves state-of-the-art performance.

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

# Appendix

The appendix is organized as follows.

Appendix A provides notations, technical preliminaries, and relevant lemmas.

Appendix B contains the proofs for Section 2.

Appendix C includes the proofs for Section 4.

Appendix D encompasses the proofs for Section 5.

Appendix E presents additional experiments details and results.

## A NOTATIONS, TECHNICAL PRELIMINARIES, AND RELEVANT LEMMAS

### A.1 NOTATIONS

In this paper, lowercase boldface letters signify vectors, while uppercase letters denote real-valued matrices. The following notations are utilized throughout this paper.

- $[n]$: $\{1, 2, ..., n\}$
- $\|\mathbf{x}\|$: Euclidean norm: $\|\mathbf{x}\| = \|\mathbf{x}\|_2 = \sqrt{\langle \mathbf{x}, \mathbf{x} \rangle}$
- $\mathbf{X}^\mathsf{T}$ : the transpose of the matrix $\mathbf{X}$
- $\mathbf{0}_{n,r}$ : A zero matrix of size $n \times r$; the subscript is omitted sometimes
- $\mathbf{I}_r$ : $\mathbf{I}_r \in \mathbb{R}^{r \times r}$, Identity matrix
- $\mathcal{M}$: Orthogonality constraint set (a.k.a., Stiefel manifold: $\mathcal{M} = \{\mathbf{X} \in \mathbb{R}^{n \times r} \,|\, \mathbf{X}^\mathsf{T}\mathbf{X} = \mathbf{I}_r\}$.
- $\mathbf{X} \succeq \mathbf{0}$(or $\succ \mathbf{0}$) : the Matrix $\mathbf{X}$ is symmetric positive semidefinite (or definite)
- $\mathrm{tr}(\mathbf{A})$ : Sum of the elements on the main diagonal $\mathbf{A}$: $\mathrm{tr}(\mathbf{A}) = \sum_i \mathbf{A}_{i,i}$
- $\|\mathbf{X}\|$ : Operator/Spectral norm: the largest singular value of $\mathbf{X}$
- $\|\mathbf{X}\|_\mathsf{F}$ : Frobenius norm: $(\sum_{ij} \mathbf{X}_{ij}^2)^{1/2}$
- $\|\mathbf{X}\|_1$: Absolute sum of the elements in $\mathbf{X}$ with $\mathbf{X} = \sum_{ij} |\mathbf{X}_{ij}|$
- $\|\mathbf{X}\|_{[k]}$: $\ell_1$ norm the the $k$ largest (in magnitude) elements of the matrix $\mathbf{X}$
- $\partial g(\mathbf{X})$ : (limiting) Euclidean subdifferential of $g(\mathbf{X})$ at $\mathbf{X}$
- $\mathrm{Proj}_\Xi(\mathbf{X}')$ : Orthogonal projection of $\mathbf{X}'$ with $\mathrm{Proj}_\Xi(\mathbf{X}') = \arg\arg\min_{\mathbf{X} \in \Xi} \|\mathbf{X}' - \mathbf{X}\|_\mathsf{F}^2$
- $\mathrm{dist}(\Xi, \Xi')$ : the distance between two sets with $\mathrm{dist}(\Xi, \Xi') \triangleq \inf_{\mathbf{X} \in \Xi, \mathbf{X}' \in \Xi'} \|\mathbf{X} - \mathbf{X}'\|_\mathsf{F}$
- $\|\partial g(\mathbf{X})\|_\mathsf{F}$: $\|\partial g(\mathbf{X})\|_\mathsf{F} = \inf_{\mathbf{Y} \in \partial g(\mathbf{X})} \|\mathbf{Y}\|_\mathsf{F} = \mathrm{dist}(\mathbf{0}, \partial g(\mathbf{X}))$.
- $\ell(\beta^t)$: the smoothness parameter of the function $\mathcal{S}(\mathbf{X}, \mathbf{y}^t; \mathbf{z}^t; \beta^t)$ *w.r.t.* $\mathbf{X}$.
- $\mathcal{I}_\mathcal{M}(\mathbf{x})$ : Indicator function of $\mathcal{M}$ with $\mathcal{I}_\mathcal{M}(\mathbf{x}) = 0$ if $\mathbf{x} \in \mathcal{M}$ and otherwise $+\infty$.

We employ the following parameters in Algorithm 1.

- $\theta$: proximal parameter
- $\chi$: correlation coefficient between $\mu^t$ and $\beta^t$, such that $\mu^t \beta^t = \chi$
- $\sigma$: over-relaxation parameter with $\sigma \in [1, 2)$
- $\alpha$: Nesterov extrapolation parameter with $\alpha \in [0, 1)$
- $\rho$: search descent parameter with $\rho \in (0, \infty)$
- $\gamma$: decay rate parameter in the line search procedure with $\gamma \in (0, 1)$
- $\delta$: sufficient decrease parameter in the line search procedure with $\delta \in (0, \infty)$
- $p$: exponent parameter used in the penalty update rule with $p \in (0, 1)$
- $\xi$: growth factor parameter used in the penalty update rule with $\xi \in (0, \infty)$

## A.2 Technical Preliminaries

**Non-convex Non-smooth Optimization**. Given the potential non-convexity and non-smoothness of the function $F(\cdot)$, we introduce tools from non-smooth analysis (Mordukhovich, 2006; Rockafellar & Wets., 2009). The domain of any extended real-valued function $F : \mathbb{R}^{n \times r} \to (-\infty, +\infty]$ is defined as $\text{dom}(F) \triangleq \{\mathbf{X} \in \mathbb{R}^{n \times r} : |F(\mathbf{X})| < +\infty\}$. At $\mathbf{X} \in \text{dom}(F)$, the Fréchet subdifferential of $F$ is defined as $\hat{\partial} F(\mathbf{X}) \triangleq \{\boldsymbol{\xi} \in \mathbb{R}^{n \times r} : \lim_{\mathbf{Z} \to \mathbf{X}} \inf_{\mathbf{Z} \neq \mathbf{X}} \frac{F(\mathbf{Z}) - F(\mathbf{X}) - \langle \boldsymbol{\xi}, \mathbf{Z} - \mathbf{X} \rangle}{\|\mathbf{Z} - \mathbf{X}\|_{\mathrm{F}}} \geq 0\}$, while the limiting subdifferential of $F(\mathbf{X})$ at $\mathbf{X} \in \text{dom}(F)$ is denoted as $\partial F(\mathbf{X}) \triangleq \{\boldsymbol{\xi} \in \mathbb{R}^n : \exists \mathbf{X}^t \to \mathbf{X}, F(\mathbf{X}^t) \to F(\mathbf{X}), \boldsymbol{\xi}^t \in \hat{\partial} F(\mathbf{X}^t) \to \boldsymbol{\xi}, \forall t\}$. The gradient of $F(\cdot)$ at $\mathbf{X}$ in the Euclidean space is denoted as $\nabla F(\mathbf{X})$. The following relations hold among $\hat{\partial} F(\mathbf{X})$, $\partial F(\mathbf{X})$, and $\nabla F(\mathbf{X})$: (***i***) $\hat{\partial} F(\mathbf{X}) \subseteq \partial F(\mathbf{X})$. (***ii***) If the function $F(\cdot)$ is convex, $\partial F(\mathbf{X})$ and $\hat{\partial} F(\mathbf{X})$ represent the classical subdifferential for convex functions, i.e., $\partial F(\mathbf{X}) = \hat{\partial} F(\mathbf{X}) = \{\boldsymbol{\xi} \in \mathbb{R}^{n \times r} : F(\mathbf{Z}) \geq F(\mathbf{X}) + \langle \boldsymbol{\xi}, \mathbf{Z} - \mathbf{X} \rangle, \forall \mathbf{Z} \in \mathbb{R}^{n \times r}\}$. (***iii***) If the function $F(\cdot)$ is differentiable, then $\hat{\partial} F(\mathbf{X}) = \partial F(\mathbf{X}) = \{\nabla F(\mathbf{X})\}$.

**Optimization with Orthogonality Constraints**. We introduce some prior knowledge of optimization involving orthogonality constraints (Absil et al., 2008b). The nearest orthogonality matrix to any arbitrary matrix $\mathbf{Y} \in \mathbb{R}^{n \times r}$ is determined as $\mathbb{P}_{\mathcal{M}}(\mathbf{Y}) = \breve{\mathbf{U}} \breve{\mathbf{V}}^{\mathsf{T}}$, where $\mathbf{Y} = \breve{\mathbf{U}} \text{Diag}(\mathbf{s}) \breve{\mathbf{V}}^{\mathsf{T}}$ represents the singular value decomposition of $\mathbf{Y}$. We use $\mathcal{N}_{\mathcal{M}}(\mathbf{X})$ to denote the limiting normal cone to $\mathcal{M}$ at $\mathbf{X}$, thus defined as $\mathcal{N}_{\mathcal{M}}(\mathbf{X}) = \partial \mathcal{I}_{\mathcal{M}}(\mathbf{X}) = \{\mathbf{Z} \in \mathbb{R}^{n \times r} : \langle \mathbf{Z}, \mathbf{X} \rangle \geq \langle \mathbf{Z}, \mathbf{Y} \rangle, \forall \mathbf{Y} \in \mathcal{M}\}$. Moreover, the tangent and normal space to $\mathcal{M}$ at $\mathbf{X} \in \mathcal{M}$ are respectively denoted as $\mathrm{T}_{\mathbf{X}} \mathcal{M}$ and $\mathrm{N}_{\mathbf{X}} \mathcal{M}$. We have: $\mathrm{T}_{\mathbf{X}} \mathcal{M} = \{\mathbf{Y} \in \mathbb{R}^{n \times r} | \mathcal{A}_X(\mathbf{Y}) = \mathbf{0}\}$ and $\mathrm{N}_{\mathbf{X}} \mathcal{M} = 2\mathbf{X}\boldsymbol{\Lambda} | \boldsymbol{\Lambda} = \boldsymbol{\Lambda}^{\mathsf{T}}, \boldsymbol{\Lambda} \in \mathbb{R}^{r \times r}\}$, where $\mathcal{A}_{\mathbf{X}}(\mathbf{Y}) \triangleq \mathbf{X}^{\mathsf{T}}\mathbf{Y} + \mathbf{Y}^{\mathsf{T}}\mathbf{X}$ for $\mathbf{Y} \in \mathbb{R}^{n \times r}$ and $\mathbf{X} \in \mathcal{M}$.

**Weakly Convex Functions**. The function $h(\mathbf{y})$ is weakly convex if there exists a constant $W_h \geq 0$ such that $h(\mathbf{y}) + \frac{1}{2}W_h\|\mathbf{y}\|_2^2$ is convex; the smallest such $W_h$ is termed the modulus of weak convexity. Weakly convex functions encompass a diverse range, including convex functions, differentiable functions with Lipschitz continuous gradient, and compositions of convex, Lipschitz-continuous functions with $C^1$-smooth mappings having Lipschitz continuous Jacobians (Drusvyatskiy & Paquette, 2019).

## A.3 Relevant Lemmas

**Lemma A.1.** *Let* $\mathbf{a}, \mathbf{b} \in \mathbb{R}^n$, *and* $\alpha \geq 0$ *be any constant. We have:* $-\|\mathbf{a} - \alpha\mathbf{b}\|_2^2 \leq (\alpha - 1)\|\mathbf{a}\|_2^2 - (\alpha^2 - \alpha)\|\mathbf{b}\|_2^2$.

*Proof.* We have: $-\|\mathbf{a} - \alpha\mathbf{b}\|_2^2 = -\|\mathbf{a}\|_2^2 - \|\alpha\mathbf{b}\|_2^2 + 2\alpha\langle\mathbf{a}, \mathbf{b}\rangle \leq -\|\mathbf{a}\|_2^2 - \|\alpha\mathbf{b}\|_2^2 + 2\alpha \cdot (\frac{1}{2}\|\mathbf{a}\|_2^2 + \frac{1}{2}\|\mathbf{b}\|_2^2) = (\alpha - 1)\|\mathbf{a}\|_2^2 - (\alpha^2 - \alpha)\|\mathbf{b}\|_2^2$. $\square$

**Lemma A.2.** *Assume* $t \geq 1$ *and* $p \in (0, 1)$. *We have:* $\frac{t^p - (t-1)^p}{1 + (t-1)^p} \leq \frac{1}{t}$.

*Proof.* We let $t \geq 1$ and $p \in (0, 1)$.

First, we define $f(t) \triangleq t^p - 2(t-1)^p - 1$. We have $\nabla f(t) = pt^p - 2p(t-1)^{p-1} = p(t-1)^{p-1}\{(\frac{t}{t-1})^{p-1} - 2\} \leq p(t-1)^{p-1}\{(\frac{t}{t-1}) - 2\} \leq p(t-1)^{p-1}\{\frac{1}{2} - 2\} \leq 0$. This implies that $f(t)$ is decreasing. Noting that $f(1) = 0$, we conclude that

$$f(t) \triangleq t^p - 2(t-1)^p - 1 \leq 0. \tag{8}$$

Second, we have:

$$g(t) \triangleq t^p - \frac{t^p + 1}{t + 1} - (t-1)^p \overset{\text{①}}{\leq} t^p - \frac{t^p}{t} - (t-1)^p = t^p((1 - \frac{1}{t}) - (1 - \frac{1}{t})^p) \overset{\text{②}}{\leq} 0, \tag{9}$$

where step ① uses $\frac{t^p + 1}{t + 1} \geq \frac{t^p}{t}$ as $t \geq t^p$; step ② uses $a \leq a^p$ for all $a \in (0, 1)$ and $p \in (0, 1)$.

Finally, we derive the following results:

$$\frac{t^p - (t-1)^p}{1 + (t-1)^p} \cdot t \overset{\text{①}}{\leq} \frac{t}{t+1} \cdot \frac{1 + t^p}{1 + (t-1)^p} \overset{\text{②}}{\leq} \frac{1 + t^p}{2 + 2(t-1)^p} \overset{\text{③}}{\leq} 1,$$

where step ① uses Inequality (9); step ② uses $\frac{t}{t+1} \leq \frac{1}{2}$; step ③ uses Inequality (8).

$\square$

**Lemma A.3.** *Let $\beta^t = \beta^0(1 + \xi t^p)$, where $t \geq 0$, $\beta^0 > 0$, $\xi, p \in (0,1)$. For all $t \geq 1$, we have:* $(\frac{\beta^t}{\beta^{t-1}} - 1)^2 \leq \frac{2}{t} - \frac{2}{t+1}$.

*Proof.* We derive: $(\frac{\beta^t}{\beta^{t-1}} - 1)^2 \overset{①}{=} (\frac{1+\xi t^p}{1+\xi(t-1)^p} - 1)^2 = (\frac{\xi t^p - \xi(t-1)^p}{1+\xi(t-1)^p})^2 \overset{②}{\leq} (\frac{t^p - (t-1)^p}{1+(t-1)^p})^2 \overset{③}{\leq} (\frac{1}{t})^2 \overset{④}{\leq} \frac{2}{t} - \frac{2}{t+1}$, where step ① uses $\beta^t = \beta^0(1 + \xi t^p)$; step ② uses $\frac{\xi}{1+\xi a} < \frac{1}{1+a}$ for all $a \geq 0$ when $\xi \in (0,1)$; step ③ uses Lemma A.2; step ④ uses the fact that $\frac{1}{t^2} \leq \frac{2}{t} - \frac{2}{t+1}$ for all $t \geq 1$.

$\square$

**Lemma A.4.** *Assume $\mathbf{a}^+ = \varrho\mathbf{a} + \mathbf{b}$, where $\mathbf{a}, \mathbf{b}, \mathbf{a}^+ \in \mathbb{R}^m$, and $\varrho \in [0,1)$. We have:* $\|\mathbf{a}^+\|_2^2 \leq \frac{\varrho}{1-\varrho}(\|\mathbf{a}\|_2^2 - \|\mathbf{a}^+\|_2^2) + \frac{1}{(1-\varrho)^2}\|\mathbf{b}\|_2^2$.

*Proof.* We have: $\|\mathbf{a}^+\|_2^2 = \|\varrho\mathbf{a} + \mathbf{b}\|_2^2 = \|\varrho\mathbf{a} + (1-\varrho) \cdot \frac{\mathbf{b}}{1-\varrho}\|_2^2 \leq \varrho\|\mathbf{a}\|_2^2 + (1-\varrho) \cdot \|\frac{\mathbf{b}}{1-\varrho}\|_2^2 = \varrho\|\mathbf{a}\|_2^2 + \frac{1}{1-\varrho}\|\mathbf{b}\|_2^2$, where the inequality holds due to the convexity of $\|\cdot\|_2^2$.

$\square$

**Lemma A.5.** *Assume that $\mathbf{a}^t \leq \varrho\mathbf{a}^{t-1} + c$, where $\varrho \in [0,1)$, $c \geq 0$, and $\{\mathbf{a}^i\}_{i=0}^\infty$ is a non-negative sequence. We have:* $\mathbf{a}^t \leq \mathbf{a}^0 + \frac{c}{1-\varrho}$ *for all $t \geq 0$.*

*Proof.* Using basic induction, we have the following results:

$$
\begin{aligned}
t = 1, \quad & \mathbf{a}^1 \leq \varrho\mathbf{a}^0 + c \\
t = 2, \quad & \mathbf{a}^2 \leq \varrho\mathbf{a}^1 + c \leq \varrho(\varrho\mathbf{a}^0 + c) + c = \varrho^2\mathbf{a}^0 + c(1 + \varrho) \\
t = 3, \quad & \mathbf{a}^3 \leq \varrho\mathbf{a}^2 + c \leq \varrho(\varrho^2\mathbf{a}^0 + (c + \varrho c)) + c = \varrho^3\mathbf{a}^0 + c(1 + \varrho + \varrho^2) \\
& \quad \cdots \\
t = n, \quad & \mathbf{a}^n \leq \varrho\mathbf{a}^{n-1} + c \leq \varrho^n\mathbf{a}^0 + c \cdot (1 + \varrho + \ldots + \varrho^{n-1}).
\end{aligned}
$$

Therefore, we obtain: $\mathbf{a}^n \leq \varrho^n\mathbf{a}^0 + c \cdot (1 + \varrho + \ldots + \varrho^{n-1}) \overset{①}{\leq} a_0 + \frac{c}{1-\varrho}$, where step ① uses $\rho^n \leq \rho < 1$, and the summation formula of geometric sequences that $1 + \varrho^1 + \varrho^2 + \ldots + \varrho^{t-1} = \frac{1-\varrho^t}{1-\varrho} < \frac{1}{1-\varrho}$.

$\square$

**Lemma A.6.** *Assume $\mathbf{X}_{\mathsf{c}}^t = \mathbf{X}^t + \alpha(\mathbf{X}^t - \mathbf{X}^{t-1})$, where $\alpha \in [0,1)$, and $\mathbf{X}^t, \mathbf{X}^{t-1} \in \mathcal{M}$. We have:*

*(a)* $\|\mathbf{X}^t - \mathbf{X}_{\mathsf{c}}^t\|_{\mathsf{F}} \leq \|\mathbf{X}^t - \mathbf{X}^{t-1}\|_{\mathsf{F}}$.

*(b)* $\|\mathbf{X}^{t+1} - \mathbf{X}_{\mathsf{c}}^t\|_{\mathsf{F}} \leq \|\mathbf{X}^{t+1} - \mathbf{X}^t\|_{\mathsf{F}} + \|\mathbf{X}^t - \mathbf{X}^{t-1}\|_{\mathsf{F}}$.

*(c)* $\|\mathcal{A}(\mathbf{X}_{\mathsf{c}}^t) - \mathbf{y}^t\| \leq \|\mathcal{A}(\mathbf{X}^t) - \mathbf{y}^t\| + \overline{\mathrm{A}}\|\mathbf{X}^t - \mathbf{X}^{t-1}\|_{\mathsf{F}}$.

*Proof.* **Part (a).** We have: $\|\mathbf{X}^t - \mathbf{X}_{\mathsf{c}}^t\|_{\mathsf{F}} \overset{①}{=} \alpha\|\mathbf{X}^t - \mathbf{X}^{t-1}\|_{\mathsf{F}} \overset{②}{\leq} \|\mathbf{X}^t - \mathbf{X}^{t-1}\|_{\mathsf{F}}$, where step ① uses $\mathbf{X}_{\mathsf{c}}^t = \mathbf{X}^t + \alpha(\mathbf{X}^t - \mathbf{X}^{t-1})$; step ② uses $\alpha \in [0,1)$.

**Part (b).** We have: $\|\mathbf{X}^{t+1} - \mathbf{X}_{\mathsf{c}}^t\|_{\mathsf{F}} \overset{①}{=} \|\mathbf{X}^{t+1} - \mathbf{X}^t - \alpha(\mathbf{X}^t - \mathbf{X}^{t-1})\|_{\mathsf{F}} \overset{②}{\leq} \|\mathbf{X}^{t+1} - \mathbf{X}^t\|_{\mathsf{F}} + \|\mathbf{X}^t - \mathbf{X}^{t-1}\|_{\mathsf{F}}$, where step ① uses $\mathbf{X}_{\mathsf{c}}^t = \mathbf{X}^t + \alpha(\mathbf{X}^t - \mathbf{X}^{t-1})$; step ② uses the triangle inequality and $\alpha \in [0,1)$.

**Part (c).** We have: $\|\mathcal{A}(\mathbf{X}_{\mathsf{c}}^t) - \mathbf{y}^t\| \overset{①}{\leq} \|\mathcal{A}(\mathbf{X}^t) - \mathbf{y}^t\| + \|\mathcal{A}(\mathbf{X}^t) - \mathcal{A}(\mathbf{X}_{\mathsf{c}}^t)\| \leq \|\mathcal{A}(\mathbf{X}^t) - \mathbf{y}^t\| + \overline{\mathrm{A}}\|\mathbf{X}^t - \mathbf{X}_{\mathsf{c}}^t\| \overset{②}{\leq} \|\mathcal{A}(\mathbf{X}^t) - \mathbf{y}^t\| + \overline{\mathrm{A}}\|\mathbf{X}^t - \mathbf{X}^{t-1}\|$, where step ① uses the triangle inequality; step ② uses Claim (*a*) of this lemma.

$\square$

**Lemma A.7.** *Let* $\mathbf{P}, \tilde{\mathbf{P}} \in \mathbb{R}^{n \times r}$, *and* $\mathbf{X}, \tilde{\mathbf{X}} \in \mathcal{M}$. *We have:*

$$\| \operatorname{Proj}_{\mathbf{T_X}\mathcal{M}}(\mathbf{P}) - \operatorname{Proj}_{\mathbf{T_{\tilde{X}}}\mathcal{M}}(\tilde{\mathbf{P}})\|_\mathsf{F} \leq 2\|\mathbf{P} - \tilde{\mathbf{P}}\|_\mathsf{F} + 2\sqrt{r}\|\mathbf{P}\|\|\mathbf{X} - \tilde{\mathbf{X}}\|_\mathsf{F}.$$

*Proof.* First, we obtain:

$$
\begin{aligned}
&\|\mathbf{X}\mathbf{P}^\mathsf{T}\mathbf{X} - \tilde{\mathbf{X}}\tilde{\mathbf{P}}^\mathsf{T}\tilde{\mathbf{X}}\|_\mathsf{F} \\
=\ & \|(\mathbf{X} - \tilde{\mathbf{X}})\mathbf{P}^\mathsf{T}\mathbf{X} + \tilde{\mathbf{X}}\mathbf{P}^\mathsf{T}(\mathbf{X} - \tilde{\mathbf{X}}) + \tilde{\mathbf{X}}(\mathbf{P} - \tilde{\mathbf{P}})^\mathsf{T}\tilde{\mathbf{X}}\|_\mathsf{F} \\
\overset{\text{①}}{\leq}\ & \|\mathbf{X} - \tilde{\mathbf{X}}\|_\mathsf{F}\|\mathbf{P}^\mathsf{T}\mathbf{X}\| + \|\tilde{\mathbf{X}}\mathbf{P}^\mathsf{T}\|\|\mathbf{X} - \tilde{\mathbf{X}}\|_\mathsf{F} + \|\tilde{\mathbf{X}}(\mathbf{P} - \tilde{\mathbf{P}})^\mathsf{T}\tilde{\mathbf{X}}\|_\mathsf{F} \\
\overset{\text{②}}{\leq}\ & 2\sqrt{r}\|\mathbf{P}\|\|\mathbf{X} - \tilde{\mathbf{X}}\|_\mathsf{F} + \|\mathbf{P} - \tilde{\mathbf{P}}\|_\mathsf{F},
\end{aligned}
\tag{10}
$$

where step ① uses the triangle inequality; step ② uses $\|\mathbf{AB}\|_\mathsf{F} \leq \|\mathbf{A}\| \cdot \|\mathbf{B}\|_\mathsf{F}$, and $\|\tilde{\mathbf{X}}\| \leq 1$.

Second, we have:

$$
\begin{aligned}
&\|\mathbf{X}\mathbf{X}^\mathsf{T}\mathbf{P} - \tilde{\mathbf{X}}\tilde{\mathbf{X}}^\mathsf{T}\tilde{\mathbf{P}}\|_\mathsf{F} \\
=\ & \|(\mathbf{X} - \tilde{\mathbf{X}})\mathbf{X}^\mathsf{T}\mathbf{P} + \tilde{\mathbf{X}}(\mathbf{X} - \tilde{\mathbf{X}})^\mathsf{T}\mathbf{P} + \tilde{\mathbf{X}}\tilde{\mathbf{X}}^\mathsf{T}(\mathbf{P} - \tilde{\mathbf{P}})\|_\mathsf{F} \\
\overset{\text{①}}{\leq}\ & \|\mathbf{X} - \tilde{\mathbf{X}}\|_\mathsf{F}\|\mathbf{X}^\mathsf{T}\mathbf{P}\| + \|\tilde{\mathbf{X}}\| \cdot \|\mathbf{X} - \tilde{\mathbf{X}}\|_\mathsf{F} \cdot \|\mathbf{P}\| + \|\tilde{\mathbf{X}}\tilde{\mathbf{X}}^\mathsf{T}\| \cdot \|\mathbf{P} - \tilde{\mathbf{P}}\|_\mathsf{F} \\
\overset{\text{②}}{\leq}\ & 2\sqrt{r}\|\mathbf{P}\|\|\mathbf{X} - \tilde{\mathbf{X}}\|_\mathsf{F} + \|\mathbf{P} - \tilde{\mathbf{P}}\|_\mathsf{F},
\end{aligned}
\tag{11}
$$

where step ① uses the triangle inequality; step ② uses $\|\mathbf{AB}\|_\mathsf{F} \leq \|\mathbf{A}\| \cdot \|\mathbf{B}\|_\mathsf{F}$, and $\|\tilde{\mathbf{X}}\| \leq 1$.

Finally, we derive:

$$
\begin{aligned}
&\| \operatorname{Proj}_{\mathbf{T_X}\mathcal{M}}(\mathbf{P}) - \operatorname{Proj}_{\mathbf{T_{\tilde{X}}}\mathcal{M}}(\tilde{\mathbf{P}})\|_\mathsf{F} \\
\overset{\text{①}}{=}\ & \|[\mathbf{P} - \tfrac{1}{2}\mathbf{X}\mathbf{P}^\mathsf{T}\mathbf{X} - \tfrac{1}{2}\mathbf{X}\mathbf{X}^\mathsf{T}\mathbf{P}] - [\tilde{\mathbf{P}} - \tfrac{1}{2}\tilde{\mathbf{X}}\tilde{\mathbf{P}}^\mathsf{T}\tilde{\mathbf{X}} - \tfrac{1}{2}\tilde{\mathbf{X}}\tilde{\mathbf{X}}^\mathsf{T}\tilde{\mathbf{P}}]\|_\mathsf{F} \\
\overset{\text{②}}{\leq}\ & \|\mathbf{P} - \tilde{\mathbf{P}}\|_\mathsf{F} + \tfrac{1}{2}\|\mathbf{X}\mathbf{P}^\mathsf{T}\mathbf{X} - \tilde{\mathbf{X}}\tilde{\mathbf{P}}^\mathsf{T}\tilde{\mathbf{X}}\|_\mathsf{F} + \tfrac{1}{2}\|\mathbf{X}\mathbf{X}^\mathsf{T}\mathbf{P} - \tilde{\mathbf{X}}\tilde{\mathbf{X}}^\mathsf{T}\tilde{\mathbf{P}}\|_\mathsf{F} \\
\overset{\text{③}}{\leq}\ & \|\mathbf{P} - \tilde{\mathbf{P}}\|_\mathsf{F} + 2\sqrt{r}\|\mathbf{P}\|\|\mathbf{X} - \tilde{\mathbf{X}}\|_\mathsf{F} + \|\mathbf{P} - \tilde{\mathbf{P}}\|_\mathsf{F}
\end{aligned}
$$

where step ① uses $\operatorname{Proj}_{\mathbf{T_X}\mathcal{M}}(\mathbf{\Delta}) = \mathbf{\Delta} - \tfrac{1}{2}\mathbf{X}(\mathbf{\Delta}^\mathsf{T}\mathbf{X} + \mathbf{X}^\mathsf{T}\mathbf{\Delta})$ for all $\mathbf{\Delta} \in \mathbb{R}^{n \times r}$ (Absil et al., 2008a); step ② uses the triangle inequality; step ③ uses Inequalities (10) and (11).

$\square$

**Lemma A.8.** *We let* $p \in (0, 1)$. *We define* $g(t) \triangleq \frac{1}{1-p}(t+1)^{(1-p)} - \frac{1}{1-p} - (1-p)t^{(1-p)}$. *We have* $g(t) \geq 0$ *for all* $t \geq 1$.

*Proof.* We assume $p \in (0, 1)$.

First, we show that $h(p) \triangleq (1-p)^{1/p} \leq \frac{1}{\exp(1)}$. Recall that it holds: $\lim_{p \to 0^+}(1+p)^{1/p} = \exp(1)$ and $\lim_{p \to 0^+}(1-p)^{1/p} = 1/\exp(1)$. Given the function $h(p)$ is a decreasing function on $p \in (0, 1)$, we have $h(p) \leq \lim_{p \to 0^+}(1-p)^{1/p} = \frac{1}{\exp(1)}$.

Second, we show that $f(q) = 2^q - 1 - q^2 \geq 0$ for all $q \in (0, 1)$. We have $\nabla f(q) = \log(2)2^q - 2q$, and $\nabla^2 f(q) = 2^q(\log(2))^2 - 2 \leq 2(\log(2))^2 - 2 \leq 0$, implying that the function $f(q)$ is concave on $q \in (0, 1)$. Noticing $f(0) = f(1) = 0$, we conclude that $f(q) \geq 0$.

Third, we show that $g(t)$ is an increasing function. We have: $\nabla g(t) = (t+1)^{-p} - (1-p)^2 t^{-p} = (t+1)^{-p} \cdot (1 - (1-p)^2(\frac{t+1}{t})^p) \overset{\text{①}}{\geq} (t+1)^{-p} \cdot (1 - (1-p)^2 2^p) \overset{\text{②}}{\geq} (t+1)^{-p} \cdot (1 - (\frac{2}{\exp(1)^2})^p) \overset{\text{③}}{\geq} 0$, where step ① uses $\frac{t+1}{t} \leq 2$ for all $t \geq 1$; step ② uses $1 - p \leq (\frac{1}{\exp(1)})^p$ for all $p \in (0, 1)$; step ③ uses $\frac{2}{\exp(1)^2} \approx 0.2707 < 1$.

Finally, we have: $\forall t \geq 1$, $g(t) \overset{\text{①}}{\geq} g(1) = (1-p)^{-1} \cdot \{2^{(1-p)} - 1 - (1-p)^2\} \overset{\text{②}}{\geq} 0$, where step ① uses the fact that $g(t)$ is an increasing function; step ② uses $2^q - 1 - q^2 \geq 0$ for all $q = 1 - p \in (0, 1)$.

$\square$

**Lemma A.9.** *Assume $p \in (0, 1)$. We have:* $(1-p)T^{(1-p)} \leq \sum_{t=1}^{T} \frac{1}{t^p} \leq \frac{T^{(1-p)}}{1-p}$.

*Proof.* We define $g(t) \triangleq \frac{1}{t^p}$ and $h(t) \triangleq \frac{1}{1-p} t^{(1-p)}$.

Using the integral test for convergence, we obtain: $\int_1^{T+1} g(x)dx \leq \sum_{t=1}^{T} g(t) \leq g(1) + \int_1^T g(x)dx$.

**Part (a).** We first consider the lower bound. We obtain: $\sum_{t=1}^{T} t^{-p} \geq \sum_{t=1}^{T} \int_t^{t+1} x^{-p}dx = \int_1^{T+1} x^{-p}dx \overset{\text{①}}{\geq} h(T+1) - h(1) = \frac{1}{1-p}(T+1)^{1-p} - \frac{1}{1-p} \overset{\text{②}}{\geq} (1-p)T^{1-p}$, where step ① uses $\nabla h(x) = x^{-p}$; step ② uses Lemma A.8.

**Part (b).** We now consider the upper bound. We have: $\sum_{t=1}^{T} t^{-p} \leq h(1) + \int_1^T x^{-p}dx \overset{\text{①}}{=} 1 + h(T) - h(1) = 1 + \frac{1}{1-p}(T)^{1-p} - \frac{1}{1-p} = \frac{T^{(1-p)}-p}{1-p} < \frac{T^{(1-p)}}{1-p}$, where step ① uses $\nabla h(x) = x^{-p}$.

$\square$

**Lemma A.10.** *Assume $(e^{t+1})^2 \leq (e^t + e^{t-1})(p^t - p^{t+1})$ and $p^t \geq p^{t+1}$, where $\{e^t, p^t\}_{t=0}^{\infty}$ are two nonnegative sequences. For all $i \geq 1$, we have:* $\sum_{t=i}^{\infty} e^{t+1} \leq e^i + e^{i-1} + 4p^i$.

*Proof.* We define $w_t \triangleq p^t - p^{t+1}$. We let $1 \leq i < T$.

First, for any $i \geq 1$, we have:

$$\sum_{t=i}^{T} w_t = \sum_{t=i}^{T}(p^t - p^{t+1}) = p^i - p^{T+1} \overset{\text{①}}{\leq} p^i, \tag{12}$$

where step ① uses $p^i \geq 0$ for all $i$.

Second, we obtain:

$$\begin{aligned}
e^{t+1} &\overset{\text{①}}{\leq} \sqrt{(e^t + e^{t-1})w_t} \\
&\overset{\text{②}}{\leq} \sqrt{\frac{\alpha}{2}(e^t + e^{t-1})^2 + (w_t)^2/(2\alpha)}, \forall \alpha > 0 \\
&\overset{\text{③}}{\leq} \sqrt{\frac{\alpha}{2}} \cdot (e^t + e^{t-1}) + w_t\sqrt{1/(2\alpha)}, \forall \alpha > 0. 
\end{aligned} \tag{13}$$

Here, step ① uses $(e^{t+1})^2 \leq (e^t + e^{t-1})(p^t - p^{t+1})$ and $w_t \triangleq p^t - p^{t+1}$; step ② uses the fact that $ab \leq \frac{\alpha}{2}a^2 + \frac{1}{2\alpha}b^2$ for all $\alpha > 0$; step ③ uses the fact that $\sqrt{a+b} \leq \sqrt{a} + \sqrt{b}$ for all $a, b \geq 0$.

Assume the parameter $\alpha$ is sufficiently small that $1 - 2\sqrt{\frac{\alpha}{2}} > 0$. Telescoping Inequality (13) over $t$ from $i$ to $T$, we obtain:

$$\begin{aligned}
&\sum_{t=i}^{T} w_t \sqrt{1/(2\alpha)} \\
&\geq \{\sum_{t=i}^{T} e^{t+1}\} - \sqrt{\frac{\alpha}{2}}\{\sum_{t=i}^{T} e^t\} - \sqrt{\frac{\alpha}{2}}\{\sum_{t=i}^{T} e^{t-1}\} \\
&= \{e^T + e^{T+1} + \sum_{t=i}^{T-2} e^{t+1}\} - \sqrt{\frac{\alpha}{2}}\{e^i + e^T + \sum_{t=i}^{T-2} e^{t+1}\} \\
&\quad - \sqrt{\frac{\alpha}{2}}\{e^{i-1} + e^i + \sum_{t=i}^{T-2} e^{t+1}\} \\
&= e^T + e^{T+1} - \sqrt{\frac{\alpha}{2}}(e^i + e^T + e^{i-1} + e^i) + (1 - 2\sqrt{\frac{\alpha}{2}})\sum_{t=i}^{T-2} e^{t+1} \\
&\overset{\text{①}}{\geq} e^T(1 - \sqrt{\frac{\alpha}{2}}) - \sqrt{\frac{\alpha}{2}}(e^i + e^{i-1} + e^i) + (1 - 2\sqrt{\frac{\alpha}{2}})\sum_{t=i}^{T-2} e^{t+1} \\
&\overset{\text{②}}{\geq} -2\sqrt{\frac{\alpha}{2}}(e^i + e^{i-1}) + (1 - 2\sqrt{\frac{\alpha}{2}})\sum_{t=i}^{T-2} e^{t+1},
\end{aligned}$$

where step ① uses $e^{T+1} \geq 0$; step ② uses $1 - \sqrt{\frac{\alpha}{2}} > 1 - 2\sqrt{\frac{\alpha}{2}} > 0$. This leads to:

$$\begin{aligned}
\sum_{t=i}^{T-2} e^{t+1} &\leq (1 - 2\sqrt{\frac{\alpha}{2}})^{-1} \cdot \{2\sqrt{\frac{\alpha}{2}}(e^i + e^{i-1}) + \sqrt{\frac{1}{2\alpha}}\sum_{t=i}^{T} w_t\} \\
&\overset{\text{①}}{=} (e^i + e^{i-1}) + 4\sum_{t=i}^{T} w_t \\
&\overset{\text{②}}{=} (e^i + e^{i-1}) + 4p^i,
\end{aligned}$$

step ① uses the fact that $(1 - 2\sqrt{\frac{\alpha}{2}})^{-1} \cdot 2\sqrt{\frac{\alpha}{2}} = 1$ and $(1 - 2\sqrt{\frac{\alpha}{2}})^{-1} \cdot \sqrt{\frac{1}{2\alpha}} = 4$ when $\alpha = 1/8$; step ② uses Inequalities (12). Letting $T \to \infty$, we conclude this lemma.

$\square$

**Lemma A.11.** *Assume* $\sum_{t=1}^{T}(1/\tilde{\beta}^t) \geq \mathcal{O}(T^a)$, *where* $a \geq 0$ *is a constant, and* $\{\tilde{\beta}^t\}_{t=1}^{T}$ *is a nonnegative increasing sequence. If $T$ is an even number, we have:* $\sum_{t=1}^{T/2}(1/\tilde{\beta}^{2t}) \geq \mathcal{O}(T^a)$.

*Proof.* We have: $\sum_{t=1}^{T/2} \frac{1}{\tilde{\beta}^{2t}} = \frac{1}{2}\sum_{t=1}^{T/2}(\frac{1}{\tilde{\beta}^{2t}} + \frac{1}{\tilde{\beta}^{2t}}) \overset{①}{\geq} \frac{1}{2}\sum_{t=1}^{T/2}(\frac{1}{\tilde{\beta}^{2t}} + \frac{1}{\tilde{\beta}^{2t+1}}) = \frac{1}{\tilde{\beta}^{2T+1}} - \frac{1}{\tilde{\beta}^1} + \sum_{t=1}^{T}\frac{1}{\tilde{\beta}^t} = \mathcal{O}(\sum_{t=1}^{T}\frac{1}{\tilde{\beta}^t}) \geq \mathcal{O}(T^a)$, where step ① uses the fact that $\{\tilde{\beta}^t\}_{t=1}^{T}$ is increasing.

$\square$

**Lemma A.12.** *Assume that* $\frac{d^t}{d^{t-2}} \leq \frac{\dot{\beta}^t+1}{\dot{\beta}^t+2}$, *and* $\sum_{i=0}^{T}(1/\dot{\beta}^i) \geq \mathcal{O}(T^a)$, *where* $a \geq 0$ *is a positive constant,* $\{d^t\}_{t=0}^{\infty}$ *and* $\{\dot{\beta}^t\}_{t=0}^{\infty}$ *are two nonnegative sequences. Assume that* $\{\dot{\beta}^t\}_{t=0}^{\infty}$ *is increasing. We have:* $d^T \leq \mathcal{O}(1/\exp(T^a))$.

*Proof.* We define $\gamma^t \triangleq \frac{1}{\dot{\beta}^t+2} \in (0,1)$.

Given $\frac{d^t}{d^{t-2}} \leq \frac{\dot{\beta}^t+1}{\dot{\beta}^t+2}$, we have $\frac{d^t}{d^{t-2}} \leq 1 - \gamma^t$, leading to:

$$d^{2t} \quad \leq \quad d^0(1 - \gamma^2)(1 - \gamma^4)(1 - \gamma^6)\ldots(1 - \gamma^{2t}). \tag{14}$$

**Part (a)**. When $T$ is an even number, we have:

$$
\begin{aligned}
d^T &= \exp(\log(d^T)) \\
&\overset{①}{\leq} \exp(\log(d^0 \cdot \prod_{t=1}^{T/2}(1 - \gamma^{2t}))) \\
&\overset{②}{=} \exp(\log(d^0) + \sum_{t=1}^{T/2}\log(1 - \gamma^{2t})) \\
&\overset{③}{\leq} \exp(\log(d^0) + \sum_{t=1}^{T/2}(-\gamma^{2t})) \\
&\overset{④}{\leq} \exp(\log(d^0)) \times \{\exp(\sum_{t=1}^{T/2}(\gamma^{2t}))\}^{-1} \\
&\overset{⑤}{\leq} d^0 \times \{\exp(\mathcal{O}(T^a))\}^{-1} = \mathcal{O}(1/\exp(T^a)),
\end{aligned}
$$

where step ① uses Inequality (14); step ② uses $\log(ab) = \log(a) + \log(b)$ for all $a > 0$ and $b > 0$; step ③ uses $\log(1 - x) \leq -x$ for all $x \in (0,1)$, and $1 - \gamma^t \in (0,1)$ for all $t$; step ④ uses $\exp(a + b) = \exp(a)\exp(b)$ for all $a > 0$ and $b > 0$; step ⑤ uses Lemma A.11 with $\tilde{\beta}^t = 1/\gamma^t = \dot{\beta}^t + 2$.

**Part (b)**. When $T$ is an odd number, analogous strategies result in the same complexity outcome.

$\square$

**Lemma A.13.** *Assume that* $[d^t]^{\tau+1} \leq \ddot{\beta}^t(d^{t-2} - d^t)$, *and* $\sum_{i=1}^{T}(1/\ddot{\beta}^i) \geq \mathcal{O}(T^a)$, *where* $\tau, a > 0$ *are positive constants,* $\{d^t\}_{t=0}^{\infty}$ *and* $\{\ddot{\beta}^t\}_{t=0}^{\infty}$ *are two nonnegative sequences. Assume that* $\{\dot{\beta}^t\}_{t=0}^{\infty}$ *is increasing. We have:* $d^T \leq \mathcal{O}(1/(T^{a/\tau}))$.

*Proof.* We let $\kappa > 1$ be any constant. We define $h(s) = s^{-\tau-1}$, where $\tau > 0$.

We consider two cases for $h(d^t)/h(d^{t-2})$.

**Case (1)**. $h(d^t) \leq \kappa h(d^{t-2})$. We define $\breve{h}(s) \triangleq -\frac{1}{\tau} \cdot s^{-\tau}$. We derive:

$$
\begin{aligned}
1 \quad &\overset{①}{\leq} \quad \ddot{\beta}^t(d^{t-2} - d^t) \cdot h(d^t) \\
&\overset{②}{\leq} \quad \ddot{\beta}^t(d^{t-2} - d^t) \cdot \kappa h(d^{t-2}) \\
&\overset{③}{\leq} \quad \ddot{\beta}^t \kappa \int_{d^t}^{d^{t-2}} h(s)ds \\
&\overset{④}{=} \quad \ddot{\beta}^t \kappa \cdot (\breve{h}(d^{t-2}) - \breve{h}(d^t)) \\
&\overset{⑤}{=} \quad \ddot{\beta}^t \kappa \cdot \frac{1}{\tau} \cdot ([d^t]^{-\tau} - [d^{t-2}]^{-\tau}),
\end{aligned}
$$

where step ① uses $[d^t]^{\tau+1} \leq \ddot{\beta}^t(d^{t-2} - d^t)$; step ② uses $h(d^t) \leq \kappa h(d^{t-2})$; step ③ uses the fact that $h(s)$ is a nonnegative and increasing function that $(a - b)h(a) \leq \int_b^a h(s)ds$ for all $a, b \in [0, \infty)$; step ④ uses the fact that $\nabla \breve{h}(s) = h(s)$; step ⑤ uses the definition of $\breve{h}(\cdot)$. This leads to:

$$
[d^t]^{-\tau} - [d^{t-2}]^{-\tau} \geq \frac{\kappa^{-1}\tau}{\ddot{\beta}^t}. \tag{15}
$$

**Case (2)**. $h(d^t) > \kappa h(d^{t-2})$. We have:

$$
\begin{aligned}
h(d^t) > \kappa h(d^{t-2}) \quad &\overset{①}{\Rightarrow} \quad [d^t]^{-(\tau+1)} > \kappa \cdot [d^{t-2}]^{-(\tau+1)} \\
&\overset{②}{\Rightarrow} \quad ([d^t]^{-(\tau+1)})^{\frac{\tau}{\tau+1}} > \kappa^{\frac{\tau}{\tau+1}} \cdot ([d^{t-2}]^{-(\tau+1)})^{\frac{\tau}{\tau+1}} \\
&\Rightarrow \quad [d^t]^{-\tau} > \kappa^{\frac{\tau}{\tau+1}} \cdot [d^{t-2}]^{-\tau},
\end{aligned} \tag{16}
$$

where step ① uses the definition of $h(\cdot)$; step ② uses the fact that if $a > b > 0$, then $a^{\dot{\tau}} > b^{\dot{\tau}}$ for any exponent $\dot{\tau} \triangleq \frac{\tau}{\tau+1} \in (0, 1)$. We further derive:

$$
\begin{aligned}
[d^t]^{-\tau} - [d^{t-2}]^{-\tau} \quad &\overset{①}{\geq} \quad (\kappa^{\frac{\tau}{\tau+1}} - 1) \cdot [d^{t-2}]^{-\tau} \\
&\overset{②}{\geq} \quad (\kappa^{\frac{\tau}{\tau+1}} - 1) \cdot [d^0]^{-\tau},
\end{aligned} \tag{17}
$$

where step ① uses Inequality (16); step ② uses $\tau > 0$ and $d^{t-2} \leq d^0$ for all $t$.

In view of Inequalities (15) and (17), we have:

$$
\begin{aligned}
[d^t]^{-\tau} - [d^{t-2}]^{-\tau} \quad &\geq \quad \min(\frac{\kappa^{-1}\tau}{\beta^t}, (\kappa^{\frac{\tau}{\tau+1}} - 1) \cdot [d^0]^{-\tau}) \\
&\geq \quad \frac{1}{\ddot{\beta}^t} \cdot \underbrace{\min(\kappa^{-1}\tau, (\kappa^{\frac{\tau}{\tau+1}} - 1) \cdot [d^0]^{-\tau} \ddot{\beta}^0)}_{\mu_0}.
\end{aligned} \tag{18}
$$

We now focus on Inequality (18).

**Part (a)**. When $T$ is an even number, telescoping Inequality (18) over $t = \{2, 4, \ldots, T\}$, we have:

$$
[d^T]^{-\tau} - [d^0]^{-\tau} \geq \mu_0 \sum_{t=1}^{T/2} \frac{1}{\ddot{\beta}^{2t}} \overset{①}{\geq} \mathcal{O}(T^a),
$$

where step ① use Lemma A.11. This leads to:

$$
d^T = ([d^T]^{-\tau})^{-1/\tau} \leq \mathcal{O}(T^a)^{-1/\tau} = \mathcal{O}(1/(T^{a/\tau})).
$$

**Part (b)**. When $T$ is an odd number, analogous strategies result in the same complexity outcome.

$\square$

# B   PROOFS FOR SECTION 2

## B.1   PROOF OF LEMMA 2.3

*Proof.* Assume $0 < \mu_2 < \mu_1 < \frac{1}{W_h}$, and fixing $\mathbf{y} \in \mathbb{R}^m$.

We define $h_{\mu_1}(\mathbf{y}) \triangleq \min_{\mathbf{v}} h(\mathbf{v}) + \frac{1}{2\mu_1}\|\mathbf{v} - \mathbf{y}\|_2^2$, and $\mathbb{P}_{\mu_1}(\mathbf{y}) = \arg\min_{\mathbf{v}} h(\mathbf{v}) + \frac{1}{2\mu_1}\|\mathbf{v} - \mathbf{y}\|_2^2$.

We define $h_{\mu_2}(\mathbf{y}) \triangleq \min_{\mathbf{v}} h(\mathbf{v}) + \frac{1}{2\mu_2}\|\mathbf{v} - \mathbf{y}\|_2^2$, and $\mathbb{P}_{\mu_2}(\mathbf{y}) = \arg\min_{\mathbf{v}} h(\mathbf{v}) + \frac{1}{2\mu_2}\|\mathbf{v} - \mathbf{y}\|_2^2$.

By the optimality of $\mathbb{P}_{\mu_1}(\mathbf{y})$ and $\mathbb{P}_{\mu_2}(\mathbf{y})$, we obtain:

$$
\begin{align}
\mathbf{y} - \mathbb{P}_{\mu_1}(\mathbf{y}) &\in \mu_1 \partial h(\mathbb{P}_{\mu_1}(\mathbf{y})) \tag{19}\\
\mathbf{y} - \mathbb{P}_{\mu_2}(\mathbf{y}) &\in \mu_2 \partial h(\mathbb{P}_{\mu_2}(\mathbf{y})). \tag{20}
\end{align}
$$

**Part (a)**. We now prove that $0 \leq h_{\mu_2}(\mathbf{y}) - h_{\mu_1}(\mathbf{y})$. For any $\mathbf{s}_1 \in \partial h(\mathbb{P}_{\mu_1}(\mathbf{y}))$ and $\mathbf{s}_2 \in \partial h(\mathbb{P}_{\mu_2}(\mathbf{y}))$, we have:

$$
\begin{align*}
&h_{\mu_1}(\mathbf{y}) - h_{\mu_2}(\mathbf{y})\\
\overset{①}{=}\ & \tfrac{1}{2\mu_1}\|\mathbf{y} - \mathbb{P}_{\mu_1}(\mathbf{y})\|_2^2 - \tfrac{1}{2\mu_2}\|\mathbf{y} - \mathbb{P}_{\mu_2}(\mathbf{y})\|_2^2 + h(\mathbb{P}_{\mu_1}(\mathbf{y})) - h(\mathbb{P}_{\mu_2}(\mathbf{y}))\\
\overset{②}{\leq}\ & \tfrac{1}{2\mu_1}\|\mathbf{y} - \mathbb{P}_{\mu_1}(\mathbf{y})\|_2^2 - \tfrac{1}{2\mu_2}\|\mathbf{y} - \mathbb{P}_{\mu_2}(\mathbf{y})\|_2^2 + \langle \mathbb{P}_{\mu_1}(\mathbf{y}) - \mathbb{P}_{\mu_2}(\mathbf{y}), \mathbf{s}_1 \rangle + \tfrac{W_h}{2}\|\mathbb{P}_{\mu_2}(\mathbf{y}) - \mathbb{P}_{\mu_1}(\mathbf{y})\|_2^2\\
\overset{③}{=}\ & \tfrac{1}{2\mu_1}\|\mu_1\mathbf{s}_1\|_2^2 - \tfrac{1}{2\mu_2}\|\mu_2\mathbf{s}_2\|_2^2 + \langle \mu_2\mathbf{s}_2 - \mu_1\mathbf{s}_1, \mathbf{s}_1 \rangle + \tfrac{W_h}{2}\|\mu_1\mathbf{s}_1 - \mu_2\mathbf{s}_2\|_2^2\\
\overset{④}{\leq}\ & \tfrac{1}{2\mu_1}\|\mu_1\mathbf{s}_1\|_2^2 - \tfrac{1}{2\mu_2}\|\mu_2\mathbf{s}_2\|_2^2 + \langle \mu_2\mathbf{s}_2 - \mu_1\mathbf{s}_1, \mathbf{s}_1 \rangle + \tfrac{1}{2\mu_1}\|\mu_1\mathbf{s}_1 - \mu_2\mathbf{s}_2\|_2^2\\
=\ & -\tfrac{\mu_2}{2}\|\mathbf{s}_2\|_2^2 \cdot (1 - \tfrac{\mu_2}{\mu_1})\\
\overset{⑤}{\leq}\ & 0,
\end{align*}
$$

where step ① uses the definition of $h_{\mu_1}(\mathbf{y})$ and $h_{\mu_2}(\mathbf{y})$; step ② uses weakly convexity of $h(\cdot)$; step ③ uses the optimality of $\mathbb{P}_{\mu_1}(\mathbf{y})$ and $\mathbb{P}_{\mu_2}(\mathbf{y})$ in Equations (19) and (20); step ④ uses $W_h \leq \frac{1}{\mu_1}$; step ⑤ uses $1 \geq \frac{\mu_2}{\mu_1}$.

**Part (b)**. We now prove that $h_{\mu_2}(\mathbf{y}) - h_{\mu_1}(\mathbf{y}) \leq \min\{\frac{\mu_1}{2\mu_2}, 1\} \cdot (\mu_1 - \mu_2)C_h^2$. For any $\mathbf{s}_1 \in \partial h(\mathbb{P}_{\mu_1}(\mathbf{y}))$ and $\mathbf{s}_2 \in \partial h(\mathbb{P}_{\mu_2}(\mathbf{y}))$, we have:

$$
\begin{align*}
&h_{\mu_2}(\mathbf{y}) - h_{\mu_1}(\mathbf{y})\\
\overset{①}{=}\ & \tfrac{1}{2\mu_2}\|\mathbf{y} - \mathbb{P}_{\mu_2}(\mathbf{y})\|_2^2 - \tfrac{1}{2\mu_1}\|\mathbf{y} - \mathbb{P}_{\mu_1}(\mathbf{y})\|_2^2 + h(\mathbb{P}_{\mu_2}(\mathbf{y})) - h(\mathbb{P}_{\mu_1}(\mathbf{y}))\\
\overset{②}{\leq}\ & \tfrac{1}{2\mu_2}\|\mathbf{y} - \mathbb{P}_{\mu_2}(\mathbf{y})\|_2^2 - \tfrac{1}{2\mu_1}\|\mathbf{y} - \mathbb{P}_{\mu_1}(\mathbf{y})\|_2^2 + \langle \mathbb{P}_{\mu_2}(\mathbf{y}) - \mathbb{P}_{\mu_1}(\mathbf{y}), \mathbf{s}_1 \rangle + \tfrac{W_h}{2}\|\mathbb{P}_{\mu_2}(\mathbf{y}) - \mathbb{P}_{\mu_1}(\mathbf{y})\|_2^2\\
\overset{③}{=}\ & \tfrac{\mu_2}{2}\|\mathbf{s}_1\|_2^2 - \tfrac{\mu_1}{2}\|\mathbf{s}_2\|_2^2 + \langle \mu_1\mathbf{s}_2 - \mu_2\mathbf{s}_1, \mathbf{s}_1 \rangle + \tfrac{W_h}{2}\|\mu_1\mathbf{s}_2 - \mu_2\mathbf{s}_1\|_2^2\\
=\ & -\tfrac{\mu_2}{2}\|\mathbf{s}_1\|_2^2 - \tfrac{\mu_1}{2}\|\mathbf{s}_2\|_2^2 + \mu_1\langle \mathbf{s}_1, \mathbf{s}_2 \rangle + \tfrac{W_h}{2}\|\mu_1\mathbf{s}_2 - \mu_2\mathbf{s}_1\|_2^2\\
\overset{④}{\leq}\ & \min\{-\tfrac{\mu_1}{2}\|\mathbf{s}_2\|_2^2 + \mu_1\langle \mathbf{s}_1, \mathbf{s}_2 \rangle + \tfrac{1}{2\mu_2}\|\mu_1\mathbf{s}_2 - \mu_2\mathbf{s}_1\|_2^2 - \tfrac{\mu_2}{2}\|\mathbf{s}_1\|_2^2,\\
& \qquad -\tfrac{\mu_1}{2}\|\mathbf{s}_2\|_2^2 + \mu_1\langle \mathbf{s}_1, \mathbf{s}_2 \rangle + \tfrac{1}{2\mu_1}\|\mu_1\mathbf{s}_2 - \mu_2\mathbf{s}_1\|_2^2 - \tfrac{\mu_2}{2}\|\mathbf{s}_1\|_2^2\}\\
=\ & \min\{(-\mu_2 + \mu_1) \cdot \tfrac{\mu_1}{2\mu_2}\|\mathbf{s}_2\|_2^2, (\mu_1 - \mu_2)\langle \mathbf{s}_1, \mathbf{s}_2 \rangle - \tfrac{\mu_2}{2}\|\mathbf{s}_1\|_2^2 + \tfrac{\mu_2^2}{2\mu_1}\|\mathbf{s}_1\|_2^2\}\\
\overset{⑤}{\leq}\ & \min\{\tfrac{\mu_1}{2\mu_2}\|\mathbf{s}_2\|_2^2 \cdot (\mu_1 - \mu_2), (\mu_1 - \mu_2)\langle \mathbf{s}_1, \mathbf{s}_2 \rangle\}\\
\overset{⑥}{\leq}\ & \min\{\tfrac{\mu_1}{2\mu_2} \cdot (\mu_1 - \mu_2), (\mu_1 - \mu_2)\} \cdot C_h^2\\
=\ & \min\{\tfrac{\mu_1}{2\mu_2}, 1\} \cdot (\mu_1 - \mu_2) \cdot C_h^2,
\end{align*}
$$

where step ① uses the definition of $h_{\mu_1}(\mathbf{y})$ and $h_{\mu_2}(\mathbf{y})$; step ② uses the weakly convexity of $h(\cdot)$; step ③ uses the optimality of $\mathbb{P}_{\mu_2}(\mathbf{y})$ and $\mathbb{P}_{\mu_1}(\mathbf{y})$ in Equations (19) and (20); step ④ uses $W_h \leq \frac{1}{\mu_1}$ and $W_h \leq \frac{1}{\mu_2}$; step ⑤ uses $\mu_2 \leq \mu_1$; step ⑥ uses $\|\mathbf{s}_1\| \leq C_h$, $\|\mathbf{s}_2\| \leq C_h$, and $\langle \mathbf{s}_1, \mathbf{s}_2 \rangle \leq \|\mathbf{s}_1\| \cdot \|\mathbf{s}_2\| \leq C_h^2$.

$\square$

## B.2 PROOF OF LEMMA 2.4

*Proof.* Assume $0 < \mu_2 < \mu_1 \leq \frac{1}{2W_h}$, and fixing $\mathbf{y} \in \mathbb{R}^m$.

Using the result in Lemma 2.2, we establish that the gradient of $h_\mu(\mathbf{y})$ *w.r.t* $\mathbf{y}$ can be computed as:

$$\nabla h_\mu(\mathbf{y}) = \mu^{-1}(\mathbf{y} - \mathbb{P}_\mu(\mathbf{y})).$$

The gradient of the mapping $\nabla h_\mu(\mathbf{y})$ *w.r.t.* the variable $1/\mu$ can be computed as: $\nabla_{1/\mu}(\nabla h_\mu(\mathbf{y})) = \mathbf{y} - \mathbb{P}_\mu(\mathbf{y})$. We further obtain:

$$\|\nabla_{1/\mu}(\nabla h_\mu(\mathbf{y}))\| = \|\mathbf{y} - \mathbb{P}_\mu(\mathbf{y})\| \overset{①}{=} \mu\|\partial h(\mathbb{P}_\mu(\mathbf{y}))\| \leq \mu C_h.$$

Here, step ① uses the optimality of $\mathbb{P}_\mu(\mathbf{y})$ that: $\mathbf{0} \in \partial h(\mathbb{P}_\mu(\mathbf{y})) + \frac{1}{\mu}(\mathbb{P}_\mu(\mathbf{y}) - \mathbf{y})$. Therefore, for all $\mu \in (0, \frac{1}{2W_h}]$, we have:

$$\frac{\|\nabla h_\mu(\mathbf{y}) - \nabla h_{\mu'}(\mathbf{y})\|_2}{|1/\mu - 1/\mu'|} \leq \mu C_h.$$

Letting $\mu = \mu_1$ and $\mu' = \mu_2$, we have: $\|\nabla h_{\mu_1}(\mathbf{y}) - \nabla h_{\mu_2}(\mathbf{y})\|_2 \leq |1 - \mu_1/\mu_2|C_h = (\mu_1/\mu_2 - 1)C_h$.

$\square$

## B.3 PROOF OF LEMMA 2.5

*Proof.* We consider the following optimization problem:

$$\bar{\mathbf{y}} = \arg\min_{\mathbf{y}} h_\mu(\mathbf{y}) + \frac{\beta}{2}\|\mathbf{y} - \mathbf{b}\|_2^2. \tag{21}$$

Given $h_\mu(\mathbf{y})$ being $(\mu^{-1})$-weakly convex and $\beta > \mu^{-1}$, Problem (21) becomes strongly convex and has a unique optimal solution, which leads to the following equivalent problem:

$$(\bar{\mathbf{y}}, \check{\mathbf{y}}) = \arg\min_{\mathbf{y}, \mathbf{y}'} h(\mathbf{y}') + \frac{1}{2\mu}\|\mathbf{y} - \mathbf{y}'\|_2^2 + \frac{\beta}{2}\|\mathbf{y} - \mathbf{b}\|_2^2,$$

We have the following first-order optimality conditions for $(\bar{\mathbf{y}}, \check{\mathbf{y}})$:

$$\frac{1}{\mu}(\bar{\mathbf{y}} - \check{\mathbf{y}}) = \beta(\mathbf{b} - \bar{\mathbf{y}}) \tag{22}$$

$$\frac{1}{\mu}(\bar{\mathbf{y}} - \check{\mathbf{y}}) \in \partial h(\check{\mathbf{y}}). \tag{23}$$

**Part (a)**. We have the following results:

$$\mathbf{0} \overset{①}{\in} \partial h(\check{\mathbf{y}}) + \frac{1}{\mu}(\check{\mathbf{y}} - \bar{\mathbf{y}})$$

$$\overset{②}{=} \partial h(\check{\mathbf{y}}) + \frac{1}{\mu}(\check{\mathbf{y}} - \frac{1}{1/\mu+\beta}(\frac{1}{\mu}\check{\mathbf{y}} + \beta\mathbf{b}))$$

$$= \partial h(\check{\mathbf{y}}) + \frac{\beta}{1+\mu\beta}(\check{\mathbf{y}} - \mathbf{b}), \tag{24}$$

where step ① uses Equality (23); step ② uses Equality (22) that $\bar{\mathbf{y}} = \frac{1}{1/\mu+\beta}(\frac{1}{\mu}\check{\mathbf{y}} + \beta\mathbf{b})$. The inclusion in (24) implies that:

$$\check{\mathbf{y}} = \arg\min_{\mathbf{v}} h(\check{\mathbf{y}}) + \frac{1}{2} \cdot \frac{\beta}{1+\mu\beta}\|\check{\mathbf{y}} - \mathbf{b}\|_2^2.$$

**Part (b)**. Combining Equalities (22) and (23), we have: $\beta(\mathbf{b} - \bar{\mathbf{y}}) \in \partial h(\check{\mathbf{y}})$.

**Part (c)**. In view of Equation (23), we have: $\bar{\mathbf{y}} - \check{\mathbf{y}} = \mu\partial h(\check{\mathbf{y}})$, leading to: $\|\check{\mathbf{y}} - \bar{\mathbf{y}}\| \leq \mu C_h$.

$\square$

### B.4 PROOFS FOR LEMMA 2.11

*Proof.* We let $\mathbf{\Delta} \in \mathbb{R}^{n \times r}$ and $\mathbf{X} \in \mathcal{M}$. We define $\mathbf{U} \triangleq \mathbf{\Delta}^{\mathsf{T}} \mathbf{X} \in \mathbb{R}^{r \times r}$.

We derive the following results:

$$
\begin{aligned}
&\| \operatorname{Proj}_{\mathbf{T_X}\mathcal{M}}(\mathbf{\Delta})\|_{\mathsf{F}}^2 - \|\mathbf{\Delta}\|_{\mathsf{F}}^2 \\
\overset{①}{=}\ & \|\mathbf{\Delta} - \tfrac{1}{2}\mathbf{X}(\mathbf{\Delta}^{\mathsf{T}}\mathbf{X} + \mathbf{X}^{\mathsf{T}}\mathbf{\Delta})\|_{\mathsf{F}}^2 - \|\mathbf{\Delta}\|_{\mathsf{F}}^2 \\
=\ & \tfrac{1}{4}\|\mathbf{X}(\mathbf{\Delta}^{\mathsf{T}}\mathbf{X} + \mathbf{X}^{\mathsf{T}}\mathbf{\Delta})\|_{\mathsf{F}}^2 - \langle \mathbf{\Delta}, \mathbf{X}(\mathbf{\Delta}^{\mathsf{T}}\mathbf{X} + \mathbf{X}^{\mathsf{T}}\mathbf{\Delta})\rangle \\
\overset{②}{=}\ & \tfrac{1}{4}\|\mathbf{\Delta}^{\mathsf{T}}\mathbf{X} + \mathbf{X}^{\mathsf{T}}\mathbf{\Delta}\|_{\mathsf{F}}^2 - \langle \mathbf{\Delta}, \mathbf{X}(\mathbf{\Delta}^{\mathsf{T}}\mathbf{X} + \mathbf{X}^{\mathsf{T}}\mathbf{\Delta})\rangle \\
\overset{③}{=}\ & \tfrac{1}{4}\|\mathbf{U} + \mathbf{U}^{\mathsf{T}}\|_{\mathsf{F}}^2 - \langle \mathbf{U} + \mathbf{U}^{\mathsf{T}}, \mathbf{U}\rangle \\
\overset{④}{=}\ & \tfrac{1}{4}\|\mathbf{U} + \mathbf{U}^{\mathsf{T}}\|_{\mathsf{F}}^2 - \langle \mathbf{U} + \mathbf{U}^{\mathsf{T}}, \mathbf{U} + \mathbf{U}^{\mathsf{T}}\rangle \cdot \tfrac{1}{2} \\
=\ & -\tfrac{1}{4}\|\mathbf{U} + \mathbf{U}^{\mathsf{T}}\|_{\mathsf{F}}^2 \leq 0,
\end{aligned}
$$

where step ① uses $\operatorname{Proj}_{\mathbf{T_X}\mathcal{M}}(\mathbf{\Delta}) = \mathbf{\Delta} - \tfrac{1}{2}\mathbf{X}(\mathbf{\Delta}^{\mathsf{T}}\mathbf{X} + \mathbf{X}^{\mathsf{T}}\mathbf{\Delta})$ for all $\mathbf{\Delta} \in \mathbb{R}^{n \times r}$ (Absil et al., 2008a); step ② uses the fact that $\|\mathbf{XP}\|_{\mathsf{F}}^2 = \operatorname{tr}(\mathbf{PX}^{\mathsf{T}}\mathbf{XP}^{\mathsf{T}}) = \|\mathbf{P}\|_{\mathsf{F}}^2$ for all $\mathbf{X} \in \mathcal{M}$; step ③ uses the definition of $\mathbf{U} \triangleq \mathbf{\Delta}^{\mathsf{T}}\mathbf{X}$; step ④ uses the symmetric properties of the matrix $(\mathbf{U} + \mathbf{U}^{\mathsf{T}})$.

$\square$

### B.5 PROOF OF LEMMA 2.12

*Proof.* We let $\rho > 0$, $\mathbf{G} \in \mathbb{R}^{n \times r}$, and $\mathbf{X} \in \mathcal{M}$.

We define $\mathbf{U} \triangleq \mathbf{G}^{\mathsf{T}}\mathbf{X}$, and $\mathbb{G}_\rho \triangleq \mathbf{G} - \rho\mathbf{XG}^{\mathsf{T}}\mathbf{X} - (1-\rho)\mathbf{XX}^{\mathsf{T}}\mathbf{G}$.

First, we have the following equalities:

$$
\begin{aligned}
\langle \mathbf{G}, \mathbb{G}_\rho\rangle &= \langle \mathbf{G}, \mathbf{G} - \rho\mathbf{XG}^{\mathsf{T}}\mathbf{X} - (1-\rho)\mathbf{XX}^{\mathsf{T}}\mathbf{G}\rangle \\
&= \langle \mathbf{G}, \mathbf{G}\rangle - \rho\operatorname{tr}(\mathbf{G}^{\mathsf{T}}\mathbf{XG}^{\mathsf{T}}\mathbf{X}) - (1-\rho)\operatorname{tr}(\mathbf{G}^{\mathsf{T}}\mathbf{XX}^{\mathsf{T}}\mathbf{G}) \\
&\overset{①}{=} \langle \mathbf{G}, \mathbf{G}\rangle - \rho\operatorname{tr}(\mathbf{UU}) - (1-\rho)\operatorname{tr}(\mathbf{UU}^{\mathsf{T}}),
\end{aligned}
\tag{25}
$$

where step ① uses $\mathbf{U} \triangleq \mathbf{G}^{\mathsf{T}}\mathbf{X}$.

Second, we derive the following equalities:

$$
\begin{aligned}
\|\mathbb{G}_\rho\|_{\mathsf{F}}^2 &= \langle \rho\mathbf{XG}^{\mathsf{T}}\mathbf{X} + (1-\rho)\mathbf{XX}^{\mathsf{T}}\mathbf{G} - \mathbf{G}, \rho\mathbf{XG}^{\mathsf{T}}\mathbf{X} + (1-\rho)\mathbf{XX}^{\mathsf{T}}\mathbf{G} - \mathbf{G}\rangle \\
&\overset{①}{=} \rho^2\operatorname{tr}(\mathbf{U}^{\mathsf{T}}\mathbf{U}) + \rho(1-\rho)\operatorname{tr}(\mathbf{U}^{\mathsf{T}}\mathbf{U}^{\mathsf{T}}) - \rho\operatorname{tr}(\mathbf{U}^{\mathsf{T}}\mathbf{U}^{\mathsf{T}}) \\
&\quad + (1-\rho)\rho\operatorname{tr}(\mathbf{UU}) + (1-\rho)^2\operatorname{tr}(\mathbf{UU}^{\mathsf{T}}) - (1-\rho)\operatorname{tr}(\mathbf{UU}^{\mathsf{T}}) \\
&\quad - \rho\operatorname{tr}(\mathbf{UU}) - (1-\rho)\operatorname{tr}(\mathbf{UU}^{\mathsf{T}}) + \langle \mathbf{G}, \mathbf{G}\rangle \\
&\overset{②}{=} (2\rho^2 - 1)\cdot\operatorname{tr}(\mathbf{U}^{\mathsf{T}}\mathbf{U}) - 2\rho^2\cdot\operatorname{tr}(\mathbf{UU}) + \langle \mathbf{G}, \mathbf{G}\rangle,
\end{aligned}
\tag{26}
$$

where step ① uses $\mathbf{U} \triangleq \mathbf{G}^{\mathsf{T}}\mathbf{X}$ and $\mathbf{X}^{\mathsf{T}}\mathbf{X} = \mathbf{I}_r$; step ② uses $\operatorname{tr}(\mathbf{U}^{\mathsf{T}}\mathbf{U}^{\mathsf{T}}) = \operatorname{tr}(\mathbf{UU})$.

Third, we have:

$$
\operatorname{tr}(\mathbf{G}^{\mathsf{T}}\mathbf{G}) - \operatorname{tr}(\mathbf{U}^{\mathsf{T}}\mathbf{U}) \overset{①}{=} \langle \mathbf{GG}^{\mathsf{T}}, \mathbf{I}_n - \mathbf{XX}^{\mathsf{T}}\rangle \overset{②}{\geq} 0,
\tag{27}
$$

where step ① uses $\mathbf{U} \triangleq \mathbf{G}^{\mathsf{T}}\mathbf{X}$; step ② uses the fact that the matrix $(\mathbf{I}_n - \mathbf{XX}^{\mathsf{T}})$ only contains eigenvalues that are 0 or 1.

**Part (a-i)**. We now prove that $\max(1, 2\rho)\langle \mathbf{G}, \mathbb{G}_\rho\rangle \geq \|\mathbb{G}_\rho\|_{\mathsf{F}}^2$. We discuss two cases. Case (*i*): $\rho \in (0, \tfrac{1}{2}]$. We have:

$$
\|\mathbb{G}_\rho\|_{\mathsf{F}}^2 - \langle \mathbf{G}, \mathbb{G}_\rho\rangle \overset{①}{=} (2\rho^2 - \rho)\cdot(\operatorname{tr}(\mathbf{UU}^{\mathsf{T}}) - \operatorname{tr}(\mathbf{UU})) \overset{②}{\leq} 0,
$$

where step ① uses Inequalities (25) and (26); step ② uses $2\rho^2 - \rho \leq 0$ for all $\rho \in (0, \frac{1}{2}]$, and $\text{tr}(\mathbf{U}\mathbf{U}) \leq \text{tr}(\mathbf{U}\mathbf{U}^{\mathsf{T}})$ for all $\mathbf{U} \in \mathbb{R}^{r \times r}$.

Case (**ii**): $\rho \in [\frac{1}{2}, \infty)$. We have:

$$\|\mathbb{G}_\rho\|_{\mathsf{F}}^2 - 2\rho\langle \mathbf{G}, \mathbb{G}_\rho \rangle \overset{①}{=} (2\rho - 1)(\text{tr}(\mathbf{U}\mathbf{U}^{\mathsf{T}}) - \langle \mathbf{G}, \mathbf{G} \rangle) \overset{②}{\leq} 0,$$

where step ① uses Inequalities (25) and (26); step ② uses $2\rho - 1 \geq 0$ for all $\rho \in [\frac{1}{2}, \infty)$, and Inequality(27). Therefore, we conclude that: $\max(1, 2\rho)\langle \mathbf{G}, \mathbb{G}_\rho \rangle \geq \|\mathbb{G}_\rho\|_{\mathsf{F}}^2$.

**Part (a-ii)**. We now prove that $\|\mathbb{G}_\rho\|_{\mathsf{F}}^2 \geq \min(1, \rho^2)\|\mathbb{G}_1\|_{\mathsf{F}}^2$. We consider two cases. Case (**i**): $\rho \in (0, 1]$. We have:

$$\rho^2\|\mathbb{G}_1\|_{\mathsf{F}}^2 - \|\mathbb{G}_\rho\|_{\mathsf{F}}^2 \overset{①}{=} (1 - \rho^2)(\text{tr}(\mathbf{U}^{\mathsf{T}}\mathbf{U}) - \langle \mathbf{G}, \mathbf{G} \rangle) \overset{②}{\leq} 0,$$

where step ① uses Inequalities (25) and (26); step ② uses $1 - \rho^2 \geq 0$, and Inequality (27).

Case (**ii**): $\rho \in (1, \infty)$. We have:

$$\|\mathbb{G}_1\|_{\mathsf{F}}^2 - \|\mathbb{G}_\rho\|_{\mathsf{F}}^2 \overset{①}{=} (2 - 2\rho^2)(\text{tr}(\mathbf{U}^{\mathsf{T}}\mathbf{U}) - \text{tr}(\mathbf{U}\mathbf{U})) \leq 0,$$

where step ① uses Inequality (26); step ② uses $4\rho^2 - 1 \leq 0$ for all $\rho \in (0, \frac{1}{2}]$, and the fact that $\text{tr}(\mathbf{U}\mathbf{U}) - \text{tr}(\mathbf{U}\mathbf{U}^{\mathsf{T}}) \leq 0$ for all $\mathbf{U} \in \mathbb{R}^{r \times r}$. Therefore, we conclude that: $\min(1, \rho^2)\|\mathbb{G}_1\|_{\mathsf{F}}^2 \leq \|\mathbb{G}_\rho\|_{\mathsf{F}}^2$.

**Part (b-i)**. We now prove that $\|\mathbb{G}_\rho\|_{\mathsf{F}} \geq \min(1, 2\rho)\|\mathbb{G}_{1/2}\|_{\mathsf{F}}$. We consider two cases. Case (**i**): $\rho \in (0, \frac{1}{2}]$. We have:

$$(2\rho)^2\|\mathbb{G}_{1/2}\|_{\mathsf{F}}^2 - \|\mathbb{G}_\rho\|_{\mathsf{F}}^2 \overset{①}{=} (4\rho^2 - 1) \cdot (\text{tr}(\mathbf{G}^{\mathsf{T}}\mathbf{G}) - \text{tr}(\mathbf{U}^{\mathsf{T}}\mathbf{U})) \overset{②}{\leq} 0,$$

where step ① uses Inequality (26); step ② uses $4\rho^2 - 1 \leq 0$ for all $\rho \in (0, \frac{1}{2}]$, and Inequality (27).

Case (**ii**): $\rho \in (\frac{1}{2}, \infty)$. We have:

$$\|\mathbb{G}_{1/2}\|_{\mathsf{F}}^2 - \|\mathbb{G}_\rho\|_{\mathsf{F}}^2 \overset{①}{=} (2\rho^2 - \frac{1}{2}) \cdot (\text{tr}(\mathbf{U}\mathbf{U}) - \text{tr}(\mathbf{U}^{\mathsf{T}}\mathbf{U})) \overset{②}{\leq} 0,$$

where step ① uses Inequalities (25) and (26); step ② uses $2\rho^2 - \frac{1}{2} \geq 0$ for all $\rho \in (\frac{1}{2}, \infty)$, and the fact that $\text{tr}(\mathbf{U}\mathbf{U}) - \text{tr}(\mathbf{U}\mathbf{U}^{\mathsf{T}}) \leq 0$ for all $\mathbf{U} \in \mathbb{R}^{r \times r}$. Therefore, we conclude that $\|\mathbb{G}_\rho\|_{\mathsf{F}} \geq \min(1, 2\rho)\|\mathbb{G}_{1/2}\|_{\mathsf{F}}$.

**Part (b-ii)**. We now prove that $\|\mathbb{G}_\rho\|_{\mathsf{F}} \leq \max(1, 2\rho)\|\mathbb{G}_{1/2}\|_{\mathsf{F}}$. We consider two cases. Case (**i**): $\rho \in (0, \frac{1}{2}]$. We have:

$$\|\mathbb{G}_{1/2}\|_{\mathsf{F}}^2 - \|\mathbb{G}_\rho\|_{\mathsf{F}}^2 \overset{①}{=} (2\rho^2 - \frac{1}{2}) \cdot (\text{tr}(\mathbf{U}\mathbf{U}) - \text{tr}(\mathbf{U}^{\mathsf{T}}\mathbf{U})) \overset{②}{\geq} 0,$$

where step ① uses Inequality (26); step ② uses $2\rho^2 - \frac{1}{2} \leq 0$ for all $\rho \in (0, \frac{1}{2}]$, and the fact that $\text{tr}(\mathbf{U}\mathbf{U}) - \text{tr}(\mathbf{U}\mathbf{U}^{\mathsf{T}}) \leq 0$ for all $\mathbf{U} \in \mathbb{R}^{r \times r}$.

Case (**ii**): $\rho \in (\frac{1}{2}, \infty)$. We have:

$$(2\rho)^2\|\mathbb{G}_{1/2}\|_{\mathsf{F}}^2 - \|\mathbb{G}_\rho\|_{\mathsf{F}}^2 \overset{①}{=} (4\rho^2 - 1) \cdot (\text{tr}(\mathbf{G}^{\mathsf{T}}\mathbf{G}) - \text{tr}(\mathbf{U}^{\mathsf{T}}\mathbf{U})) \overset{②}{\geq} 0,$$

where step ① uses Inequalities (25) and (26); step ② uses $4\rho^2 - 1 \geq 0$ for all $\rho \in (\frac{1}{2}, \infty)$, and Inequality (27). Therefore, we conclude that: $\|\mathbb{G}_\rho\|_{\mathsf{F}} \geq \min(1, 2\rho)\|\mathbb{G}_{1/2}\|_{\mathsf{F}}$.

$\square$

### B.6 PROOF OF LEMMA 2.13

*Proof.* Recall that the following first-order optimality conditions are equivalent for all $\mathbf{X} \in \mathbb{R}^{n \times r}$:

$$\left(\mathbf{0} \in \partial \mathcal{I}_{\mathcal{M}}(\mathbf{X}) + \nabla f(\mathbf{X})\right) \Leftrightarrow \left(\mathbf{0} \in \text{Proj}_{\mathbf{T}_{\mathbf{X}}\mathcal{M}}(\nabla f(\mathbf{X}))\right). \tag{28}$$

Therefore, we derive the following results:

$$
\begin{aligned}
\mathrm{dist}(\mathbf{0}, \partial \mathcal{I}_{\mathcal{M}}(\mathbf{X}) + \nabla f(\mathbf{X})) &= \inf_{\mathbf{R} \in \nabla f(\mathbf{X}) + \partial \mathcal{I}_{\mathcal{M}}(\mathbf{X})} \|\mathbf{R}\|_{\mathsf{F}} \\
&\overset{①}{=} \inf_{\mathbf{R} \in \mathrm{Proj}_{\mathbf{T}_{\mathbf{X}}\mathcal{M}}(\nabla f(\mathbf{X}))} \|\mathbf{R}\|_{\mathsf{F}} \\
&= \|\mathrm{Proj}_{\mathbf{T}_{\mathbf{X}}\mathcal{M}}(\nabla f(\mathbf{X}))\|_{\mathsf{F}} \\
&\overset{②}{=} \|\nabla f(\mathbf{X}) - \tfrac{1}{2}\mathbf{X}(\mathbf{X}^{\mathsf{T}}\nabla f(\mathbf{X}) + \nabla f(\mathbf{X})^{\mathsf{T}}\mathbf{X})\|_{\mathsf{F}} \\
&= \|(\mathbf{I} - \tfrac{1}{2}\mathbf{X}\mathbf{X}^{\mathsf{T}})(\nabla f(\mathbf{X}) - \mathbf{X}\nabla f(\mathbf{X})^{\mathsf{T}}\mathbf{X})\|_{\mathsf{F}} \\
&\overset{③}{\leq} \|\nabla f(\mathbf{X}) - \mathbf{X}\nabla f(\mathbf{X})^{\mathsf{T}}\mathbf{X}\|_{\mathsf{F}},
\end{aligned}
$$

where step ① uses Formulation (28); step ② uses $\mathrm{Proj}_{\mathbf{T}_{\mathbf{X}}\mathcal{M}}(\boldsymbol{\Delta}) = \boldsymbol{\Delta} - \tfrac{1}{2}\mathbf{X}(\boldsymbol{\Delta}^{\mathsf{T}}\mathbf{X} + \mathbf{X}^{\mathsf{T}}\boldsymbol{\Delta})$ for all $\boldsymbol{\Delta} \in \mathbb{R}^{n \times r}$ (Absil et al., 2008a); step ③ uses the norm inequality $\|\mathbf{A}\mathbf{B}\|_{\mathsf{F}} \leq \|\mathbf{A}\|\|\mathbf{B}\|_{\mathsf{F}}$, and fact that the matrix $\mathbf{I} - \tfrac{1}{2}\mathbf{X}\mathbf{X}^{\mathsf{T}}$ only contains eigenvalues that are $\tfrac{1}{2}$ or 1.

$\square$

# C   PROOFS FOR SECTION 4

## C.1   PROOF OF LEMMA 4.1

*Proof.* We define $L(\mathbf{X}, \mathbf{y}; \mathbf{z}; \beta, \mu) \triangleq f(\mathbf{X}) - g(\mathbf{X}) + h_{\mu}(\mathbf{y}) + \langle \mathbf{z}, \mathcal{A}(\mathbf{X}) - \mathbf{y} \rangle + \frac{\beta}{2}\|\mathcal{A}(\mathbf{X}) - \mathbf{y}\|_2^2$.

We define $\dot{\sigma} \triangleq (\sigma - 1)/(2 - \sigma)$, and $\ddot{\sigma} \triangleq (\sigma/(2 - \sigma))^2$.

**Part (a-i)**. Using the first-order optimality condition of $\mathbf{y}^{t+1} \in \arg\min_{\mathbf{y}} L(\mathbf{X}^{t+1}, \mathbf{y}, \mathbf{z}^t; \beta^t, \mu^t)$ in Algorithm 1, for all $t \geq 0$, we have:

$$
\begin{aligned}
\mathbf{0} &= \nabla h_{\mu^t}(\mathbf{y}^{t+1}) + \beta^t(\mathbf{y}^{t+1} - \mathbf{y}^t) + \nabla_{\mathbf{y}}\mathcal{S}(\mathbf{X}^{t+1}, \mathbf{y}^t; \mathbf{z}^t; \beta^t) \\
&\overset{①}{=} \nabla h_{\mu^t}(\mathbf{y}^{t+1}) + \beta^t(\mathbf{y}^{t+1} - \mathbf{y}^t) - \mathbf{z}^t + \beta^t(\mathbf{y}^t - \mathcal{A}(\mathbf{X}^{t+1})) \\
&= \nabla h_{\mu^t}(\mathbf{y}^{t+1}) - \mathbf{z}^t + \beta^t(\mathbf{y}^{t+1} - \mathcal{A}(\mathbf{X}^{t+1})) \\
&\overset{②}{=} \nabla h_{\mu^t}(\mathbf{y}^{t+1}) - \mathbf{z}^t + \tfrac{1}{\sigma}(\mathbf{z}^t - \mathbf{z}^{t+1}),
\end{aligned} \tag{29}
$$

where step ① uses $\nabla_{\mathbf{y}}\mathcal{S}(\mathbf{X}^{t+1}, \mathbf{y}; \mathbf{z}^t; \beta^t) = -\mathbf{z}^t + \beta^t(\mathbf{y} - \mathcal{A}(\mathbf{X}^{t+1}))$; step ② uses $\mathbf{z}^{t+1} = \mathbf{z}^t + \sigma\beta^t(\mathcal{A}(\mathbf{X}^{t+1}) - \mathbf{y}^{t+1})$.

**Part (a-ii)**. We obtain:

$$
\begin{aligned}
\partial h(\breve{\mathbf{y}}^{t+1}) - \mathbf{z}^t &\overset{①}{\ni} \beta^t(\mathbf{b} - \mathbf{y}^{t+1}) - \mathbf{z}^t \\
&\overset{②}{=} \beta^t \mathbf{y}^t - \nabla_{\mathbf{y}}\mathcal{S}^t(\mathbf{X}^{t+1}, \mathbf{y}^t; \mathbf{z}^t; \beta^t) - \beta^t \mathbf{y}^{t+1} - \mathbf{z}^t \\
&\overset{③}{=} \beta^t \mathbf{y}^t - \beta^t(\mathbf{y}^t - \mathcal{A}(\mathbf{X}^{t+1})) - \beta^t \mathbf{y}^{t+1} \\
&= \beta^t(\mathcal{A}(\mathbf{X}^{t+1}) - \mathbf{y}^{t+1}) \\
&\overset{④}{=} \tfrac{1}{\sigma}(\mathbf{z}^{t+1} - \mathbf{z}^t),
\end{aligned}
$$

where step ① uses the result in Lemma 2.5 that $\beta^t(\mathbf{b} - \mathbf{y}^{t+1}) \in \partial h(\breve{\mathbf{y}}^{t+1})$; step ② uses $\mathbf{b} \triangleq \mathbf{y}^t - \nabla_{\mathbf{y}}\mathcal{S}^t(\mathbf{X}^{t+1}, \mathbf{y}^t; \mathbf{z}^t; \beta^t)/\beta^t$, as shown in Algorithm 1; step ③ uses $\nabla_{\mathbf{y}}\mathcal{S}^t(\mathbf{X}^{t+1}, \mathbf{y}; \mathbf{z}^t; \beta^t) = -\mathbf{z}^t + \beta^t(\mathbf{y} - \mathcal{A}(\mathbf{X}^{t+1}))$; step ④ uses $\mathbf{z}^{t+1} - \mathbf{z}^t = \sigma\beta^t(\mathcal{A}(\mathbf{X}^{t+1}) - \mathbf{y}^{t+1})$.

**Part (b)**. First, we derive:

$$
\begin{aligned}
&\|\nabla h_{\mu^{t-1}}(\mathbf{y}^t) - \nabla h_{\mu^t}(\mathbf{y}^{t+1})\| \\
&\overset{①}{\leq} \|\nabla h_{\mu^{t-1}}(\mathbf{y}^t) - \nabla h_{\mu^t}(\mathbf{y}^t)\| + \|\nabla h_{\mu^t}(\mathbf{y}^t) - \nabla h_{\mu^t}(\mathbf{y}^{t+1})\| \\
&\overset{②}{\leq} \|\nabla h_{\mu^t}(\mathbf{y}^t) - \nabla h_{\mu^{t-1}}(\mathbf{y}^t)\| + \tfrac{1}{\mu^t}\|\mathbf{y}^{t+1} - \mathbf{y}^t\| \\
&\overset{③}{\leq} C_h(\tfrac{\mu^{t-1}}{\mu^t} - 1) + \tfrac{\beta^t}{\chi}\|\mathbf{y}^{t+1} - \mathbf{y}^t\|,
\end{aligned} \tag{30}
$$

where step ① uses $\|\mathbf{a} - \mathbf{b}\| \leq \|\mathbf{a} - \mathbf{c}\| + \|\mathbf{c} - \mathbf{b}\|$; step ② uses the fact that the function $h_{\mu^t}(\mathbf{y})$ is $\frac{1}{\mu^t}$-smooth *w.r.t.* $\mathbf{y}$ that: $\|\nabla h_{\mu^t}(\mathbf{y}^{t+1}) - \nabla h_{\mu^t}(\mathbf{y}^t)\| \leq \frac{1}{\mu^t}\|\mathbf{y}^{t+1} - \mathbf{y}^t\|$; step ③ uses the fact that $\|\nabla h_{\mu^t}(\mathbf{y}^t) - \nabla h_{\mu^{t-1}}(\mathbf{y}^t)\| \leq (\mu^{t-1}/\mu^t - 1)C_h$ which holds due to Lemma 2.4, and the equality $\mu^t \beta^t = \chi$.

Second, we have from Equality (29):

$$\forall t \geq 0, \ \mathbf{0} \in \sigma \nabla h_{\mu^t}(\mathbf{y}^{t+1}) - \sigma \mathbf{z}^t + (\mathbf{z}^t - \mathbf{z}^{t+1}),$$
$$\forall t \geq 1, \ \mathbf{0} \in \sigma \nabla h_{\mu^{t-1}}(\mathbf{y}^t) - \sigma \mathbf{z}^{t-1} + (\mathbf{z}^{t-1} - \mathbf{z}^t).$$

Combining these two equalities yields:

$$\forall t \geq 1, \ \mathbf{z}^{t+1} - \mathbf{z}^t = (\sigma - 1)(\mathbf{z}^{t-1} - \mathbf{z}^t) + \sigma(\nabla h_{\mu^t}(\mathbf{y}^{t+1}) - \nabla h_{\mu^{t-1}}(\mathbf{y}^t)).$$

Applying Lemma A.4 with $\mathbf{a}^+ = \mathbf{z}^{t+1} - \mathbf{z}^t$, $\mathbf{a} = \mathbf{z}^{t-1} - \mathbf{z}^t$, $\mathbf{b} = \sigma\{\nabla h_{\mu^t}(\mathbf{y}^{t+1}) - \nabla h_{\mu^{t-1}}(\mathbf{y}^t)\}$, and $\varrho = \sigma - 1 \in [0, 1)$, we have:

$$
\begin{aligned}
&\|\mathbf{z}^{t+1} - \mathbf{z}^t\|_2^2 \\
\leq \ & \tfrac{\varrho}{1-\varrho}(\|\mathbf{z}^{t-1} - \mathbf{z}^t\|_2^2 - \|\mathbf{z}^{t+1} - \mathbf{z}^t\|_2^2) + \tfrac{1}{(1-\varrho)^2}\|\sigma(\nabla h_{\mu^t}(\mathbf{y}^{t+1}) - \nabla h_{\mu^{t-1}}(\mathbf{y}^t))\|_2^2 \\
\overset{①}{=} \ & \dot{\sigma}(\|\mathbf{z}^t - \mathbf{z}^{t-1}\|_2^2 - \|\mathbf{z}^{t+1} - \mathbf{z}^t\|_2^2) + \ddot{\sigma}\|\nabla h_{\mu^t}(\mathbf{y}^{t+1}) - \nabla h_{\mu^{t-1}}(\mathbf{y}^t)\|_2^2 \\
\overset{②}{\leq} \ & \dot{\sigma}(\|\mathbf{z}^t - \mathbf{z}^{t-1}\|_2^2 - \|\mathbf{z}^{t+1} - \mathbf{z}^t\|_2^2) + 2\ddot{\sigma}\{\tfrac{(\beta^t)^2}{\chi^2}\|\mathbf{y}^{t+1} - \mathbf{y}^t\|_2^2 + C_h^2(\mu^{t-1}/\mu^t - 1)^2\} \\
\overset{③}{\leq} \ & \dot{\sigma}(\|\mathbf{z}^t - \mathbf{z}^{t-1}\|_2^2 - \|\mathbf{z}^{t+1} - \mathbf{z}^t\|_2^2) + 2\ddot{\sigma}\tfrac{(\beta^t)^2}{\chi^2}\|\mathbf{y}^{t+1} - \mathbf{y}^t\|_2^2 + 2\ddot{\sigma}C_h^2(\tfrac{2}{t} - \tfrac{2}{t+1}),
\end{aligned}
$$

where step ① uses the definitions of $\{\dot{\sigma}, \ddot{\sigma}\}$; step ② uses Inequality (30), and the inequality $(a+b)^2 \leq 2a^2 + 2b^2$ for all $a, b \in \mathbb{R}$; step ③ uses Lemma A.3 that $(\tfrac{\beta^t}{\beta^{t-1}} - 1)^2 \leq \tfrac{2}{t} - \tfrac{2}{t+1}$ for all $t \geq 1$;

$\square$

## C.2 PROOF OF LEMMA 4.3

*Proof.* **Part (a)**. We have:

$$\beta^{t+1} - \beta^t \cdot (1 + \xi) \overset{①}{=} \beta^0 \xi(t+1)^p - \beta^0 \xi t^p - \beta^t \xi \overset{②}{\leq} \beta^0 \xi - \beta^t \xi \overset{③}{\leq} 0,$$

where step ① uses $\beta^t = \beta^0(1 + \xi t^p)$; step ② uses $(t+1)^p - t^p \leq 1$ for all $p \in (0, 1)$; step ③ uses $\beta^0 \leq \beta^t$ and $\xi > 0$.

**Part (b)**. It holds with $\underline{\ell} = \overline{A}^2$ and $\overline{\ell} = \overline{A}^2 + L_f/\beta^0$.

$\square$

## C.3 PROOF OF LEMMA 4.4

*Proof.* We define $\overline{X} \triangleq \sqrt{r}$, $\overline{z} \triangleq \|\mathbf{z}^0\| + \tfrac{\sigma C_h}{2-\sigma}$, $\overline{y} \triangleq \overline{A}\sqrt{r} + \tfrac{2\overline{z}}{\beta^0}$, where $\sigma \in [1, 2)$.

We let $\underline{\Theta} \triangleq F(\bar{\mathbf{X}}) - \mu^0 C_h^2 - C_h(\overline{A}\sqrt{r} + \overline{y}) - \tfrac{\overline{z}^2}{2\beta^0}$, where $\bar{\mathbf{X}}$ is the optimal solution of Problem (1).

**Part (a)**. Given $\mathbf{X}^{t+1} \in \mathcal{M}$, we have: $\|\mathbf{X}^t\|_{\mathsf{F}} \leq \overline{X} \triangleq \sqrt{r}$.

**Part (b)**. We show that $\|\mathbf{z}^t\| \leq \overline{z}$. For all $t \geq 0$, we have:

$$
\begin{aligned}
\|\mathbf{z}^{t+1}\| \ & \overset{①}{\leq} \ \|(\sigma - 1)\mathbf{z}^t\| + \|(\sigma - 1)\mathbf{z}^t + \mathbf{z}^{t+1}\| \\
& \overset{②}{=} \ (\sigma - 1)\|\mathbf{z}^t\| + \|\sigma \partial h(\check{\mathbf{y}}^{t+1})\| \\
& \overset{③}{=} \ (\sigma - 1)\|\mathbf{z}^t\| + \sigma C_h,
\end{aligned}
$$

step ① uses the triangle inequality; step ② uses $\mathbf{z}^{t+1} + (\sigma - 1)\mathbf{z}^t \in \sigma \partial h(\check{\mathbf{y}}^{t+1})$, as shown in Lemma 4.1(*a*); step ③ uses $C_h$-Lipschitz continuity of $h(\mathbf{y})$. Applying Lemma A.5 with $\mathbf{a}_t = \|\mathbf{z}^{t+1}\|$, $c = \sigma C_h$, and $\varrho = \sigma - 1 \in [0, 1)$, we have:

$$\forall t \geq 0, \ \|\mathbf{z}^{t+1}\| \leq \|\mathbf{z}^0\| + \tfrac{c}{1-\varrho} = \|\mathbf{z}^0\| + \tfrac{\sigma C_h}{2-\sigma} \triangleq \overline{z}.$$

**Part (c)**. We show that $\|\mathbf{y}^t\| \leq \overline{\mathrm{y}}$. For all $t \geq 0$, we have:

$$
\begin{aligned}
\|\mathbf{y}^{t+1}\| &= \|\mathcal{A}(\mathbf{X}^{t+1}) - \tfrac{\mathbf{z}^{t+1} - \mathbf{z}^t}{\sigma \beta^t}\| \\
&\overset{①}{\leq} \|\mathcal{A}(\mathbf{X}^{t+1})\| + \tfrac{1}{\beta^0} \|\mathbf{z}^{t+1} - \mathbf{z}^t\| \\
&\overset{②}{\leq} \overline{\mathrm{A}}\sqrt{r} + \tfrac{1}{\beta^0} \cdot 2\overline{\mathrm{z}} \triangleq \overline{\mathrm{y}},
\end{aligned}
$$

where step ① uses the triangle inequality, $\sigma \geq 1$, and $\frac{1}{\beta^t} \leq \frac{1}{\beta^0}$; step ② uses $\|\mathcal{A}(\mathbf{X})\|_{\mathsf{F}} \leq \overline{\mathrm{A}}\|\mathbf{X}\|_{\mathsf{F}} \leq \overline{\mathrm{A}}\sqrt{r}$, and $\|\mathbf{z}^t\| \leq \overline{\mathrm{z}}$.

**Part (d)**. We show that $\Theta^t \geq \underline{\Theta}$. For all $t \geq 1$, we have:

$$
\begin{aligned}
\Theta^t &\triangleq L(\mathbf{X}^t, \mathbf{y}^t, \mathbf{z}^t; \beta^t, \mu^{t-1}) + \mu^{t-1}C_h^2 + \mathbb{T}^t + \mathbb{Z}^t + \mathbb{X}^t \\
&\overset{①}{\geq} f(\mathbf{X}^t) - g(\mathbf{X}^t) + h_{\mu^{t-1}}(\mathbf{y}^t) + \langle \mathbf{z}^t, \mathcal{A}(\mathbf{X}^t) - \mathbf{y}^t \rangle + \tfrac{\beta^t}{2}\|\mathcal{A}(\mathbf{X}^t) - \mathbf{y}^t\|_2^2 \\
&= f(\mathbf{X}^t) - g(\mathbf{X}^t) + h_{\mu^{t-1}}(\mathbf{y}^t) + \tfrac{\beta^t}{2}\|\mathcal{A}(\mathbf{X}^t) - \mathbf{y}^t + \mathbf{z}^t/\beta^t\|_2^2 - \tfrac{\beta^t}{2}\|\mathbf{z}^t/\beta^t\|_2^2 \\
&\overset{②}{\geq} f(\mathbf{X}^t) - g(\mathbf{X}^t) + h_{\mu^{t-1}}(\mathcal{A}(\mathbf{X}^t)) - C_h\|\mathcal{A}(\mathbf{X}^t) - \mathbf{y}^t\| - \tfrac{1}{2\beta^t}\|\mathbf{z}^t\|_2^2 \\
&\overset{③}{\geq} f(\mathbf{X}^t) - g(\mathbf{X}^t) + h(\mathcal{A}(\mathbf{X}^t)) - \mu^{t-1}C_h^2 - C_h(\|\mathcal{A}(\mathbf{X}^t)\| + \|\mathbf{y}^t\|) - \tfrac{1}{2\beta^t}\|\mathbf{z}^t\|_2^2 \\
&\overset{④}{\geq} F(\overline{\mathbf{X}}) - \mu^0 C_h^2 - C_h(\overline{\mathrm{A}}\sqrt{r} + \overline{\mathrm{y}}) - \tfrac{\overline{\mathrm{z}}^2}{2\beta^0} \triangleq \underline{\Theta},
\end{aligned}
$$

where step ① uses the definition of $L(\mathbf{X}, \mathbf{y}; \mathbf{z}; \beta; \mu)$ and the positivity of $\{\mu^t, \mathbb{T}^t, \mathbb{Z}^t, \mathbb{X}^t\}$; step ② uses the $L_h$-Lipschitz continuity of $h_{\mu^{t-1}}(\mathbf{y})$, ensuring $h_{\mu^{t-1}}(\mathbf{y}^t) \geq h_{\mu^{t-1}}(\mathbf{y}) - C_h\|\mathbf{y}^t - \mathbf{y}\|$, with the specific choice of $\mathbf{y} = \mathcal{A}(\mathbf{X}^t)$; step ③ uses $h(\mathbf{y}) - h_\mu(\mathbf{y}) \leq \mu C_h^2$, which has been shown in Lemma 2.3; step ④ uses $\mu^t \leq \mu^0$, $\beta^t \geq \beta^0$, $\|\mathcal{A}(\mathbf{X})\| \leq \overline{\mathrm{A}}\|\mathbf{X}\|_{\mathsf{F}} \leq \overline{\mathrm{A}}\sqrt{r}$ for all $\mathbf{X} \in \mathcal{M}$; $\|\mathbf{y}^t\| \leq \overline{\mathrm{y}}$, and $\|\mathbf{z}^t\| \leq \overline{\mathrm{z}}$.

$\square$

## C.4    Proof of Lemma 4.5

*Proof.* We define $L(\mathbf{X}, \mathbf{y}; \mathbf{z}; \beta, \mu) \triangleq f(\mathbf{X}) - g(\mathbf{X}) + h_\mu(\mathbf{y}) + \langle \mathbf{z}, \mathcal{A}(\mathbf{X}) - \mathbf{y} \rangle + \tfrac{\beta}{2}\|\mathcal{A}(\mathbf{X}) - \mathbf{y}\|_2^2$.

We define $\omega \triangleq \frac{1}{\sigma} + \frac{\xi}{2\sigma^2} + \frac{\varepsilon_z}{\sigma^2}$.

We define $\mathbb{Z}^t \triangleq \omega \dot{\sigma} \sigma^2 \beta^{t-1} \|\mathcal{A}(\mathbf{X}^t) - \mathbf{y}^t\|_2^2 = \frac{\omega \dot{\sigma}}{\beta^{t-1}} \|\mathbf{z}^t - \mathbf{z}^{t-1}\|_2^2$, where we use $\mathbf{z}^{t+1} - \mathbf{z}^t = \beta^t \sigma(\mathcal{A}(\mathbf{X}^{t+1}) - \mathbf{y}^{t+1})$.

**Part (a)**. We focus on the sufficient decrease for variables $\{\mu, \mathbf{y}\}$. First, we have:

$$
\begin{aligned}
\Xi &\triangleq \langle \mathbf{y}^t - \mathbf{y}^{t+1}, \mathbf{z}^t \rangle + \tfrac{\beta^t}{2}\|\mathbf{y}^{t+1} - \mathcal{A}(\mathbf{X}^{t+1})\|_2^2 - \tfrac{\beta^t}{2}\|\mathbf{y}^t - \mathcal{A}(\mathbf{X}^{t+1})\|_2^2 \\
&\overset{①}{=} \langle \mathbf{y}^t - \mathbf{y}^{t+1}, \mathbf{z}^t + \beta^t(\mathcal{A}(\mathbf{X}^{t+1}) - \mathbf{y}^{t+1}) \rangle - \tfrac{\beta^t}{2}\|\mathbf{y}^{t+1} - \mathbf{y}^t\|_2^2 \\
&\overset{②}{=} -\tfrac{\beta^t}{2}\|\mathbf{y}^{t+1} - \mathbf{y}^t\|_2^2 + \langle \mathbf{y}^t - \mathbf{y}^{t+1}, \mathbf{z}^t + \tfrac{1}{\sigma}(\mathbf{z}^{t+1} - \mathbf{z}^t) \rangle \\
&\overset{③}{=} -\tfrac{\beta^t}{2}\|\mathbf{y}^{t+1} - \mathbf{y}^t\|_2^2 + \langle \mathbf{y}^t - \mathbf{y}^{t+1}, \nabla h_{\mu^t}(\mathbf{y}^{t+1}) \rangle \\
&\overset{④}{\leq} \{\tfrac{1}{\chi} - \beta^t\}\tfrac{1}{2}\|\mathbf{y}^{t+1} - \mathbf{y}^t\|_2^2 + h_{\mu^t}(\mathbf{y}^t) - h_{\mu^t}(\mathbf{y}^{t+1}),
\end{aligned}
\tag{31}
$$

where step ① uses the Pythagoras Relation that $\frac{1}{2}\|\mathbf{y}^+ - \mathbf{a}\|_2^2 - \frac{1}{2}\|\mathbf{y} - \mathbf{a}\|_2^2 = -\frac{1}{2}\|\mathbf{y}^+ - \mathbf{y}\|_2^2 + \langle \mathbf{y} - \mathbf{y}^+, \mathbf{a} - \mathbf{y}^+ \rangle$ for all $\mathbf{y}, \mathbf{y}^+, \mathbf{a} \in \mathbb{R}^m$; step ② uses $\mathbf{z}^{t+1} - \mathbf{z}^t = \sigma \beta^t(\mathcal{A}(\mathbf{X}^{t+1}) - \mathbf{y}^{t+1})$; step ③ uses $\nabla h_{\mu^t}(\mathbf{y}^{t+1}) = \mathbf{z}^t + \frac{1}{\sigma}(\mathbf{z}^{t+1} - \mathbf{z}^t)$, as shown in Lemma 4.1(*a*); step ④ uses the fact that the function

$h_{\mu^t}(\mathbf{y})$ is $(1/\mu^t)$-weakly convex *w.r.t* $\mathbf{y}$, and $\mu^t\beta^t = \chi$. Furthermore, we have:

$$L(\mathbf{X}^{t+1}, \mathbf{y}^{t+1}; \mathbf{z}^t; \beta^t, \mu^t) - L(\mathbf{X}^{t+1}, \mathbf{y}^t, \mathbf{z}^t; \beta^t, \mu^{t-1})$$

$$\overset{①}{=} h_{\mu^t}(\mathbf{y}^{t+1}) - h_{\mu^{t-1}}(\mathbf{y}^t) + \underbrace{\langle \mathbf{y}^t - \mathbf{y}^{t+1}, \mathbf{z}^t \rangle + \tfrac{\beta^t}{2}\|\mathbf{y}^{t+1} - \mathcal{A}(\mathbf{X}^{t+1})\|_2^2 - \tfrac{\beta^t}{2}\|\mathbf{y}^t - \mathcal{A}(\mathbf{X}^{t+1})\|_2^2}_{=\Xi}$$

$$\overset{②}{\leq} \tfrac{1/\chi-1}{2}\beta^t\|\mathbf{y}^{t+1} - \mathbf{y}^t\|_2^2 + h_{\mu^t}(\mathbf{y}^t) - h_{\mu^{t-1}}(\mathbf{y}^t)$$

$$\overset{③}{=} \tfrac{1/\chi-1}{2}\beta^t\|\mathbf{y}^{t+1} - \mathbf{y}^t\|_2^2 + (\mu^{t-1} - \mu^t)C_h^2, \tag{32}$$

where step ① uses the definition of $L(\mathbf{X}, \mathbf{y}; \mathbf{z}; \beta, \mu)$; step ② uses Inequality (31); step ③ uses Lemma 2.3 that $h_{\mu^t}(\mathbf{y}) - h_{\mu^{t-1}}(\mathbf{y}) \leq \min\{\tfrac{\mu^{t-1}}{2\mu^t}, 1\} \cdot (\mu^{t-1} - \mu^t)C_h^2 \leq (\mu^{t-1} - \mu^t)C_h^2$ for all $\mathbf{y}$.

**Part (b).** We focus on the sufficient decrease for variables $\{\mathbf{z}, \beta\}$. We have:

$$L(\mathbf{X}^{t+1}, \mathbf{y}^{t+1}; \mathbf{z}^{t+1}; \beta^{t+1}, \mu^t) - L(\mathbf{X}^{t+1}, \mathbf{y}^{t+1}; \mathbf{z}^t; \beta^t, \mu^t) + \varepsilon_z\beta^t\|\mathcal{A}(\mathbf{X}^{t+1}) - \mathbf{y}^{t+1}\|_2^2$$

$$\overset{①}{=} \langle \mathcal{A}(\mathbf{X}^{t+1}) - \mathbf{y}^{t+1}, \mathbf{z}^{t+1} - \mathbf{z}^t \rangle + \tfrac{\beta^{t+1}-\beta^t}{2}\|\mathcal{A}(\mathbf{X}^{t+1}) - \mathbf{y}^{t+1}\|_2^2 + \varepsilon_z\beta^t\|\mathcal{A}(\mathbf{X}^{t+1}) - \mathbf{y}^{t+1}\|_2^2$$

$$\overset{②}{=} \{\tfrac{1}{\sigma} + \tfrac{\beta^{t+1}-\beta^t}{2\sigma^2\beta^t} + \tfrac{\varepsilon_z}{\sigma^2}\} \cdot \tfrac{1}{\beta^t}\|\mathbf{z}^{t+1} - \mathbf{z}^t\|_2^2$$

$$\overset{③}{=} \underbrace{\{\tfrac{1}{\sigma} + \tfrac{\xi}{2\sigma^2} + \tfrac{\varepsilon_z}{\sigma^2}\}}_{\triangleq\omega} \cdot \tfrac{1}{\beta^t}\|\mathbf{z}^{t+1} - \mathbf{z}^t\|_2^2$$

$$\overset{④}{\leq} \tfrac{\omega\dot{\sigma}}{\beta^t}(\|\mathbf{z}^t - \mathbf{z}^{t-1}\|_2^2 - \|\mathbf{z}^{t+1} - \mathbf{z}^t\|_2^2) + \tfrac{2\omega\ddot{\sigma}}{\chi^2}\beta^t\|\mathbf{y}^{t+1} - \mathbf{y}^t\|_2^2 + \tfrac{2\omega\ddot{\sigma}}{\beta^t}C_h^2(\tfrac{2}{t} - \tfrac{2}{t+1})$$

$$\overset{⑤}{\leq} \underbrace{\tfrac{\omega\dot{\sigma}}{\beta^{t-1}}\|\mathbf{z}^t - \mathbf{z}^{t-1}\|_2^2}_{\triangleq\mathbb{Z}^t} - \tfrac{\omega\dot{\sigma}}{\beta^t}\|\mathbf{z}^{t+1} - \mathbf{z}^t\|_2^2 + \tfrac{2\omega\ddot{\sigma}}{\chi^2}\beta^t\|\mathbf{y}^{t+1} - \mathbf{y}^t\|_2^2 + \underbrace{\tfrac{2\omega\ddot{\sigma}}{\beta^0}C_h^2(\tfrac{2}{t} - \tfrac{2}{t+1})}_{=\mathbb{T}^t-\mathbb{T}^{t+1}}, \tag{33}$$

where step ① uses the definition of $L(\mathbf{X}, \mathbf{y}; \mathbf{z}; \beta; \mu)$; step ② uses $\mathbf{z}^{t+1} - \mathbf{z}^t = \sigma\beta^t(\mathcal{A}(\mathbf{X}^{t+1}) - \mathbf{y}^{t+1})$; step ③ uses $\beta^{t+1} \leq (1+\xi)\beta^t$; step ④ uses the upper bound for $\|\mathbf{z}^{t+1} - \mathbf{z}^t\|_2^2$ as shown in Lemma 4.1(**b**); step ⑤ uses $\beta^t \geq \beta^{t-1} \geq \beta^0$.

Adding Inequalities (32) and (33) together, we have:

$$L(\mathbf{X}^{t+1}, \mathbf{y}^{t+1}, \mathbf{z}^{t+1}; \beta^{t+1}, \mu^t) - L(\mathbf{X}^{t+1}, \mathbf{y}^t, \mathbf{z}^t; \beta^t, \mu^{t-1}) + (\mu^t - \mu^{t-1})C_h^2$$

$$+ \mathbb{T}^{t+1} - \mathbb{T}^t + \mathbb{Z}^{t+1} - \mathbb{Z}^t + \varepsilon_z\beta^t\|\mathcal{A}(\mathbf{X}^{t+1}) - \mathbf{y}^{t+1}\|_2^2$$

$$\leq \tfrac{1}{2}\{\tfrac{1}{\chi} - 1 + \tfrac{4\omega\ddot{\sigma}}{\chi^2}\}\beta^t\|\mathbf{y}^{t+1} - \mathbf{y}^t\|_2^2$$

$$\overset{①}{\leq} \underbrace{\tfrac{1}{2}\{-1 + \tfrac{1+4\omega\ddot{\sigma}}{\chi}\}}_{\triangleq -\varepsilon_y}\beta^t\|\mathbf{y}^{t+1} - \mathbf{y}^t\|_2^2,$$

where step ① uses $\chi \geq 1$.

$\square$

## C.5 PROOF OF LEMMA 4.6

*Proof.* We define $\mathcal{S}(\mathbf{X}, \mathbf{y}^t; \mathbf{z}^t; \beta^t) \triangleq f(\mathbf{X}) + \langle \mathbf{z}^t, \mathcal{A}(\mathbf{X}) - \mathbf{y}^t \rangle + \tfrac{\beta^t}{2}\|\mathcal{A}(\mathbf{X}) - \mathbf{y}^t\|_2^2$.

We let $\mathbf{G}^t \in \nabla_{\mathbf{X}}\mathcal{S}(\mathbf{X}_c^t, \mathbf{y}^t; \mathbf{z}^t; \beta^t) - \partial g(\mathbf{X}^t)$.

We define $\mathbb{X}^t \triangleq \tfrac{1}{2}(\alpha + \theta\alpha)\ell(\beta^t)\|\mathbf{X}^t - \mathbf{X}^{t-1}\|_{\mathsf{F}}^2$.

We define $\varepsilon_x' \triangleq (\theta - 1 - \alpha - \theta\alpha) - (1+\xi)(\alpha + \theta\alpha) > 0$, and $\varepsilon_x \triangleq \tfrac{1}{2}\varepsilon_x'\underline{\ell} > 0$.

First, using the optimality condition of $\mathbf{X}^{t+1} \in \mathcal{M}$, we have:

$$\langle \mathbf{X}^{t+1} - \mathbf{X}^t, \mathbf{G}^t \rangle + \tfrac{\theta\ell(\beta^t)}{2}\|\mathbf{X}^{t+1} - \mathbf{X}_c^t\|_{\mathsf{F}}^2 \leq \langle \mathbf{X}^t - \mathbf{X}^t, \mathbf{G}^t \rangle + \tfrac{\theta\ell(\beta^t)}{2}\|\mathbf{X}^t - \mathbf{X}_c^t\|_{\mathsf{F}}^2. \tag{34}$$

Second, we have:

$$L(\mathbf{X}^{t+1}, \mathbf{y}^t, \mathbf{z}^t; \mu^t, \beta^t) - L(\mathbf{X}^t, \mathbf{y}^t, \mathbf{z}^t; \mu^t, \beta^t)$$

$$= \mathcal{S}(\mathbf{X}^{t+1}, \mathbf{y}^t; \mathbf{z}^t; \beta^t) - \mathcal{S}(\mathbf{X}^t, \mathbf{y}^t; \mathbf{z}^t; \beta^t) + g(\mathbf{X}^t) - g(\mathbf{X}^{t+1})$$

$$\overset{①}{\leq} \frac{\ell(\beta^t)}{2}\|\mathbf{X}^{t+1} - \mathbf{X}^t\|_{\mathsf{F}}^2 + \langle \mathbf{X}^{t+1} - \mathbf{X}^t, \nabla_{\mathbf{X}}\mathcal{S}(\mathbf{X}^t, \mathbf{y}^t; \mathbf{z}^t; \beta^t)\rangle + \langle \mathbf{X}^t - \mathbf{X}^{t+1}, \partial g(\mathbf{X}^t)\rangle, \quad (35)$$

where step ① uses the $\ell(\beta^t)$-smoothness of $\mathcal{S}(\mathbf{X}, \mathbf{y}^t; \mathbf{z}^t; \beta^t)$ and convexity of $g(\mathbf{X})$;

Third, we derive:

$$\langle \mathbf{X}^{t+1} - \mathbf{X}^t, \nabla_{\mathbf{X}}\mathcal{S}(\mathbf{X}^t, \mathbf{y}^t; \mathbf{z}^t; \beta^t) - \nabla_{\mathbf{X}}\mathcal{S}(\mathbf{X}_{\mathsf{c}}^t, \mathbf{y}^t; \mathbf{z}^t; \beta^t)\rangle$$

$$\overset{①}{\leq} \|\mathbf{X}^{t+1} - \mathbf{X}^t\|_{\mathsf{F}} \cdot \|\nabla_{\mathbf{X}}\mathcal{S}(\mathbf{X}^t, \mathbf{y}^t; \mathbf{z}^t; \beta^t) - \nabla_{\mathbf{X}}\mathcal{S}(\mathbf{X}_{\mathsf{c}}^t, \mathbf{y}^t; \mathbf{z}^t; \beta^t)\|_{\mathsf{F}}$$

$$\overset{②}{\leq} \|\mathbf{X}^{t+1} - \mathbf{X}^t\|_{\mathsf{F}} \cdot \ell(\beta^t)\|\mathbf{X}^t - \mathbf{X}_{\mathsf{c}}^t\|_{\mathsf{F}}$$

$$\overset{③}{\leq} \alpha\ell(\beta^t)\|\mathbf{X}^{t+1} - \mathbf{X}^t\|_{\mathsf{F}} \cdot \|\mathbf{X}^t - \mathbf{X}^{t-1}\|_{\mathsf{F}}$$

$$\overset{④}{\leq} \frac{\alpha\ell(\beta^t)}{2}\|\mathbf{X}^{t+1} - \mathbf{X}^t\|_{\mathsf{F}}^2 + \frac{\alpha\ell(\beta^t)}{2}\|\mathbf{X}^t - \mathbf{X}^{t-1}\|_{\mathsf{F}}^2, \quad (36)$$

where step ① uses the norm inequality; step ② uses the $\ell(\beta^t)$-smoothness of $\mathcal{S}(\mathbf{X}, \mathbf{y}^t; \mathbf{z}^t; \beta^t)$; step ③ uses $\mathbf{X}_{\mathsf{c}}^t = \mathbf{X}^t + \alpha(\mathbf{X}^t - \mathbf{X}^{t-1})$; step ④ uses $ab \leq \frac{1}{2}a^2 + \frac{1}{2}b^2$ for all $a \in \mathbb{R}$ and $b \in \mathbb{R}$.

Summing Inequalities (34),(36), and (35), we obtain:

$$L(\mathbf{X}^{t+1}, \mathbf{y}^t, \mathbf{z}^t; \mu^t, \beta^t) - L(\mathbf{X}^t, \mathbf{y}^t, \mathbf{z}^t; \mu^t, \beta^t)$$

$$\leq \frac{\ell(\beta^t)}{2}\{(1+\alpha)\|\mathbf{X}^{t+1} - \mathbf{X}^t\|_{\mathsf{F}}^2 + \alpha\|\mathbf{X}^t - \mathbf{X}^{t-1}\|_{\mathsf{F}} + \theta\|\mathbf{X}^t - \mathbf{X}_{\mathsf{c}}^t\|_{\mathsf{F}}^2 - \theta\|\mathbf{X}^{t+1} - \mathbf{X}_{\mathsf{c}}^t\|_{\mathsf{F}}^2\}$$

$$\overset{①}{=} \frac{\ell(\beta^t)}{2}\{(1+\alpha)\|\mathbf{X}^{t+1} - \mathbf{X}^t\|_{\mathsf{F}}^2 + (\alpha + \theta\alpha^2)\|\mathbf{X}^t - \mathbf{X}^{t-1}\|_{\mathsf{F}}^2 - \theta\|\mathbf{X}^{t+1} - \mathbf{X}^t - \alpha(\mathbf{X}^t - \mathbf{X}^{t-1})\|_{\mathsf{F}}^2\}$$

$$\overset{②}{\leq} \frac{\ell(\beta^t)}{2}\{(1+\alpha)\|\mathbf{X}^{t+1} - \mathbf{X}^t\|_{\mathsf{F}}^2 + (\alpha + \theta\alpha^2)\|\mathbf{X}^t - \mathbf{X}^{t-1}\|_{\mathsf{F}}^2$$

$$+\theta(\alpha - 1)\|\mathbf{X}^{t+1} - \mathbf{X}^t\|_{\mathsf{F}}^2 - \theta\alpha(\alpha - 1)\|\mathbf{X}^t - \mathbf{X}^{t-1}\|_{\mathsf{F}}^2\}$$

$$= \underbrace{\frac{1}{2}(\alpha + \theta\alpha)\ell(\beta^t)\|\mathbf{X}^t - \mathbf{X}^{t-1}\|_{\mathsf{F}}^2}_{\triangleq \mathbb{X}^t} + \frac{\ell(\beta^t)}{2} \cdot \|\mathbf{X}^{t+1} - \mathbf{X}^t\|_{\mathsf{F}}^2 \cdot \{1 + \alpha + \theta\alpha - \theta\}$$

$$= \mathbb{X}^t - \mathbb{X}^{t+1} + \frac{1}{2} \cdot \|\mathbf{X}^{t+1} - \mathbf{X}^t\|_{\mathsf{F}}^2 \cdot \{\ell(\beta^t)(1 + \alpha + \theta\alpha - \theta) + \ell(\beta^{t+1})(\alpha + \theta\alpha)\}$$

$$\overset{③}{\leq} \mathbb{X}^t - \mathbb{X}^{t+1} + \frac{1}{2} \cdot \|\mathbf{X}^{t+1} - \mathbf{X}^t\|_{\mathsf{F}}^2 \cdot \ell(\beta^t)\underbrace{\{(1 + \alpha + \theta\alpha - \theta) + (1 + \xi)(\alpha + \theta\alpha)\}}_{\triangleq -\varepsilon_x'}$$

$$\overset{④}{\leq} \mathbb{X}^t - \mathbb{X}^{t+1} - \frac{1}{2} \cdot \|\mathbf{X}^{t+1} - \mathbf{X}^t\|_{\mathsf{F}}^2 \cdot \varepsilon_x' \cdot \beta^t\underline{\ell}$$

$$\overset{⑤}{=} \mathbb{X}^t - \mathbb{X}^{t+1} - \varepsilon_x \cdot \beta^t\|\mathbf{X}^{t+1} - \mathbf{X}^t\|_{\mathsf{F}}^2,$$

where step ① uses $\mathbf{X}_{\mathsf{c}}^t = \mathbf{X}^t + \alpha(\mathbf{X}^t - \mathbf{X}^{t-1})$; step ② uses Lemma A.1 with $\mathbf{a} = \mathbf{X}^{t+1} - \mathbf{X}^t$, and $\mathbf{b} = \mathbf{X}^t - \mathbf{X}^{t-1}$; step ③ uses the fact that $\ell(\beta^{t+1}) \leq (1 + \xi)\ell(\beta^t)$, which is implied by $\beta^{t+1} \leq (1 + \xi)\beta^t$; step ④ uses Lemma 4.3 that $\beta^t\underline{\ell} \leq \ell(\beta^t) \leq \beta^t\overline{\ell}$; step ⑤ uses $\varepsilon_x \triangleq \frac{1}{2}\varepsilon_x'\underline{\ell} > 0$.

$$\square$$

## C.6 PROOF OF LEMMA 4.7

*Proof.* We define: $\Theta^t \triangleq L(\mathbf{X}^t, \mathbf{y}^t; \mathbf{z}^t; \beta^t, \mu^{t-1}) + \mu^{t-1}C_h^2 + \mathbb{T}^t + \mathbb{Z}^t + \mathbb{X}^t,$

We define $\tilde{e}_t \triangleq \|\mathbf{y}^t - \mathbf{y}^{t-1}\|^2 + \|\mathcal{A}(\mathbf{X}^t) - \mathbf{y}^t\|^2 + \|\mathbf{X}^t - \mathbf{X}^{t-1}\|_{\mathsf{F}}^2.$

**Part (a)**. Using Lemma 4.5, we have:

$$L(\mathbf{X}^{t+1}, \mathbf{y}^{t+1}; \mathbf{z}^{t+1}; \beta^{t+1}, \mu^t) - L(\mathbf{X}^{t+1}, \mathbf{y}^t; \mathbf{z}^t; \beta^t, \mu^{t-1}) - (\mu^{t-1} - \mu^t)C_h^2$$

$$\leq \mathbb{T}^t - \mathbb{T}^{t+1} + \mathbb{Z}^t - \mathbb{Z}^{t+1} - \varepsilon_y\beta^t\|\mathbf{y}^{t+1} - \mathbf{y}^t\|_2^2 - \varepsilon_z\beta^t\|\mathcal{A}(\mathbf{X}^{t+1}) - \mathbf{y}^{t+1}\|_2^2. \quad (37)$$

Using Lemma 4.6, we have:

$$L(\mathbf{X}^{t+1}, \mathbf{y}^t; \mathbf{z}^t; \beta^t, \mu^{t-1}) - L(\mathbf{X}^t, \mathbf{y}^t; \mathbf{z}^t; \beta^t, \mu^{t-1}) \leq \mathbb{X}^t - \mathbb{X}^{t+1} - \varepsilon_x \beta^t \|\mathbf{X}^{t+1} - \mathbf{X}^t\|_{\mathsf{F}}^2.$$

Adding these two inequalities together and using the definition of $\Theta^t$, we have:

$$\begin{aligned}
\Theta^t - \Theta^{t+1} &\geq \varepsilon_y \beta^t \|\mathbf{y}^{t+1} - \mathbf{y}^t\|_2^2 + \varepsilon_x \beta^t \|\mathbf{X}^{t+1} - \mathbf{X}^t\|_{\mathsf{F}}^2 + \varepsilon_z \beta^t \|\mathcal{A}(\mathbf{X}^{t+1}) - \mathbf{y}^{t+1}\|_2^2 \\
&\geq \min(\varepsilon_y, \varepsilon_x, \varepsilon_z) \cdot \beta^t \cdot \tilde{e}_{t+1}.
\end{aligned}$$

**Part (b)**. Telescoping this inequality over $t$ from 1 to $T$, we have:

$$\begin{aligned}
\sum_{t=1}^T \beta^t \tilde{e}_{t+1} &\leq \frac{1}{\min(\varepsilon_y, \varepsilon_x, \varepsilon_z)} \cdot \sum_{t=1}^T (\Theta^t - \Theta^{t+1}) \\
&= \frac{1}{\min(\varepsilon_y, \varepsilon_x, \varepsilon_z)} \cdot (\Theta^1 - \Theta^{T+1}) \\
&\overset{\text{①}}{\leq} \frac{1}{\min(\varepsilon_y, \varepsilon_x, \varepsilon_z)} \cdot (\Theta^1 - \underline{\Theta}),
\end{aligned} \tag{38}$$

where step ① uses $\Theta^t \geq \underline{\Theta}$. Furthermore, we have:

$$\sum_{t=1}^T \beta^t \tilde{e}_{t+1} = \sum_{t=1}^T \frac{1}{\beta^t}(\beta^t)^2 \tilde{e}_{t+1} \geq \frac{1}{\beta^T} \sum_{t=1}^T (\beta^t)^2 \tilde{e}_{t+1} \overset{\text{①}}{\geq} \frac{1}{3T\beta^T}(\sum_{t=1}^T \beta^t e^{t+1})^2, \tag{39}$$

where step ① uses $\sum_{i=1}^n \mathbf{x}_i^2 \geq \frac{1}{n}(\sum_{i=1}^n |\mathbf{x}_i|)^2$ for all $\mathbf{x} \in \mathbb{R}^n$. Combining Inequalities (38) and (39), we have: $\sum_{t=1}^T \beta^t e^{t+1} \leq \{\frac{\Theta^1 - \underline{\Theta}}{\min(\varepsilon_y, \varepsilon_x, \varepsilon_z)} \cdot 3T\beta^T\}^{1/2} = \mathcal{O}(T^{(1+p)/2})$.

$\square$

### C.7 Proof of Theorem 4.8

*Proof.* We define $\mathrm{Crit}(\mathbf{X}, \mathbf{y}, \mathbf{z}) \triangleq \|\mathcal{A}(\mathbf{X}) - \mathbf{y}\| + \|\partial h(\mathbf{y}) - \mathbf{z}\| + \|\mathrm{Proj}_{\mathbf{T}_{\mathbf{x}}\mathcal{M}}(\nabla f(\mathbf{X}) - \partial g(\mathbf{X}) + \mathcal{A}^{\mathsf{T}}(\mathbf{z}))\|_{\mathsf{F}}$.

We define $\dot{\mathbf{G}} \triangleq \nabla f(\mathbf{X}^t) - \partial g(\mathbf{X}^t) + \mathcal{A}^{\mathsf{T}}(\mathbf{z}^t)$.

We define $\ddot{\mathbf{G}} \triangleq \nabla f(\mathbf{X}_{\mathsf{c}}^t) - \partial g(\mathbf{X}^t) + \mathcal{A}^{\mathsf{T}}(\mathbf{z}^t + \beta^t \mathcal{A}(\mathbf{X}_{\mathsf{c}}^t) - \mathbf{y}^t) + \theta\ell(\beta^t)(\mathbf{X}^{t+1} - \mathbf{X}_{\mathsf{c}}^t)$.

We first derive the following inequalities:

$$\begin{aligned}
&\|\ddot{\mathbf{G}} - \dot{\mathbf{G}}\|_{\mathsf{F}} \\
&\overset{\text{①}}{=} \|\nabla f(\mathbf{X}^t) - \nabla f(\mathbf{X}_{\mathsf{c}}^t) - \beta^t \mathcal{A}^{\mathsf{T}}(\mathcal{A}(\mathbf{X}_{\mathsf{c}}^t) - \mathbf{y}^t) - \theta\ell(\beta^t)(\mathbf{X}^{t+1} - \mathbf{X}_{\mathsf{c}}^t)\|_{\mathsf{F}} \\
&\overset{\text{②}}{\leq} L_f \|\mathbf{X}^t - \mathbf{X}_{\mathsf{c}}^t\|_{\mathsf{F}} + \beta^t \overline{\mathrm{A}} \|\mathcal{A}(\mathbf{X}_{\mathsf{c}}^t) - \mathbf{y}^t\| + \theta\ell(\beta^t)\|\mathbf{X}^{t+1} - \mathbf{X}_{\mathsf{c}}^t\|_{\mathsf{F}} \\
&\overset{\text{③}}{\leq} L_f \|\mathbf{X}^t - \mathbf{X}^{t-1}\|_{\mathsf{F}} + \beta^t \overline{\mathrm{A}}\{\|\mathcal{A}(\mathbf{X}^t) - \mathbf{y}^t\| + \overline{\mathrm{A}}\|\mathbf{X}^t - \mathbf{X}^{t-1}\|_{\mathsf{F}}\} \\
&\quad + \theta\ell(\beta^t)(\|\mathbf{X}^{t+1} - \mathbf{X}^t\|_{\mathsf{F}} + \|\mathbf{X}^t - \mathbf{X}^{t-1}\|_{\mathsf{F}}) \\
&\overset{\text{④}}{\leq} (L_f + \beta^t \overline{\mathrm{A}}^2 + \theta\ell(\beta^t))\|\mathbf{X}^t - \mathbf{X}^{t-1}\|_{\mathsf{F}} + \beta^t \overline{\mathrm{A}}\|\mathcal{A}(\mathbf{X}^t) - \mathbf{y}^t\| + \theta\ell(\beta^t)\|\mathbf{X}^{t+1} - \mathbf{X}^t\|_{\mathsf{F}} \\
&\overset{\text{⑤}}{=} \mathcal{O}(\beta^{t-1} e^t) + \mathcal{O}(\beta^t e^{t+1}),
\end{aligned} \tag{40}$$

where step ① uses the definitions of $\{\ddot{\mathbf{G}}, \dot{\mathbf{G}}\}$; step ② uses the triangle inequality; step ③ uses the fact that $f(\mathbf{X})$ is $L_f$-smooth, $\|\mathbf{X}^t - \mathbf{X}_{\mathsf{c}}^t\|_{\mathsf{F}} \leq \|\mathbf{X}^t - \mathbf{X}^{t-1}\|_{\mathsf{F}}$, $\|\mathbf{X}^{t+1} - \mathbf{X}_{\mathsf{c}}^t\|_{\mathsf{F}} \leq \|\mathbf{X}^{t+1} - \mathbf{X}^t\|_{\mathsf{F}} + \|\mathbf{X}^t - \mathbf{X}^{t-1}\|_{\mathsf{F}}$, and $\|\mathcal{A}(\mathbf{X}_{\mathsf{c}}^t) - \mathbf{y}^t\| \leq \|\mathcal{A}(\mathbf{X}^t) - \mathbf{y}^t\|_{\mathsf{F}} + \overline{\mathrm{A}}\|\mathbf{X}^t - \mathbf{X}^{t-1}\|_{\mathsf{F}}$, as shown in Lemma A.6.

We derive the following inequalities:

$$\begin{aligned}
&\|\mathrm{Proj}_{\mathbf{T}_{\mathbf{X}^t}\mathcal{M}}(\dot{\mathbf{G}})\|_{\mathsf{F}} \\
&\overset{\text{①}}{=} \|\mathrm{Proj}_{\mathbf{T}_{\mathbf{X}^t}\mathcal{M}}(\dot{\mathbf{G}}) + \mathrm{Proj}_{\mathbf{T}_{\mathbf{X}^{t+1}}\mathcal{M}}(\ddot{\mathbf{G}})\|_{\mathsf{F}} \\
&\overset{\text{②}}{\leq} 2\|\dot{\mathbf{G}} - \ddot{\mathbf{G}}\|_{\mathsf{F}} + 2\sqrt{r}\|\dot{\mathbf{G}}\| \cdot \|\mathbf{X}^{t+1} - \mathbf{X}^t\|_{\mathsf{F}} \\
&\overset{\text{③}}{\leq} \mathcal{O}(\beta^{t-1} e^t) + \mathcal{O}(\beta^t e^{t+1}) + 2\sqrt{r}(C_f + C_g + \overline{\mathrm{A}}\overline{\mathrm{z}})\|\mathbf{X}^{t+1} - \mathbf{X}^t\|_{\mathsf{F}} \\
&= \mathcal{O}(\beta^{t-1} e^t) + \mathcal{O}(\beta^t e^{t+1}),
\end{aligned}$$

where step ① uses the optimality of $\mathbf{X}^{t+1}$ that:

$$\mathbf{0} = \mathrm{Proj}_{\mathbf{T}_{\mathbf{X}^{t+1}}\mathcal{M}}(\ddot{\mathbf{G}});$$

step ② uses the result of Lemma A.7 by applying

$$\mathbf{X} = \mathbf{X}^t, \ \tilde{\mathbf{X}} = \mathbf{X}^{t+1}, \ \mathbf{P} = \dot{\mathbf{G}}, \ \text{and} \ \tilde{\mathbf{P}} = \ddot{\mathbf{G}};$$

step ③ uses Inequality (40), and the fact that $\|\dot{\mathbf{G}}\| = \|\nabla f(\mathbf{X}^t) - \partial g(\mathbf{X}^t) + \mathcal{A}^{\mathsf{T}}(\mathbf{z}^t))\| \leq \|\nabla f(\mathbf{X}^t) - \partial g(\mathbf{X}^t) + \mathcal{A}^{\mathsf{T}}(\mathbf{z}^t)\|_{\mathsf{F}} \leq C_f + C_g + \overline{\mathrm{A}}\overline{z}$.

Finally, we derive:

$$
\begin{aligned}
&\tfrac{1}{T}\textstyle\sum_{t=1}^{T} \mathrm{Crit}(\mathbf{X}^t, \breve{\mathbf{y}}^t, \mathbf{z}^t) \\
&\overset{①}{=} \tfrac{1}{T}\textstyle\sum_{t=1}^{T}\{\|\mathcal{A}(\mathbf{X}^t) - \breve{\mathbf{y}}^t\| + \|\partial h(\breve{\mathbf{y}}^t) - \mathbf{z}^t\| + \|\mathrm{Proj}_{\mathbf{T}_{\mathbf{X}^t}\mathcal{M}}(\dot{\mathbf{G}})\|_{\mathsf{F}}\} \\
&\overset{②}{\leq} \tfrac{1}{T}\textstyle\sum_{t=1}^{T}\{\|\mathcal{A}(\mathbf{X}^t) - \mathbf{y}^t\| + \|(1 - \tfrac{1}{\sigma})(\mathbf{z}^t - \mathbf{z}^{t-1})\| + \|\mathrm{Proj}_{\mathbf{T}_{\mathbf{X}^t}\mathcal{M}}(\dot{\mathbf{G}})\|_{\mathsf{F}}\} \\
&\overset{③}{=} \tfrac{1}{T}\textstyle\sum_{t=1}^{T}\{\mathcal{O}(\beta^{t-1}e^t) + \mathcal{O}(\beta^t e^{t+1})\} \\
&\overset{④}{=} \mathcal{O}(T^{(p-1)/2}) = \mathcal{O}(T^{-1/3}),
\end{aligned}
$$

where step ① uses the definition of $\mathrm{Crit}(\mathbf{X}, \mathbf{y}, \mathbf{z})$; step ② uses $\mathbf{z}^{t+1} - \partial h(\breve{\mathbf{y}}^{t+1}) \ni (1 - \tfrac{1}{\sigma})(\mathbf{z}^{t+1} - \mathbf{z}^t)$, as shown in Lemma 4.1; step ③ uses $\|\mathbf{z}^t - \mathbf{z}^{t-1}\| = \|\sigma\beta^{t-1}(\mathcal{A}(\mathbf{X}^t) - \mathbf{y}^t)\| \leq 2\beta^t\|\mathcal{A}(\mathbf{X}^t) - \mathbf{y}^t\| = \mathcal{O}(\beta^{t-1}e^t)$; step ④ uses the choice $p = 1/3$ and Lemma 4.7(**b**).

$\square$

## C.8 PROOF OF LEMMA 4.10

*Proof.* We define $\mathcal{S}(\mathbf{X}, \mathbf{y}^t; \mathbf{z}^t; \beta^t) \triangleq f(\mathbf{X}) + \langle \mathbf{z}^t, \mathcal{A}(\mathbf{X}) - \mathbf{y}^t\rangle + \tfrac{\beta^t}{2}\|\mathcal{A}(\mathbf{X}) - \mathbf{y}^t\|_2^2$.

We let $\mathbf{G}^t \in \nabla_{\mathbf{X}}\mathcal{S}(\mathbf{X}^t, \mathbf{y}^t; \mathbf{z}^t; \beta^t) - \partial g(\mathbf{X}^t)$.

We define $\eta^t \triangleq \tfrac{b^t \gamma^j}{\beta^t} \in (0, \infty)$.

**Part (a).** Initially, we show that $\|\mathbf{G}^t\|_{\mathsf{F}}$ is always bounded for $t$ with $\mathbf{X} \in \mathcal{M}$. We have:

$$
\begin{aligned}
\|\mathbf{G}^t\|_{\mathsf{F}} &= \|\nabla f(\mathbf{X}^t) - \partial g(\mathbf{X}^t) + \mathcal{A}^{\mathsf{T}}[\mathbf{z}^t + \beta^t(\mathcal{A}(\mathbf{X}^t) - \mathbf{y}^t)]\|_{\mathsf{F}} \\
&\overset{①}{=} \|\nabla f(\mathbf{X}^t) - \partial g(\mathbf{X}^t) + \mathcal{A}^{\mathsf{T}}[\mathbf{z}^t + \tfrac{\beta^t}{\sigma\beta^{t-1}}(\mathbf{z}^t - \mathbf{z}^{t-1})]\|_{\mathsf{F}} \\
&\overset{②}{\leq} \|\nabla f(\mathbf{X}^t)\|_{\mathsf{F}} + \|\partial g(\mathbf{X}^t)\|_{\mathsf{F}} + \overline{\mathrm{A}} \cdot \{\|\mathbf{z}^t\| + \tfrac{\beta^t}{\sigma\beta^{t-1}}(\|\mathbf{z}^t\| + \|\mathbf{z}^{t-1}\|)\} \\
&\overset{③}{\leq} C_f + C_g + \overline{\mathrm{A}} \cdot (\overline{z} + 2(1 + \xi)\overline{z}) \triangleq \overline{g},
\end{aligned}
$$

where step ① uses $\mathbf{z}^{t+1} = \mathbf{z}^t + \sigma\beta^t(\mathcal{A}(\mathbf{X}^{t+1}) - \mathbf{y}^{t+1})$; step ② uses the triangle inequality; step ③ uses $\|\nabla f(\mathbf{X}^t)\|_{\mathsf{F}} \leq C_f$, $\|\nabla g(\mathbf{X}^t)\|_{\mathsf{F}} \leq C_g$, $\|\nabla \mathcal{A}(\mathbf{X}^t)\|_{\mathsf{F}} \leq \|\nabla\mathcal{A}(\mathbf{X}^t)\| \leq \overline{\mathrm{A}}$, $\|\mathbf{z}^t\| \leq \overline{z}$, $\tfrac{1}{\sigma} \leq 1$, $\beta^t \leq \beta^{t-1}(1 + \xi)$; step ④ uses $\xi \leq 1$.

We derive the following inequalities:

$$L(\mathbf{X}^{t+1}, \mathbf{y}^t; \mathbf{z}^t; \beta^t, \mu^t) - L(\mathbf{X}^t, \mathbf{y}^t; \mathbf{z}^t; \beta^t, \mu^t) = \dot{\mathcal{L}}(\mathbf{X}^{t+1}) - \dot{\mathcal{L}}(\mathbf{X}^t)$$

$$\overset{①}{=} \{\mathcal{S}^t(\mathbf{X}^{t+1}, \mathbf{y}^t; \mathbf{z}^t; \beta^t) - g(\mathbf{X}^{t+1})\} - \{\mathcal{S}^t(\mathbf{X}^t, \mathbf{y}^t; \mathbf{z}^t; \beta^t) - g(\mathbf{X}^t)\}$$

$$\overset{②}{\leq} \tfrac{1}{2}\ell(\beta^t)\|\mathbf{X}^{t+1} - \mathbf{X}^t\|_{\mathsf{F}}^2 + \langle \mathbf{G}^t, \mathbf{X}^{t+1} - \mathbf{X}^t \rangle$$

$$\overset{③}{=} \tfrac{1}{2}\ell(\beta^t)\| \mathrm{Retr}_{\mathbf{X}^t}(-\eta^t \mathbb{G}_\rho^t) - \mathbf{X}^t\|_{\mathsf{F}}^2 + \langle \mathbf{G}^t, \mathrm{Retr}_{\mathbf{X}^t}(-\eta^t \mathbb{G}_\rho^t) - \mathbf{X}^t + \eta^t \mathbb{G}_\rho^t \rangle - \eta^t \langle \mathbf{G}^t, \mathbb{G}_\rho^t \rangle$$

$$\overset{④}{\leq} \tfrac{1}{2}\ell(\beta^t)\| \mathrm{Retr}_{\mathbf{X}^t}(-\eta^t \mathbb{G}_\rho^t) - \mathbf{X}^t\|_{\mathsf{F}}^2 + \overline{g}\| \mathrm{Retr}_{\mathbf{X}^t}(-\eta^t \mathbb{G}_\rho^t) - \mathbf{X}^t + \eta^t \mathbb{G}_\rho^t\|_{\mathsf{F}} - \tfrac{\eta^t}{\max(1,2\rho)}\|\mathbb{G}_\rho^t\|_{\mathsf{F}}^2$$

$$\overset{⑤}{\leq} \tfrac{1}{2}\ell(\beta^t)\dot{k}\|\eta^t \mathbb{G}_\rho^t\|_{\mathsf{F}}^2 + \tfrac{1}{2}\overline{g}\ddot{k}\|\eta^t \mathbb{G}_\rho^t\|_{\mathsf{F}}^2 - \tfrac{\eta^t}{\max(1,2\rho)}\|\mathbb{G}_\rho^t\|_{\mathsf{F}}^2$$

$$\overset{⑥}{=} \eta^t\|\mathbb{G}_\rho^t\|_{\mathsf{F}}^2 \cdot \{\tfrac{1}{2}\ell(\beta^t)\dot{k}\tfrac{b^t \gamma^j}{\beta^t} + \tfrac{1}{2}\overline{g}\ddot{k}\tfrac{b^t \gamma^j}{\beta^t} - \tfrac{1}{\max(1,2\rho)}\}$$

$$\overset{⑦}{\leq} \eta^t\|\mathbb{G}_\rho^t\|_{\mathsf{F}}^2 \cdot \{(\tfrac{\overline{b}}{2}\dot{k}\overline{\ell} + \tfrac{\overline{b}}{2\beta^0}\ddot{k}\overline{g})\gamma^j - \tfrac{1}{\max(1,2\rho)}\}$$

$$\overset{⑧}{\leq} \eta^t\|\mathbb{G}_\rho^t\|_{\mathsf{F}}^2 \cdot \{-\delta\}, \tag{41}$$

where step ① uses the definitions of $L(\mathbf{X}, \mathbf{y}; \mathbf{z}; \beta, \mu)$; step ② uses the fact that the function $g(\mathbf{X})$ is convex and the function $\mathcal{S}(\mathbf{X}, \mathbf{y}^t; \mathbf{z}^t; \beta^t)$ is $\ell(\beta^t)$-smooth *w.r.t.* $\mathbf{X}$; step ③ uses $\mathbf{X}^{t+1} = \mathrm{Retr}_{\mathbf{X}^t}(-\eta^t \mathbb{G}_\rho^t)$; step ④ uses the Cauchy-Schwarz Inequality, $\|\mathbf{G}^t\|_{\mathsf{F}} \leq \overline{g}$, and Lemma 2.12(*a*) that $\langle \mathbf{G}^t, \mathbb{G}_\rho^t \rangle \geq \tfrac{1}{\max(1,2\rho)}\|\mathbb{G}_\rho^t\|_{\mathsf{F}}^2$; step ⑤ uses Lemma 2.10 with $\mathbf{\Delta} \triangleq -\eta^t \mathbb{G}_\rho^t$ given that $\mathbf{X}^t \in \mathcal{M}$ and $\mathbf{\Delta} \in \mathbf{T}_{\mathbf{X}^t}\mathcal{M}$; step ⑥ uses $\eta^t \triangleq \tfrac{b^t \gamma^j}{\beta^t}$; step ⑦ uses $\ell(\beta^t) \leq \beta^t \overline{\ell}$, $\beta^0 \leq \beta^t$, and $b^t \leq \overline{b}$; step ⑧ uses the fact that $\gamma^j$ is sufficiently small such that:

$$\gamma^j \leq \frac{2(\frac{1}{\max(1,2\rho)} - \delta)}{\overline{\ell}\dot{k}\overline{b} + \overline{g}\ddot{k}\overline{b}/\beta^0} \triangleq \overline{\gamma}. \tag{42}$$

Given Inequality (41) coincides with the condition of the line search procedure, we complete the proof.

**Part (b)**. We derive the following inequalities:

$$L(\mathbf{X}^{t+1}, \mathbf{y}^t; \mathbf{z}^t; \beta^t, \mu^t) - L(\mathbf{X}^t, \mathbf{y}^t; \mathbf{z}^t; \beta^t, \mu^t)$$

$$\overset{①}{\leq} -\|\mathbb{G}_\rho^t\|_{\mathsf{F}}^2 \delta \eta^t$$

$$\overset{②}{\leq} -\|\mathbb{G}_{1/2}^t\|_{\mathsf{F}}^2 \delta \eta^t \cdot \min(1, 2\rho)^2$$

$$\overset{③}{=} -\tfrac{1}{\beta^t}\|\mathbb{G}_{1/2}^t\|_{\mathsf{F}}^2 \cdot \delta b^t \gamma^{j-1}\gamma \cdot \min(1, 2\rho)^2$$

$$\overset{④}{\leq} -\tfrac{1}{\beta^t}\|\mathbb{G}_{1/2}^t\|_{\mathsf{F}}^2 \cdot \underbrace{\delta \underline{b}\overline{\gamma}\gamma \cdot \min(1, 2\rho)^2}_{\triangleq \varepsilon_x},$$

where step ① uses Inequality (41); step ② uses Lemma 2.12(*b*) that $\|\mathbb{G}_\rho\|_{\mathsf{F}} \geq \min(1, 2\rho)\|\mathbb{G}_{1/2}\|_{\mathsf{F}}$; step ③ uses the definition $\eta^t \triangleq \tfrac{b^t \gamma^j}{\beta^t}$; step ④ uses $b^t \geq \underline{b}$, and the following inequality:

$$\gamma^{j-1} \geq \overline{\gamma} \geq \gamma^j,$$

which can be implied by the stopping criteria of the line search procedure.

$\square$

## C.9    PROOF OF LEMMA 4.12

*Proof.* We define: $\Theta^t \triangleq L(\mathbf{X}^t, \mathbf{y}^t; \mathbf{z}^t; \beta^t, \mu^{t-1}) + \mu^{t-1}C_h^2 + \mathbb{T}^t + \mathbb{Z}^t + 0 \times \mathbb{X}^t,$

We define $\tilde{e}_t \triangleq \|\mathbf{y}^t - \mathbf{y}^{t-1}\|^2 + \|\mathcal{A}(\mathbf{X}^t) - \mathbf{y}^t\|^2 + \|\tfrac{1}{\beta^t}\mathbb{G}_{1/2}^t\|_{\mathsf{F}}^2.$

**Part (a)**. Using Lemma 4.5, we have:
$$L(\mathbf{X}^{t+1}, \mathbf{y}^{t+1}; \mathbf{z}^{t+1}; \beta^{t+1}, \mu^t) - L(\mathbf{X}^{t+1}, \mathbf{y}^t; \mathbf{z}^t; \beta^t, \mu^{t-1}) - (\mu^{t-1} - \mu^t)C_h^2$$
$$\leq \quad \mathbb{T}^t - \mathbb{T}^{t+1} + \mathbb{Z}^t - \mathbb{Z}^{t+1} - \varepsilon_y \beta^t \|\mathbf{y}^{t+1} - \mathbf{y}^t\|_2^2 - \varepsilon_z \beta^t \|\mathcal{A}(\mathbf{X}^{t+1}) - \mathbf{y}^{t+1}\|_2^2. \quad (43)$$

Using Lemma 4.10, we have:
$$L(\mathbf{X}^{t+1}, \mathbf{y}^t, \mathbf{z}^t; \beta^t, \mu^{t-1}) - L(\mathbf{X}^t, \mathbf{y}^t, \mathbf{z}^t; \beta^t, \mu^{t-1}) \leq 0 \times \mathbb{X}^t - 0 \times \mathbb{X}^{t+1} - \varepsilon_x \beta^t \|\tfrac{1}{\beta^t}\mathbb{G}_{1/2}^t\|_{\mathsf{F}}^2.$$

Adding these two inequalities together and using the definition of $\Theta^t$, we have:
$$\Theta^t - \Theta^{t+1} \quad \geq \quad \varepsilon_y \beta^t \|\mathbf{y}^{t+1} - \mathbf{y}^t\|_2^2 + \varepsilon_x \beta^t \|\tfrac{1}{\beta^t}\mathbb{G}_{1/2}^t\|_{\mathsf{F}}^2 + \varepsilon_z \beta^t \|\mathcal{A}(\mathbf{X}^{t+1}) - \mathbf{y}^{t+1}\|_2^2$$
$$\geq \quad \min(\varepsilon_y, \varepsilon_x, \varepsilon_z) \cdot \beta^t \cdot \tilde{e}_{t+1}.$$

**Part (b)**. Using the same strategy as in deriving Lemma 4.7(**b**), we finish the proof.

$\square$

## C.10 PROOF OF THEOREM 4.13

*Proof.* We define $\mathrm{Crit}(\mathbf{X}, \mathbf{y}, \mathbf{z}) \triangleq \|\mathcal{A}(\mathbf{X}) - \mathbf{y}\| + \|\partial h(\mathbf{y}) - \mathbf{z}\| + \|\mathrm{Proj}_{\mathbf{T}_\mathbf{X}\mathcal{M}}(\nabla f(\mathbf{X}) - \partial g(\mathbf{X}) + \mathcal{A}^\mathsf{T}(\mathbf{z}))\|_{\mathsf{F}}$.

We define $\dot{\mathbf{G}} \triangleq \nabla f(\mathbf{X}^t) - \partial g(\mathbf{X}^t) + \mathcal{A}^\mathsf{T}(\mathbf{z}^t)$, and $\ddot{\mathbf{G}} \triangleq \beta^t \mathcal{A}^\mathsf{T}(\mathcal{A}(\mathbf{X}^t) - \mathbf{y}^t)$.

We let $\mathbf{G} = \mathbf{G}^t \in \partial_\mathbf{X} L(\mathbf{X}^t, \mathbf{y}^t; \mathbf{z}^t; \beta^t, \mu^t)$.

First, we obtain:
$$\mathbb{G}_{1/2}^t \quad \overset{①}{=} \quad \mathbf{G} - \tfrac{1}{2}\mathbf{X}^t\mathbf{G}^\mathsf{T}\mathbf{X}^t - \tfrac{1}{2}\mathbf{X}^t[\mathbf{X}^t]^\mathsf{T}\mathbf{G}$$
$$\overset{②}{=} \quad (\dot{\mathbf{G}} - \tfrac{1}{2}\mathbf{X}^t\dot{\mathbf{G}}^\mathsf{T}\mathbf{X}^t - \tfrac{1}{2}\mathbf{X}^t[\mathbf{X}^t]^\mathsf{T}\dot{\mathbf{G}}) + (\ddot{\mathbf{G}} - \tfrac{1}{2}\mathbf{X}^t\ddot{\mathbf{G}}^\mathsf{T}\mathbf{X}^t - \tfrac{1}{2}\mathbf{X}^t[\mathbf{X}^t]^\mathsf{T}\ddot{\mathbf{G}})$$
$$\overset{③}{=} \quad \mathrm{Proj}_{\mathbf{T}_{\mathbf{X}^t}\mathcal{M}}(\dot{\mathbf{G}}) + \mathrm{Proj}_{\mathbf{T}_{\mathbf{X}^t}\mathcal{M}}(\ddot{\mathbf{G}})$$

where step ① uses the definition $\mathbb{G}_\rho^t \triangleq \mathbf{G} - \rho\mathbf{X}^t\mathbf{G}^\mathsf{T}\mathbf{X}^t - (1-\rho)\mathbf{X}^t[\mathbf{X}^t]^\mathsf{T}\mathbf{G}$, as shown in Algorithm 1; step ② uses $\mathbf{G} \in \dot{\mathbf{G}} + \ddot{\mathbf{G}}$; step ③ uses the fact that $\mathrm{Proj}_{\mathbf{T}_\mathbf{X}\mathcal{M}}(\boldsymbol{\Delta}) = \boldsymbol{\Delta} - \tfrac{1}{2}\mathbf{X}(\boldsymbol{\Delta}^\mathsf{T}\mathbf{X} + \mathbf{X}^\mathsf{T}\boldsymbol{\Delta})$ for all $\boldsymbol{\Delta} \in \mathbb{R}^{n \times r}$ (Absil et al., 2008a). This leads to:
$$\|\mathrm{Proj}_{\mathbf{T}_{\mathbf{X}^t}\mathcal{M}}(\dot{\mathbf{G}})\|_{\mathsf{F}} \quad = \quad \|\mathbb{G}_{1/2}^t - \mathrm{Proj}_{\mathbf{T}_{\mathbf{X}^t}\mathcal{M}}(\ddot{\mathbf{G}})\|_{\mathsf{F}}$$
$$\overset{①}{\leq} \quad \|\mathbb{G}_{1/2}^t\|_{\mathsf{F}} + \|\mathrm{Proj}_{\mathbf{T}_{\mathbf{X}^t}\mathcal{M}}(\ddot{\mathbf{G}})\|_{\mathsf{F}}$$
$$\overset{②}{\leq} \quad \|\mathbb{G}_{1/2}^t\|_{\mathsf{F}} + \|\ddot{\mathbf{G}}\|_{\mathsf{F}}$$
$$\leq \quad \|\mathbb{G}_{1/2}^t\|_{\mathsf{F}} + \beta^t\overline{\mathrm{A}}\|\mathcal{A}(\mathbf{X}^t) - \mathbf{y}^t\|$$
$$\leq \quad \beta^t e^{t+1} + \mathcal{O}(\beta^{t-1}e^t),$$

where step ① uses the triangle inequality; step ② uses Lemma 2.11 that $\|\mathrm{Proj}_{\mathbf{T}_\mathbf{X}\mathcal{M}}(\boldsymbol{\Delta})\|_{\mathsf{F}} \leq \|\boldsymbol{\Delta}\|_{\mathsf{F}}$ for all $\boldsymbol{\Delta} \in \mathbb{R}^{n \times r}$.

Finally, we derive:
$$\tfrac{1}{T}\sum_{t=1}^T \mathrm{Crit}(\mathbf{X}^t, \check{\mathbf{y}}^t, \mathbf{z}^t)$$
$$\overset{①}{=} \quad \tfrac{1}{T}\sum_{t=1}^T\{\|\mathcal{A}(\mathbf{X}^t) - \check{\mathbf{y}}^t\| + \|\partial h(\check{\mathbf{y}}^t) - \mathbf{z}^t\| + \|\mathrm{Proj}_{\mathbf{T}_{\mathbf{X}^t}\mathcal{M}}(\dot{\mathbf{G}})\|_{\mathsf{F}}\}$$
$$\overset{②}{\leq} \quad \tfrac{1}{T}\sum_{t=1}^T\{\|\mathcal{A}(\mathbf{X}^t) - \mathbf{y}^t\| + \|(1 - \tfrac{1}{\sigma})(\mathbf{z}^t - \mathbf{z}^{t-1})\| + \|\mathrm{Proj}_{\mathbf{T}_{\mathbf{X}^t}\mathcal{M}}(\dot{\mathbf{G}})\|_{\mathsf{F}}\}$$
$$\overset{③}{=} \quad \tfrac{1}{T}\sum_{t=1}^T\{\mathcal{O}(\beta^t e^{t+1}) + \mathcal{O}(\beta^{t-1}e^t)\}$$
$$\overset{④}{=} \quad \mathcal{O}(T^{(p-1)/2}) = \mathcal{O}(T^{-1/3}),$$

where step ① uses the definition of $\mathrm{Crit}(\mathbf{X}, \mathbf{y}, \mathbf{z})$; step ② uses $\mathbf{z}^{t+1} - \partial h(\check{\mathbf{y}}^{t+1}) \ni (1 - \tfrac{1}{\sigma})(\mathbf{z}^{t+1} - \mathbf{z}^t)$, as shown in Lemma 4.1; step ③ uses $\|\mathbf{z}^t - \mathbf{z}^{t-1}\| = \|\sigma\beta^{t-1}(\mathcal{A}(\mathbf{X}^t) - \mathbf{y}^t)\| \leq 2\beta^t\|\mathcal{A}(\mathbf{X}^t) - \mathbf{y}^t\| = \mathcal{O}(\beta^{t-1}e^t)$; step ④ uses the choice $p = 1/3$ and Lemma 4.7(**b**).

$\square$

## D  PROOFS FOR SECTION 5

### D.1  PROOF OF LEMMA 5.4

We begin by presenting the following four useful lemmas.

**Lemma D.1.** *For both* OADMM-EP *and* OADMM-RR*, we have:*

$$(\mathbf{d_X}, \mathbf{d_{X^-}}, \mathbf{d_y}, \mathbf{d_z}) \in \partial\Theta(\mathbf{X}^t, \mathbf{X}^{t-1}, \mathbf{y}^t, \mathbf{z}^t; \beta^t, \beta^{t-1}, \mu^{t-1}, t), \tag{44}$$

*where* $\mathbf{d_X} \triangleq \mathbb{A}^t + \{\beta^t + 2\omega\ddot{\sigma}\sigma^2\beta^{t-1}\} \cdot \mathcal{A}^\mathsf{T}(\mathcal{A}(\mathbf{X}^t) - \mathbf{y}^t) + \alpha(\theta+1)\ell(\beta^t)(\mathbf{X}^t - \mathbf{X}^{t-1})$, $\mathbf{d_{X^-}} \triangleq \alpha(\theta + 1)\ell(\beta^t)(\mathbf{X}^{t-1} - \mathbf{X}^t)$, $\mathbf{d_y} \triangleq \nabla h_{\mu^{t-1}}(\mathbf{y}^t) - \mathbf{z}^t + (\mathbf{y}^t - \mathcal{A}(\mathbf{X}^t)) \cdot (\beta^t + 2\omega\ddot{\sigma}\sigma^2\beta^{t-1})$, $\mathbf{d_z} \triangleq \mathcal{A}(\mathbf{X}^t) - \mathbf{y}^t$. *Here,* $\mathbb{A}^t \triangleq \partial I_{\mathcal{M}}(\mathbf{X}^t) + \nabla f(\mathbf{X}^t) - \nabla g(\mathbf{X}^t) + \mathcal{A}^\mathsf{T}(\mathbf{z}^t)$.

*Proof.* We define the Lyapunov function as: $\Theta(\mathbf{X}, \mathbf{X}^-, \mathbf{y}, \mathbf{z}; \beta, \beta^-, \mu^-, t) \triangleq L(\mathbf{X}, \mathbf{y}; \mathbf{z}; \beta, \mu^-) + \omega\ddot{\sigma}\sigma^2\beta^- \|\mathcal{A}(\mathbf{X}) - \mathbf{y}\|_2^2 + \frac{\alpha(\theta+1)\ell(\beta)}{2}\|\mathbf{X} - \mathbf{X}^-\|_F^2 + \frac{4\omega\ddot{\sigma}}{\beta^0}C_h^2\frac{1}{t} + C_h^2\mu^-$.

Using this definition, we can promptly derive the conclusion of the lemma.

□

**Lemma D.2.** *For* OADMM-EP*, we define* $\{\mathbf{d_X}, \mathbf{d_{X^-}}, \mathbf{d_y}, \mathbf{d_z}\}$ *as in Lemma D.1. There exists a constant* $K$ *such that:*

$$\frac{1}{\beta^t}\{\|\mathbf{d_X}\|_F + \|\mathbf{d_{X^-}}\|_F + \|\mathbf{d_y}\| + \|\mathbf{d_z}\|\} \le K\{\mathcal{X}^t + \mathcal{Z}^t + \mathcal{X}^{t-1} + \mathcal{Z}^{t-1}\}. \tag{45}$$

*Here,* $\mathcal{X}^t \triangleq \|\mathbf{X}^t - \mathbf{X}^{t-1}\|_F$, *and* $\mathcal{Z}^t \triangleq \|\mathcal{A}(\mathbf{X}^t) - \mathbf{y}^t\|$.

*Proof.* First, we obtain:

$$\frac{1}{\beta^t}\|\mathbb{A}^t\|_F = \|\partial I_{\mathcal{M}}(\mathbf{X}^t) + \nabla f(\mathbf{X}^t) - \nabla g(\mathbf{X}^t) + \mathcal{A}^\mathsf{T}(\mathbf{z}^t)\|_F$$

$$\overset{①}{=} \frac{1}{\beta^t}\|\nabla g(\mathbf{X}^{t-1}) - \nabla g(\mathbf{X}^t) + \nabla f(\mathbf{X}^t) - \nabla f(\mathbf{X}_\mathsf{c}^{t-1}) - \theta\ell(\beta^{t-1})(\mathbf{X}^t - \mathbf{X}_\mathsf{c}^{t-1})$$

$$\qquad + \mathcal{A}^\mathsf{T}(\mathbf{z}^t - \mathbf{z}^{t-1}) - \beta^{t-1}\mathcal{A}^\mathsf{T}(\mathcal{A}(\mathbf{X}_\mathsf{c}^{t-1}) - \mathbf{y}^{t-1}))\|_F$$

$$\overset{②}{\le} \frac{1}{\beta^t}L_g\|\mathbf{X}^t - \mathbf{X}^{t-1}\|_F + \frac{1}{\beta^t}(L_f + \theta\ell(\beta^{t-1}))\|\mathbf{X}^t - \mathbf{X}_\mathsf{c}^{t-1}\|_F$$

$$\qquad + \frac{1}{\beta^t}\overline{A}\|\mathbf{z}^t - \mathbf{z}^{t-1}\| + \frac{1}{\beta^t}\beta^{t-1}\overline{A}\{\|\mathcal{A}(\mathbf{X}^{t-1}) - \mathbf{y}^{t-1}\| + \overline{A}\|\mathbf{X}^{t-1} - \mathbf{X}^{t-2}\|_F\}$$

$$= \mathcal{O}(\|\mathbf{X}^t - \mathbf{X}^{t-1}\|_F) + \mathcal{O}(\|\mathcal{A}(\mathbf{X}^t) - \mathbf{y}^t\|)$$

$$\qquad + \mathcal{O}(\|\mathbf{X}^{t-1} - \mathbf{X}^{t-2}\|_F) + \mathcal{O}(\|\mathcal{A}(\mathbf{X}^{t-1}) - \mathbf{y}^{t-1}\|), \tag{46}$$

where step ① uses the optimality of $\mathbf{X}^{t+1}$ for OADMM-EP that:

$$\partial I_{\mathcal{M}}(\mathbf{X}^{t+1}) - \nabla g(\mathbf{X}^t)$$

$$\ni -\theta\ell(\beta^t)(\mathbf{X}^{t+1} - \mathbf{X}_\mathsf{c}^t) - \nabla_\mathbf{X}\mathcal{S}(\mathbf{X}_\mathsf{c}^t, \mathbf{y}^t; \mathbf{z}^t; \beta^t)$$

$$= -\theta\ell(\beta^t)(\mathbf{X}^{t+1} - \mathbf{X}_\mathsf{c}^t) - \nabla f(\mathbf{X}_\mathsf{c}^t) - \mathcal{A}^\mathsf{T}[\mathbf{z}^t + \beta^t(\mathcal{A}(\mathbf{X}_\mathsf{c}^t) - \mathbf{y}^t)]; \tag{47}$$

step ② uses the triangle inequality, the $L_f$-Lipschitz continuity of $\nabla f(\mathbf{X})$ for all $\mathbf{X}$; the $L_g$-Lipschitz continuity of $\nabla g(\mathbf{X})$, and the upper bound of $\|\mathcal{A}(\mathbf{X}_\mathsf{c}^t) - \mathbf{y}^t\|$ as shown in Lemma A.6(*c*); step ③ uses the upper bound of $\|\mathbf{X}^t - \mathbf{X}_\mathsf{c}^{t-1}\|_F$, and $\mathbf{z}^t - \mathbf{z}^{t-1} = \sigma\beta^{t-1}(\mathcal{A}(\mathbf{X}^t) - \mathbf{y}^t)$.

**Part (a).** We bound the term $\frac{1}{\beta^t}\|\mathbf{d_X}\|_F$. We have:

$$\frac{1}{\beta^t}\|\mathbf{d_X}\|_F$$

$$\overset{①}{=} \frac{1}{\beta^t}\|\mathbb{A}^t + (\beta^t + 2\omega\ddot{\sigma}\sigma^2\beta^{t-1})\mathcal{A}^\mathsf{T}(\mathcal{A}(\mathbf{X}^t) - \mathbf{y}^t) + \alpha(\theta+1)\ell(\beta^t)(\mathbf{X}^t - \mathbf{X}^{t-1})\|_F$$

$$\overset{②}{\le} \frac{1}{\beta^t}\|\mathbb{A}^t\|_F + (1 + 2\omega\ddot{\sigma}\sigma^2)\overline{A}\|\mathcal{A}(\mathbf{X}^t) - \mathbf{y}^t\|_F + \alpha(\theta+1)\overline{\ell}\|\mathbf{X}^t - \mathbf{X}^{t-1}\|_F$$

$$\overset{③}{\le} \mathcal{O}(\|\mathbf{X}^t - \mathbf{X}^{t-1}\|_F) + \mathcal{O}(\|\mathcal{A}(\mathbf{X}^t) - \mathbf{y}^t\|) + \mathcal{O}(\|\mathbf{X}^{t-1} - \mathbf{X}^{t-2}\|_F) + \mathcal{O}(\|\mathcal{A}(\mathbf{X}^{t-1}) - \mathbf{y}^{t-1}\|),$$

where step ① uses the definition of $\mathbf{d_X}$ in Lemma D.1; step ② uses the triangle inequality, $\beta^{t-1} \leq \beta^t$, and $\ell(\beta^t) \leq \beta^t \bar{\ell}$; step ③ uses Inequality (46).

**Part (b)**. We bound the term $\frac{1}{\beta^t}\|\mathbf{d_{X^-}}\|_\mathsf{F}$. We have:

$$\frac{1}{\beta^t}\|\mathbf{d_{X^-}}\|_\mathsf{F} \overset{①}{=} \frac{1}{\beta^t}\alpha(\theta+1)\ell(\beta^t)\|\mathbf{X}^{t-1} - \mathbf{X}^t\|_\mathsf{F} \overset{②}{=} \mathcal{O}(\|\mathbf{X}^t - \mathbf{X}^{t-1}\|_\mathsf{F}), \tag{48}$$

where step ① uses the definition of $\mathbf{d_{X^-}}$ in Lemma D.1; step ② uses $\ell(\beta^t) \leq \beta^t \bar{\ell}$.

**Part (c)**. We bound the term $\frac{1}{\beta^t}\|\mathbf{d_y}\|_\mathsf{F}$. We have:

$$\begin{aligned}
\frac{1}{\beta^t}\|\mathbf{d_y}\| &\overset{①}{=} \frac{1}{\beta^t}\|\nabla h_{\mu^{t-1}}(\mathbf{y}^t) - \mathbf{z}^t + (\mathbf{y}^t - \mathcal{A}(\mathbf{X}^t)) \cdot (\beta^t + 2\omega\ddot{\sigma}\sigma^2\beta^{t-1})\| \\
&\overset{②}{=} \frac{1}{\beta^t}\|(1 - \tfrac{1}{\sigma})(\mathbf{z}^{t-1} - \mathbf{z}^t) + (\mathbf{y}^t - \mathcal{A}(\mathbf{X}^t)) \cdot (\beta^t + 2\omega\ddot{\sigma}\sigma^2\beta^{t-1})\| \\
&\overset{③}{=} \mathcal{O}(\|\mathcal{A}(\mathbf{X}^t) - \mathbf{y}^t\|),
\end{aligned}$$

where step ① uses the definition of $\mathbf{d_y}$ in Lemma D.1; step ② uses the fact that $\mathbf{z}^t - \frac{1}{\sigma}(\mathbf{z}^t - \mathbf{z}^{t+1}) = \nabla h_{\mu^t}(\mathbf{y}^{t+1})$, as shown in Lemma 4.1; step ③ uses $\frac{1}{\beta^t}(\mathbf{z}^{t+1} - \mathbf{z}^t) = \sigma(\mathcal{A}(\mathbf{X}^{t+1}) - \mathbf{y}^{t+1})$, and $\beta^{t-1} = \mathcal{O}(\beta^t)$.

**Part (d)**. We bound the term $\frac{1}{\beta^t}\|\mathbf{d_z}\|_\mathsf{F}$. We have: $\frac{1}{\beta^t}\|\mathbf{d_z}\| \leq \frac{1}{\beta^0}\|\mathcal{A}(\mathbf{X}^t) - \mathbf{y}^t\|$.

**Part (e)**. Combining the upper bounds for the terms $\{\frac{1}{\beta^t}\|\mathbf{d_X}\|_\mathsf{F}, \frac{1}{\beta^t}\|\mathbf{d_{X^-}}\|_\mathsf{F}, \frac{1}{\beta^t}\|\mathbf{d_y}\|_\mathsf{F}, \frac{1}{\beta^t}\|\mathbf{d_z}\|_\mathsf{F}\}$, we finish the proof of this lemma. $\square$

**Lemma D.3.** *For* OADMM-RR, *we define* $\{\mathbf{d_X}, \mathbf{d_{X^-}}, \mathbf{d_y}, \mathbf{d_z}\}$ *as in Lemma D.1. There exists a constant $K$ such that :*

$$\frac{1}{\beta^t}\{\|\mathbf{d_X}\|_\mathsf{F} + \|\mathbf{d_{X^-}}\|_\mathsf{F} + \|\mathbf{d_y}\| + \|\mathbf{d_z}\|\} \leq K\{\mathcal{X}^t + \mathcal{Z}^t\},$$

*Here, $\mathcal{X}^t \triangleq \|\frac{1}{\beta^t}\mathbb{G}_{1/2}\|_\mathsf{F}$, and $\mathcal{Z}^t \triangleq \|\mathcal{A}(\mathbf{X}^t) - \mathbf{y}^t\|$.*

*Proof.* We define $\mathbf{G}^t \triangleq \nabla f(\mathbf{X}^t) - \nabla g(\mathbf{X}^t) + A^\mathsf{T}(\mathbf{z}^t + \beta^t(\mathcal{A}(\mathbf{X}^t) - \mathbf{y}^t))$.

We define $\dot{\mathcal{L}}(\mathbf{X}) \triangleq L(\mathbf{X}, \mathbf{y}^t; \mathbf{z}^t; \beta^t, \mu^t)$, we have: $\nabla\dot{\mathcal{L}}(\mathbf{X}^t) = \mathbf{G}^t$.

First, given $\mathbf{X}^t \in \mathcal{M}$, we obtain:

$$\begin{aligned}
\frac{1}{\beta^t}\|\partial I_\mathcal{M}(\mathbf{X}^t) + \nabla\dot{\mathcal{L}}(\mathbf{X}^t)\|_\mathsf{F} &\overset{①}{\leq} \frac{1}{\beta^t}\|\nabla\dot{\mathcal{L}}(\mathbf{X}^t) - \mathbf{X}^t[\nabla\dot{\mathcal{L}}(\mathbf{X}^t)]^\mathsf{T}\mathbf{X}^t\|_\mathsf{F} \\
&\overset{②}{=} \frac{1}{\beta^t}\|\mathbf{G}^t - \mathbf{X}^t[\mathbf{G}^t]^\mathsf{T}\mathbf{X}^t\|_\mathsf{F} = \frac{1}{\beta^t}\|\mathbb{G}_1^t\|_\mathsf{F} \\
&\overset{③}{\leq} \frac{1}{\beta^t}\max(1, 1/\rho) \cdot \|\mathbb{G}_{1/2}\|_\mathsf{F} = \mathcal{O}(\mathcal{X}^t), \tag{49}
\end{aligned}$$

where step ① uses Lemma 2.13; step ② uses the definitions of $\{\mathbf{G}^t, \mathbf{D}_\rho^t\}$ as in Algorithm 1; step ③ uses $\|\mathbb{G}_1\|_\mathsf{F} \leq \max(1, 1/\rho)\|\mathbb{G}_\rho\|_\mathsf{F}$, as shown in Lemma 2.12(*b*).

**Part (a)**. We bound the term $\frac{1}{\beta^t}\|\mathbf{d_X}\|_\mathsf{F}$. We have:

$$\begin{aligned}
&\frac{1}{\beta^t}\|\mathbf{d_X}\|_\mathsf{F} \\
\overset{①}{=}\ & \frac{1}{\beta^t}\|\partial I_\mathcal{M}(\mathbf{X}^t) + \nabla\dot{\mathcal{L}}(\mathbf{X}^t) + 2\omega\ddot{\sigma}\sigma^2\beta^{t-1}\mathcal{A}^\mathsf{T}(\mathcal{A}(\mathbf{X}^t) - \mathbf{y}^t)\|_\mathsf{F} \\
\overset{②}{\leq}\ & \frac{1}{\beta^t}\|\partial I_\mathcal{M}(\mathbf{X}^t) + \nabla\dot{\mathcal{L}}(\mathbf{X}^t)\|_\mathsf{F} + 2\omega\ddot{\sigma}\sigma^2\|\mathcal{A}^\mathsf{T}(\mathcal{A}(\mathbf{X}^t) - \mathbf{y}^t)\|_\mathsf{F} \\
\overset{③}{\leq}\ & \mathcal{O}(\mathcal{X}^t) + \mathcal{O}(\mathcal{Z}^t),
\end{aligned}$$

where step ① uses $\mathbf{d_X} = \partial I_\mathcal{M}(\mathbf{X}^t) + \nabla f(\mathbf{X}^t) - \nabla g(\mathbf{X}^t) + \mathcal{A}^\mathsf{T}(\mathbf{z}^t) + \{\beta^t + 2\omega\ddot{\sigma}\sigma^2\beta^{t-1}\} \cdot \mathcal{A}^\mathsf{T}(\mathcal{A}(\mathbf{X}^t) - \mathbf{y}^t)$ with the choice $\alpha = 0$ for OADMM-RR; step ② uses the triangle inequality and $\beta^{t-1} \leq \beta^t$; step ② uses Inequality (49).

**Part (b)**. We bound the term $\frac{1}{\beta^t}\|\mathbf{d_{X^-}}\|_\mathsf{F}$. Given $\alpha = 0$, we conclude that $\frac{1}{\beta^t}\|\mathbf{d_{X^-}}\|_\mathsf{F} = 0$.

**Part (c).** We bound the terms $\frac{1}{\beta^t}\|\mathbf{d_y}\|_{\mathsf{F}}$ and $\frac{1}{\beta^t}\|\mathbf{d_z}\|_{\mathsf{F}}$. Considering that the same strategies for updating $\{\mathbf{y}^t, \mathbf{z}^t\}$ are employed, their bounds in OADMM-RR are identical to those in OADMM-ER.

**Part (d).** Combining the upper bounds for the terms $\{\frac{1}{\beta^t}\|\mathbf{d_X}\|_{\mathsf{F}}, \frac{1}{\beta^t}\|\mathbf{d_{X^-}}\|_{\mathsf{F}}, \frac{1}{\beta^t}\|\mathbf{d_y}\|_{\mathsf{F}}, \frac{1}{\beta^t}\|\mathbf{d_z}\|_{\mathsf{F}}\}$, we finish the proof of this lemma.

$\square$

Now, we proceed to prove the main result of this lemma.

**Lemma D.4.** *(Subgradient Bounds) (a)* For OADMM-EP, there exists a constant $K > 0$ such that: $\mathrm{dist}(\mathbf{0}, \partial\Theta(\mathrm{w}^t; \mathrm{u}^t)) \leq \beta^t K(e^t + e^{t-1})$. *(b)* For OADMM-RR, there exists a constant $K > 0$ such that: $\mathrm{dist}(\mathbf{0}, \partial\Theta(\mathrm{w}^t; \mathrm{u}^t)) \leq \beta^t K e^t$. Here, $\mathrm{dist}(\mathbf{0}, \partial\Theta(\mathrm{w}^t; \mathrm{u}^t)) \triangleq \{\mathrm{dist}^2(\mathbf{0}, \partial_{\mathbf{X}}\Theta(\mathrm{w}^t; \mathrm{u}^t)) + \mathrm{dist}^2(\mathbf{0}, \partial_{\mathbf{X^-}}\Theta(\mathrm{w}^t; \mathrm{u}^t)) + \mathrm{dist}^2(\mathbf{0}, \partial_{\mathbf{y}}\Theta(\mathrm{w}^t; \mathrm{u}^t)) + \mathrm{dist}^2(\mathbf{0}, \partial_{\mathbf{z}}\Theta(\mathrm{w}^t; \mathrm{u}^t))\}^{1/2}$.

*Proof.* For OADMM-EP, we have:

$$\mathrm{dist}(\mathbf{0}, \partial\Theta(\mathrm{w}^t; \mathrm{u}^t)) = \sqrt{\|\mathbf{d_X}\|_{\mathsf{F}}^2 + \|\mathbf{d_{X^-}}\|_{\mathsf{F}}^2 + \|\mathbf{d_y}\|_{\mathsf{F}}^2 + \|\mathbf{d_z}\|_{\mathsf{F}}^2}$$

$$\overset{\textcircled{1}}{\leq} \|\mathbf{d_X}\|_{\mathsf{F}} + \|\mathbf{d_{X^-}}\|_{\mathsf{F}} + \|\mathbf{d_y}\|_{\mathsf{F}} + \|\mathbf{d_z}\|_{\mathsf{F}}$$

$$\overset{\textcircled{2}}{\leq} K\beta^t\{\mathcal{X}^t + \mathcal{Z}^t + \mathcal{X}^{t-1} + \mathcal{Z}^{t-1}\}$$

$$\overset{\textcircled{3}}{\leq} K\beta^t(e^t + e^{t-1}),$$

where step $\textcircled{1}$ uses $\sqrt{a+b} \leq \sqrt{a} + \sqrt{b}$ for all $a \geq 0$ and $b \geq 0$; step $\textcircled{2}$ uses Lemma D.2; step $\textcircled{3}$ uses the definition of $K$.

For OADMM-RR, using Lemma D.3 and similar strategies, we have: $\mathrm{dist}(\mathbf{0}, \partial\Theta(\mathrm{w}^t; \mathrm{u}^t)) \leq \beta^t K e^t$.

$\square$

### D.2 PROOF OF THEOREM 5.6

*Proof.* We define $\dot{K} \triangleq 3K/\min(\varepsilon_x, \varepsilon_y, \varepsilon_z)$.

Firstly, using Assumption 5.1, we have:

$$\varphi'(\Theta(\mathrm{w}^t; \mathrm{u}^t) - \Theta(\mathrm{w}^\infty; \mathrm{u}^\infty)) \cdot \mathrm{dist}(\mathbf{0}, \partial\Theta(\mathrm{w}^t; \mathrm{u}^t)) \geq 1. \tag{50}$$

Secondly, given the desingularization function $\varphi(\cdot)$ is concave, for any $a, b \in \mathbb{R}$, we have: $\varphi(b) + (a - b)\varphi'(a) \leq \varphi(a)$. Applying the inequality above with $a = \Theta(\mathrm{w}^t; \mathrm{u}^t) - \Theta(\mathrm{w}^\infty; \mathrm{u}^\infty)$ and $b = \Theta(\mathrm{w}^{t+1}; \mathrm{u}^{t+1}) - \Theta(\mathrm{w}^\infty; \mathrm{u}^\infty)$, we have:

$$(\Theta(\mathrm{w}^t; \mathrm{u}^t) - \Theta(\mathrm{w}^{t+1}; \mathrm{u}^{t+1})) \cdot \varphi'(\Theta(\mathrm{w}^t; \mathrm{u}^t) - \Theta(\mathrm{w}^\infty; \mathrm{u}^\infty))$$

$$\leq \underbrace{\varphi(\Theta(\mathrm{w}^t; \mathrm{u}^t) - \Theta(\mathrm{w}^\infty; \mathrm{u}^\infty))}_{\triangleq \varphi^t} - \underbrace{\varphi(\Theta(\mathrm{w}^{t+1}; \mathrm{u}^{t+1}) - \Theta(\mathrm{w}^\infty; \mathrm{u}^\infty))}_{\triangleq \varphi^{t+1}}. \tag{51}$$

Third, we derive the following inequalities for OADMM-EP:

$$\min(\varepsilon_z, \varepsilon_y, \varepsilon_x)\beta^t\{\|\mathcal{A}(\mathbf{X}^{t+1}) - \mathbf{y}^{t+1}\|_2^2 + \|\mathbf{y}^{t+1} - \mathbf{y}^t\|_2^2 + \|\mathbf{X}^{t+1} - \mathbf{X}^t\|_{\mathsf{F}}^2\}$$

$$\overset{\textcircled{1}}{\leq} \varepsilon_z\beta^t\|\mathcal{A}(\mathbf{X}^{t+1}) - \mathbf{y}^{t+1}\|_2^2 + \varepsilon_y\beta^t\|\mathbf{y}^{t+1} - \mathbf{y}^t\|_2^2 + \varepsilon_x\ell(\beta^t)\|\mathbf{X}^{t+1} - \mathbf{X}^t\|_{\mathsf{F}}^2$$

$$\overset{\textcircled{2}}{\leq} \Theta^t - \Theta^{t+1} = \Theta(\mathrm{w}^t; \mathrm{u}^t) - \Theta(\mathrm{w}^{t+1}; \mathrm{u}^{t+1})$$

$$\overset{\textcircled{3}}{\leq} (\varphi^t - \varphi^{t+1}) \cdot \frac{1}{\varphi'(\Theta(\mathrm{w}^t; \mathrm{u}^t) - \Theta(\mathrm{w}^\infty; \mathrm{u}^\infty)))}$$

$$\overset{\textcircled{4}}{\leq} (\varphi^t - \varphi^{t+1}) \cdot \mathrm{dist}(\mathbf{0}, \partial\Theta(\mathrm{w}^t; \mathrm{u}^t))$$

$$\overset{\textcircled{5}}{\leq} (\varphi^t - \varphi^{t+1}) \cdot K\beta^t(e^t + e^{t-1}), \tag{52}$$

where step ① uses $\ell(\beta^t) \geq \beta^t \underline{\ell}$; step ② uses Lemma 4.7; step ③ uses Inequality (51); step ④ uses Inequality (50); step ⑤ uses Lemma 5.4. We further derive the following inequalities:

$$
\begin{aligned}
(e^{t+1})^2 &\triangleq (\|\mathcal{A}(\mathbf{X}^{t+1}) - \mathbf{y}^{t+1}\|_2 + \|\mathbf{y}^{t+1} - \mathbf{y}^t\|_2 + \|\mathbf{X}^{t+1} - \mathbf{X}^t\|_\mathsf{F})^2 \\
&\overset{①}{\leq} 3 \cdot \{\|\mathcal{A}(\mathbf{X}^{t+1}) - \mathbf{y}^{t+1}\|_2^2 + \|\mathbf{y}^{t+1} - \mathbf{y}^t\|_2^2 + \|\mathbf{X}^{t+1} - \mathbf{X}^t\|_\mathsf{F}^2\} \\
&\overset{②}{\leq} \{3K / \min(\varepsilon_z, \varepsilon_y, \varepsilon_x)\} \cdot (e^t + e^{t-1}) \cdot (\varphi^t - \varphi^{t+1}),
\end{aligned}
\tag{53}
$$

where step ① uses the norm inequality that $(a + b + c)^2 \leq 3(a^2 + b^2 + c^2)$ for any $a, b, c \in \mathbb{R}$; step ② uses Inequality (52).

Fourth, we derive the following inequalities for OADMM-RR:

$$
\begin{aligned}
&\min(\varepsilon_z, \varepsilon_y, \varepsilon_x)\beta^t \{\|\mathcal{A}(\mathbf{X}^{t+1}) - \mathbf{y}^{t+1}\|_2^2 + \|\mathbf{y}^{t+1} - \mathbf{y}^t\|_2^2 + \|\tfrac{1}{\beta}\mathbb{G}_{1/2}^t\|_\mathsf{F}^2\} \\
&\leq \varepsilon_z\beta^t\|\mathcal{A}(\mathbf{X}^{t+1}) - \mathbf{y}^{t+1}\|_2^2 + \varepsilon_y\beta^t\|\mathbf{y}^{t+1} - \mathbf{y}^t\|_2^2 + \tfrac{\varepsilon_x}{\beta}\|\mathbb{G}_{1/2}^t\|_\mathsf{F}^2 \\
&\overset{①}{\leq} \Theta^t - \Theta^{t+1} = \Theta(\mathbb{w}^t; \mathbb{u}^t) - \Theta(\mathbb{w}^{t+1}; \mathbb{u}^{t+1}) \\
&\overset{②}{\leq} (\varphi^t - \varphi^{t+1}) \cdot \tfrac{1}{\varphi'(\Theta(\mathbb{w}^t;\mathbb{u}^t) - \Theta(\mathbb{w}^\infty;\mathbb{u}^\infty)))} \\
&\overset{③}{\leq} (\varphi^t - \varphi^{t+1}) \cdot \mathrm{dist}(\mathbf{0}, \partial\Theta(\mathbb{w}^t; \mathbb{u}^t)) \\
&\overset{④}{\leq} (\varphi^t - \varphi^{t+1}) \cdot K\beta^t(e^t + e^{t-1}),
\end{aligned}
\tag{54}
$$

where step ① uses Lemma 4.12; step ② uses Inequality (51); step ③ uses Inequality (50); step ④ uses Lemma 5.4. We further derive the following inequalities:

$$
\begin{aligned}
(e^{t+1})^2 &\triangleq (\|\mathcal{A}(\mathbf{X}^{t+1}) - \mathbf{y}^{t+1}\| + \|\mathbf{y}^{t+1} - \mathbf{y}^t\| + \|\tfrac{1}{\beta}\mathbb{G}_{1/2}^t\|_\mathsf{F})^2 \\
&\overset{①}{\leq} 3 \cdot \{\|\mathcal{A}(\mathbf{X}^{t+1}) - \mathbf{y}^{t+1}\|_2^2 + \|\mathbf{y}^{t+1} - \mathbf{y}^t\|_2^2 + \|\tfrac{1}{\beta}\mathbb{G}_{1/2}^t\|_\mathsf{F}^2\} \\
&\overset{②}{\leq} \{3K / \min(\varepsilon_z, \varepsilon_y, \varepsilon_x)\} \cdot (\varphi^t - \varphi^{t+1}) \cdot (e^t + e^{t-1}),
\end{aligned}
\tag{55}
$$

where step ① uses the norm inequality that $(a + b + c)^2 \leq 3(a^2 + b^2 + c^2)$ for any $a, b, c \in \mathbb{R}$; step ② uses Inequality (54).

**Part (a)**. Given Inequalities (53) and (55), we establish the following unified inequality applicable to both OADMM-EP and OADMM-RR:

$$
(e^{t+1})^2 \leq (e^t + e^{t-1}) \cdot \underbrace{\{3K / \min(\varepsilon_z, \varepsilon_y, \varepsilon_x)\}}_{\triangleq \dot{K}} \cdot (\varphi^t - \varphi^{t+1}).
\tag{56}
$$

**Part (b)**. Considering Inequality (56) and applying Lemma A.10 with $p^t \triangleq \dot{K}\varphi^t$, we have:

$$
\forall t, \sum_{i=t}^{\infty} e^{i+1} \leq e^t + e^{t-1} + 4\dot{K}\varphi^t.
$$

Letting $t = 1$, we have: $\sum_{i=1}^{\infty} e^{i+1} \leq e^1 + e^0 + 4\dot{K}\varphi^1$.

$\square$

### D.3 Proof of Lemma 5.8

*Proof.* We define $d^t \triangleq \sum_{i=t}^{\infty} e^{i+1}$.

**Part (a-i)**. For OADMM-EP, we have for all $t \geq 1$: $\|\mathbf{X}^t - \mathbf{X}^\infty\|_\mathsf{F} \overset{①}{\leq} \sum_{i=t}^{\infty} \|\mathbf{X}^i - \mathbf{X}^{i+1}\|_\mathsf{F} \leq \sum_{i=t}^{\infty}\{\|\mathbf{X}^{i+1} - \mathbf{X}^i\|_\mathsf{F} + \|\mathbf{y}^{i+1} - \mathbf{y}^i\| + \|\mathcal{A}(\mathbf{X}^{i+1}) - \mathbf{y}^{i+1}\|\} = \sum_{i=t}^{\infty} e^{i+1} \triangleq d^t$, where step ① use the triangle inequality.

**Part (a-ii)**. For OADMM-RR, we have: $\|\mathbf{X}^{t+1} - \mathbf{X}^t\|_\mathsf{F} \overset{①}{=} \|\mathrm{Retr}_{\mathbf{X}^t}(-\eta^t\mathbb{G}_\rho^t) - \mathbf{X}^t\|_\mathsf{F} \overset{②}{\leq} \dot{k}\|\eta^t\mathbb{G}_\rho^t\|_\mathsf{F} \overset{③}{\leq} \dot{k}\eta^t\max(2\rho, 1)\|\mathbb{G}_{1/2}^t\|_\mathsf{F} \overset{④}{=} \dot{k}\max(2\rho, 1)\tfrac{b^t\gamma^j}{\beta^t}\|\mathbb{G}_{1/2}^t\|_\mathsf{F} \overset{⑤}{\leq} \dot{k}\max(2\rho, 1)\overline{b}\overline{\gamma} \cdot$

$\|\frac{1}{\beta^t}\mathbb{G}^t_{1/2}\|_{\mathsf{F}} = \mathcal{O}(\|\frac{1}{\beta^t}\mathbb{G}^t_{1/2}\|_{\mathsf{F}})$, where step ① uses the update rule of $\mathbf{X}^{t+1}$; step ② uses Lemma 2.10; step ③ uses Lemma 2.12(**c**); step ④ uses the definition of $\eta^t \triangleq \frac{b^t\gamma^j}{\beta^t}$; step ⑤ uses $b^t \leq \bar{b}$, and the fact that $\gamma^j \leq \bar{\gamma}$. Furthermore, we derive for all $t \geq 1$: $\|\mathbf{X}^t - \mathbf{X}^\infty\|_{\mathsf{F}} \leq \sum_{i=t}^\infty \|\mathbf{X}^i - \mathbf{X}^{i+1}\|_{\mathsf{F}} \leq \mathcal{O}(\sum_{i=t}^\infty \|\frac{1}{\beta^t}\mathbb{G}^i_{1/2}\|_{\mathsf{F}}) \leq \mathcal{O}(\sum_{i=t}^\infty e^{i+1}) = \mathcal{O}(d^t)$.

**Part (b)**. We define $\varphi^t \triangleq \varphi(s^t)$, where $s^t \triangleq \Theta(\mathbb{w}^t; \mathbb{u}^t) - \Theta(\mathbb{w}^\infty; \mathbb{u}^\infty)$. Using the definition of $d^t$, we derive:

$$
\begin{aligned}
d^t \quad &\triangleq \quad \sum_{i=t}^\infty e^{i+1} \\
&\overset{①}{\leq} \quad e^t + e^{t-1} + 4\dot{K}\varphi^t \\
&\overset{②}{=} \quad e^t + e^{t-1} + 4\dot{K}\tilde{c}\cdot\{[s^t]^{\tilde{\sigma}}\}^{\frac{1-\tilde{\sigma}}{\tilde{\sigma}}} \\
&\overset{③}{=} \quad e^t + e^{t-1} + 4\dot{K}\tilde{c}\cdot\{\tilde{c}(1-\tilde{\sigma})\cdot\frac{1}{\varphi'(s^t)}\}^{\frac{1-\tilde{\sigma}}{\tilde{\sigma}}} \\
&\overset{④}{\leq} \quad e^t + e^{t-1} + 4\dot{K}\tilde{c}\cdot\{\tilde{c}(1-\tilde{\sigma})\cdot\mathrm{dist}(\mathbf{0},\partial\Theta(\mathbb{w}^t;\mathbb{u}^t))\}^{\frac{1-\tilde{\sigma}}{\tilde{\sigma}}} \\
&\overset{⑤}{\leq} \quad e^t + e^{t-1} + 4\dot{K}\tilde{c}\cdot\{\tilde{c}(1-\tilde{\sigma})\cdot\beta^t K(e^t+e^{t-1})\}^{\frac{1-\tilde{\sigma}}{\tilde{\sigma}}} \\
&\overset{⑥}{=} \quad d^{t-2} - d^t + 4\dot{K}\tilde{c}\cdot\{\tilde{c}(1-\tilde{\sigma})\cdot\beta^t K(d^{t-2}-d^t)\}^{\frac{1-\tilde{\sigma}}{\tilde{\sigma}}} \\
&= \quad d^{t-2} - d^t + \underbrace{4\dot{K}\tilde{c}\cdot[\tilde{c}(1-\tilde{\sigma})K]^{\frac{1-\tilde{\sigma}}{\tilde{\sigma}}}}_{\triangleq\ddot{K}}\cdot\{(\beta^t(d^{t-2}-d^t))^{\frac{1-\tilde{\sigma}}{\tilde{\sigma}}}\},
\end{aligned}
$$

where step ① uses $\sum_{i=t}^\infty e^{i+1} \leq e^t + e^{t-1} + 4\dot{K}\varphi^t$, as shown in Theorem 5.6(**b**); step ② uses the definitions that $\varphi^t \triangleq \varphi(s^t)$, $s^t \triangleq \Theta(\mathbb{w}^t;\mathbb{u}^t) - \Theta(\mathbb{w}^\infty;\mathbb{u}^\infty)$, and $\varphi(s) = \tilde{c}s^{1-\tilde{\sigma}}$; step ③ uses $\varphi'(s) = \tilde{c}(1-\tilde{\sigma})\cdot[s]^{-\tilde{\sigma}}$, leading to $[s^t]^{\tilde{\sigma}} = \tilde{c}(1-\tilde{\sigma})\cdot\frac{1}{\varphi'(s^t)}$; step ④ uses Assumption 5.1 that $1 \leq \mathrm{dist}(\mathbf{0},\partial\Theta(\mathbb{w}^t;\mathbb{u}^t))\cdot\varphi'(s^t)$; step ⑤ uses $\mathrm{dist}(\mathbf{0},\partial\Theta(\mathbb{w}^t;\mathbb{u}^t)) \leq K(e^t+e^{t-1})$ for both OADMM-EP and OADMM-RR, as shown in Lemma 5.4; step ⑥ uses the fact that $e^t = d^{t-1} - d^t$, which implies:

$$e^t + e^{t-1} = (d^{t-1}-d^t) + (d^{t-2}-d^{t-1}) = d^{t-2} - d^t.$$

$\square$

D.4 PROOF OF THEOREM 5.9

*Proof.* Using Lemma 5.8(**b**), we have:

$$d^t \leq d^{t-2} - d^t + \ddot{K}\cdot\{(\beta^t(d^{t-2}-d^t))^{\frac{1-\tilde{\sigma}}{\tilde{\sigma}}}\}. \tag{57}$$

We consider two cases for Inequality (57).

**Part (a)**. $\tilde{\sigma} \in (\frac{1}{4}, \frac{1}{2}]$. We define $u \triangleq \frac{p(1-\tilde{\sigma})}{\tilde{\sigma}} \in [\frac{1}{3}, 1)$, where $p = \frac{1}{3}$ is a fixed constant.

We define $\tilde{\beta}^t \triangleq \ddot{K}(\beta^t)^{\frac{1-\tilde{\sigma}}{\tilde{\sigma}}}$. We define $t' \triangleq \{i \,|\, d^{i-2} - d^i \leq 1\}$.

For all $t \geq t'$, we have from Inequality (57):

$$
\begin{aligned}
d^t \quad &\leq \quad d^{t-2} - d^t + (d^{t-2}-d^t)^{\frac{1-\tilde{\sigma}}{\tilde{\sigma}}}\cdot\underbrace{\ddot{K}(\beta^t)^{\frac{1-\tilde{\sigma}}{\tilde{\sigma}}}}_{\triangleq\tilde{\beta}^t} \\
&\overset{①}{\leq} \quad d^{t-2} - d^t + (d^{t-2}-d^t)\cdot\tilde{\beta}^t \\
&\leq \quad d^{t-2}\cdot\frac{\tilde{\beta}^t+1}{\tilde{\beta}^t+2},
\end{aligned}
\tag{58}
$$

where step ① uses the fact that $[\Delta^{(1-\tilde{\sigma})/\tilde{\sigma}}]/\Delta = \Delta^{(1-2\tilde{\sigma})/\tilde{\sigma}} = \Delta^{(1/\tilde{\sigma}-2)} \leq \Delta^0 = 1$ for all $\Delta = d^{t-2} - d^t \in [0, 1]$ and $\tilde{\sigma} \in (0, \frac{1}{2}]$.

Furthermore, We derive:

$$\sum_{t=1}^{T}(\tilde{\beta}^t)^{-1} \overset{①}{=} \mathcal{O}\left(\sum_{t=1}^{T}[t^p]^{-\frac{1-\tilde{\sigma}}{\tilde{\sigma}}}\right) \overset{②}{=} \mathcal{O}(\sum_{t=1}^{T} t^{-u}) \overset{③}{\geq} \mathcal{O}(T^{1-u}),$$

where step ① uses $\tilde{\beta}^t \triangleq \ddot{K}(\beta^t)^{\frac{1-\tilde{\sigma}}{\tilde{\sigma}}}$ and $\beta^t \triangleq \beta^0(1+\xi t^p) = \mathcal{O}(t^p)$; step ② uses the definition of $u$; step ③ uses Lemma A.9 that: $\sum_{t=1}^{T} t^{-u} \geq (1-u)T^{1-u} = \mathcal{O}(T^{1-u})$ for all $u \in (0,1)$.

Applying Lemma Lemma A.12 with $a = 1 - u$, we have:

$$d^T \leq \mathcal{O}\left(\frac{1}{\exp(T^{1-u})}\right).$$

**Part (b)**. $\tilde{\sigma} \in (\frac{1}{2}, 1)$. We define $w \triangleq \frac{1-\tilde{\sigma}}{\tilde{\sigma}} \in (0,1)$, and $\tau \triangleq 1/w - 1 \in (0,\infty)$.

We define $\tilde{\beta}^t = \dot{K}^{1/w}\beta^t$, where $\dot{K} \triangleq \ddot{K} + R^{1-w}(\beta^0)^{-w}$, and $R \triangleq d^0$.

Notably, we have: $d^{t-2} - d^t \leq d^0 \triangleq R$ for all $t \geq 2$.

For all $t \geq 2$, we have from Inequality (57):

$$
\begin{aligned}
d^t &\leq & d^{t-2} - d^t + \ddot{K}(\beta^t)^{\frac{1-\tilde{\sigma}}{\tilde{\sigma}}}(d^{t-2} - d^t)^{\frac{1-\tilde{\sigma}}{\tilde{\sigma}}} \\
&\overset{①}{=} & \ddot{K}\{\beta^t(d^{t-2} - d^t)\}^w + d^{t-2} - d^t \\
&\overset{②}{\leq} & \ddot{K}\{\beta^t(d^{t-2} - d^t)\}^w + (d^{t-2} - d^t)^w \cdot R^{1-w} \\
&\overset{③}{\leq} & \ddot{K}\{\beta^t(d^{t-2} - d^t)\}^w + (d^{t-2} - d^t)^w \cdot R^{1-w} \cdot (\frac{\beta^t}{\beta^0})^w \\
&= & \{\beta^t(d^{t-2} - d^t)\}^w \cdot \underbrace{(\ddot{K} + R^{1-w} \cdot (\beta^0)^{-w})}_{\triangleq \dot{K}},
\end{aligned}
$$

where step ① uses the the definition of $w$; step ② uses the fact that $\max_{x \in (0,R]} \frac{x}{x^w} \leq R^{1-w}$ if $w \in (0,1)$ and $R > 0$; step ③ uses $\beta^0 \leq \beta^t$ and $w \in (0,1)$. We further obtain:

$$\underbrace{[d^t]^{1/w}}_{=[d^t]^{\tau+1}} \leq (d^{t-2} - d^t) \cdot \underbrace{\beta^t \dot{K}^{1/w}}_{\triangleq \tilde{\beta}^t}.$$

Additionally, we have:

$$\sum_{t=1}^{T}(1/\tilde{\beta}^t) \overset{①}{=} \mathcal{O}(\sum_{t=1}^{T}(1/\beta^t)) \overset{②}{=} \mathcal{O}(\sum_{t=1}^{T} t^{-p}) \overset{③}{\geq} \mathcal{O}(T^{1-p}),$$

where step ① uses $\tilde{\beta}^t = \dot{K}^{1/w}\beta^t$; step ② uses $\beta^t \triangleq \beta^0(1+\xi t^p) = \mathcal{O}(t^p)$; step ③ uses Lemma A.9 that: $\sum_{t=1}^{T} t^{-u} \geq (1-p)T^{1-u} = \mathcal{O}(T^{1-p})$ for all $p \in (0,1)$.

Applying Lemma A.13 with $a = 1 - p$, we have:

$$d^T \leq \mathcal{O}(1/(T^{(1-p)/\tau})).$$

**Part (c)**. Finally, using the fact $\|\mathbf{X}^T - \mathbf{X}^\infty\|_F \leq \mathcal{O}(d^T)$ as shown in Lemma D.3(**b**), we finish the proof of this theorem.

$\square$

# E   ADDITIONAL EXPERIMENTS DETAILS AND RESULTS

▶ **Datasets**. In our experiments, we utilize several datasets comprising both randomly generated and publicly available real-world data. These datasets are structured as data matrices $\mathbf{D} \in \mathbb{R}^{\dot{m} \times \dot{d}}$. They are denoted as follows: 'mnist-$\dot{m}$-$\dot{d}$', 'TDT2-$\dot{m}$-$d'$', 'sector-$m'$-$d'$', and 'randn-$\dot{m}$-$\dot{d}$', where $\text{randn}(m, n)$ generates a standard Gaussian random matrix of size $m \times n$. The construction of $\mathbf{D} \in \mathbb{R}^{\dot{m} \times \dot{d}}$ involves randomly selecting $\dot{m}$ examples and $\dot{d}$ dimensions from the original

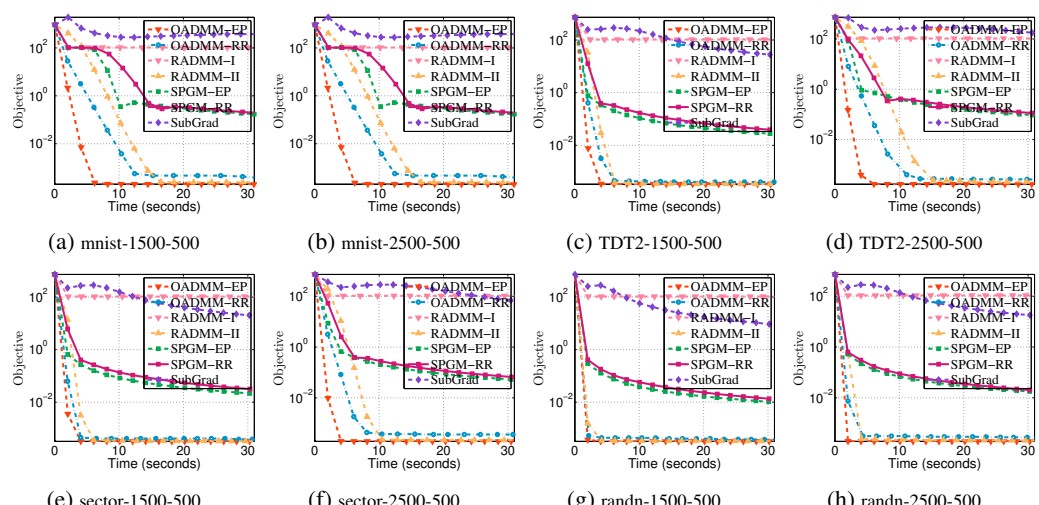

Figure 3: The convergence curve of the compared methods with $\dot{\rho} = 10$.

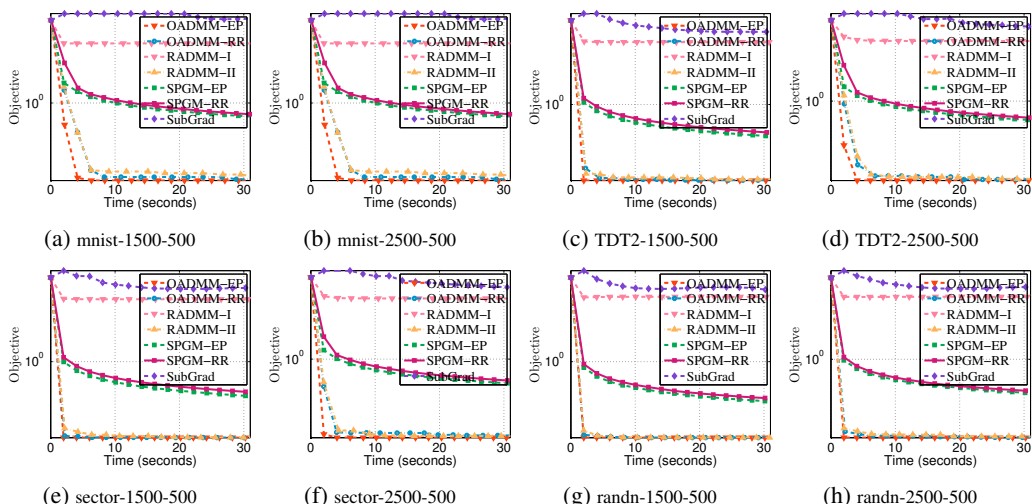

Figure 4: The convergence curve of the compared methods with $\dot{\rho} = 100$.

real-world dataset, sourced from `http://www.cad.zju.edu.cn/home/dengcai/Data/TextData.html` and `https://www.csie.ntu.edu.tw/~cjlin/libsvm/`. Subsequently, we normalize each column of $\mathbf{D}$ to possess a unit norm and center the data by subtracting the mean, denoted as $\mathbf{D} \Leftarrow \mathbf{D} - \mathbf{1}\mathbf{1}^{\mathsf{T}}\mathbf{D}$.

▶ **Additional experiment Results**. We present additional experimental results in Figures 3, 4, and 5. The figures demonstrate that the proposed OADMM method generally outperforms the other methods, with OADMM-EP surpassing OADMM-RR. These results reinforce our previous conclusions.

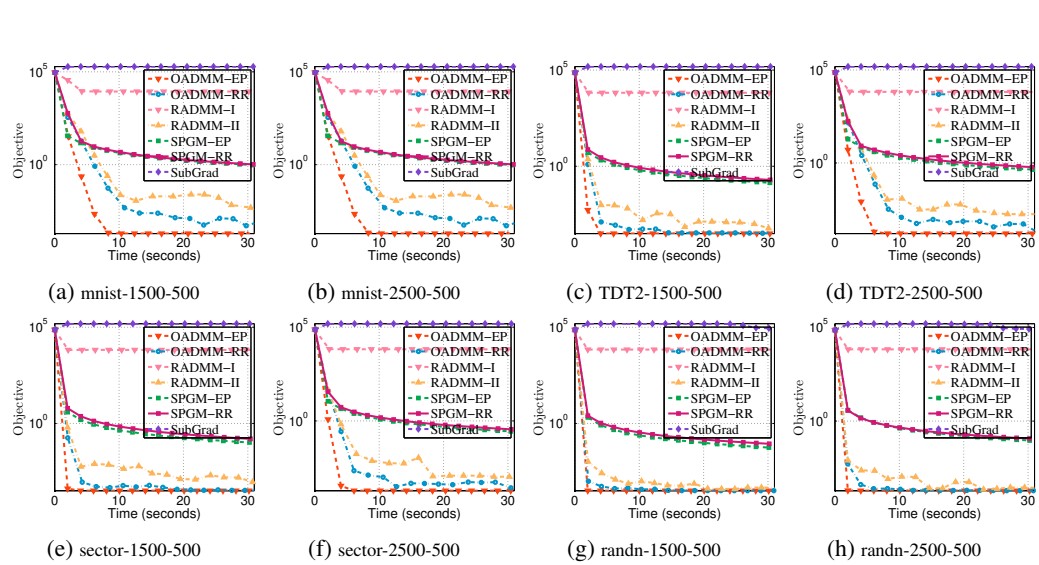

Figure 5: The convergence curve of the compared methods with $\dot{\rho} = 1000$.

