# OpenReview forum: "ADMM for Nonsmooth Composite Optimization under Orthogonality Constraints"
_ICLR.cc/2025/Conference — Submitted to ICLR 2025_

### Official Review · Reviewer_yRDE · 2024-11-01

**Soundness:** 3
**Presentation:** 2
**Contribution:** 2
**Rating:** 3
**Confidence:** 4

**Summary:**

This paper addresses a class of structured, nonconvex, nonsmooth optimization problems with orthogonality constraints. The authors propose OADMM to solve these problems by utilizing efficient proximal linearized strategies. Under mild assumptions, OADMM is shown to converge to a critical point with an ergodic rate of $O(1/\epsilon^3)$, and exhibits either super-exponential or polynomial convergence depending on the setting, based on the Kurdyka-Lojasiewicz (KL) inequality. Numerical experiments indicate that OADMM achieves state-of-the-art performance.

**Strengths:**

1. The authors propose a new algorithm OADMM.
2. The OADMM algorithm proposed in the paper demonstrates an iteration complexity of $O(\epsilon^{-3})$, which aligns with the best-known complexity result for this type of problem.
3. Additionally, the paper establishes the convergence of the entire sequence and derives a convergence rate under the KL assumption.
4. Numerical results show the fast convergence.

**Weaknesses:**

1. This paper is not well-written. A lot of technical results are not well organized. I find it hard to get the novel/essential technical lemmas.
2. The authors study optimization problems of a very general form. However, they only perform experiments on sparse PCA (the linear operator A=I).
3. Line 133 and 461: "(iii) SPGM-EP" should be corrected to "RR."
4. The paper suggests that a Nesterov extrapolation strategy with parameter $\alpha$ can accelerate the convergence of primal variables. It would be beneficial if the authors could provide theoretical or numerical evidence to substantiate this claim. Although this is not critical given the paper's primary focus on theoretical convergence analysis, such verification could further enhance the contribution.
5. In the convergence rate analysis, the paper imposes a smoothness assumption on the function $g$. Given this assumption, it seems possible to absorb $g$ into the smooth function $f$, potentially simplifying the analysis.

**Questions:**

1. Could the authors point out the core technical difficulties in this paper?
2. Could the authors compare OADMM with the ManPG (by linearizing the concave part) in the experiments?
3. Note that the sparse PCA is not weakly convex, it is not rigorious to use the Riemannian subgradient method in the experiment. That is partly why subgradient methods do not work. See Line 473.
4. The applications are rather limited. The authors only provide the sparse PCA example. Could the authors find other applications with the linear operator A not being the identity operator?
5. There is some mistake in Line 602 in the reference part.
6. I do not understand why (a) in Lemma 2.2 holds. Could the authors explain it?
7. The paper claims that the iteration complexity improves from $O(\epsilon^{-4})$ to $O(\epsilon^{-3})$  due to the decreasing step size in lines 142-143. However, in lines 255–259, the paper also claims that the over-relaxation step size $\sigma$ for the dual variable and the Nesterov extrapolation parameter $\alpha$ accelerate the convergence. This is somewhat confusing. My understanding is that the complexity might remain the same if we set $\alpha = 0$, $\sigma = 1$, and $\theta > 1$ (in this case, OADMM would reduce to the classical linearized proximal ADMM). In other words, $\alpha$ and $\sigma$ primarily enhance numerical performance rather than complexity. Could the authors clarify this point to make the mechanism behind the complexity improvement more explicit?
8. The paper does not include a convergence rate analysis for cases where the KL exponent $\tilde{\sigma} < 1/4$. Could the authors provide an explanation for this omission?
9. The description of SPGM as a penalty method (lines 100–103) seems inconsistent with the approach in the original paper, where the problem is solved by taking a gradient step with respect to the smoothed problem $\min_{x \in M} f(x) + h_\mu(x)$ . Are these descriptions equivalent? Clarification on this equivalence would be helpful.
10. It seems that $\mathcal{A}$ could be a smooth mapping with a bounded operator norm and Lipschitz continuous gradient to ensure that the function $S$ is smooth. Is the assumption that $\mathcal{A}$ is a linear mapping is due to practical computation considerations?

---

### Official Review · Reviewer_GNWd · 2024-11-02

**Soundness:** 2
**Presentation:** 2
**Contribution:** 2
**Rating:** 3
**Confidence:** 4

**Summary:**

The paper proposes an ADMM method for solving a class of nonsmooth composite optimization problems with orthogonality constraints. The authors analyze the convergence rate and present results from numerical experiments.

**Strengths:**

1. The problem class considered seems general and encompasses many application problems with orthogonality constraints.
2. The proposed ADMM methods incorporate several practical strategies, such as prox-linear updates and the Barzilai-Borwein step size.

**Weaknesses:**

Here’s a refined version of each point for improved clarity and grammar:
1. The definition of the stationary point (i.e., Crit) is not rigorous, and its validation is missing. This makes some difficulty in understanding the complexity results.
2. The algorithm includes many parameters. How can the choices of penalty and smoothing parameters be justified?
3. The presentation could be improved. For example:
   a. Is there any intuition behind why a one-step update on the $X$ variable ensures convergence, especially given the nonconvexity of the $X$-subproblem?
   b. Why is the smoothing parameter $\mu$ important?
4. The numerical results appear a bit preliminary, and it would be beneficial to include more applications for comparison.

**Questions:**

Here’s a refined version of each point to improve clarity and readability:

1. Definition 2.7: Since the subgradient is a set, how is Crit calculated?
2. Definition 2.7:  The defined stationarity differs from that in the existing literature because the problem is more complex. Please validate this notion of stationarity with established concepts for problem (1).
3. Algorithm 1: Only one iteration is performed for each \( X \) update. Is the BB step size truly beneficial, given that the gradients $G^{t-1}$ and $G^t$ are computed based on different objectives?
4. Algorithm 1: In classical ADMM, the penalty parameter is typically fixed. Does a constant $\beta^t$ ensure the same convergence guarantee?
5. The connection between Crit and $e^t$ is unclear. From Lemmas 4.7 and 4.12, it appears that the average of $e^t$ converges. However, the definition of $e^t$ involves the optimality of subproblems, such as $x^t - x^{t-1}$ and $y^t - y^{t-1}$. Careful elaboration on $e^t$ and the stationarity of the original problem is needed.
6. Assumption 5.1: How are the KL assumptions on $L$ connected to problem (1)? Currently, it seems standard to show the rate under the KL of $\Theta$.
7. Line 453-454: There are mismatches in the descriptions.
8. The RADMM algorithm cannot handle nonconvex nonsmooth terms in the objectives. How is it implemented?
9. SPGM-EP: It is unclear how these two algorithms are implemented. Is $\|x\|\_1 - \|x\|\_{[k]}$ weakly convex, and is its Moreau envelope smoothing used?
10. Line 471-478: The description of RADMM is missing.
11. Abstract: What does "O" in OADMM stand for?
12. Line 132: "OADMM-EP" should be "OADMM-RR." Are these abbreviations necessary here?

---

### Official Review · Reviewer_9k7U · 2024-11-03

**Soundness:** 3
**Presentation:** 2
**Contribution:** 2
**Rating:** 5
**Confidence:** 4

**Summary:**

This paper studies nonconvex nonsmooth problems with orthogonality constraints and proposes a Riemannian primal-dual algorithm called OADMM, which comes with global convergence guarantees and sequential convergence under KL assumption.

**Strengths:**

The overall complexity conclusions presented in the paper are reasonable and align with similar algorithms proposed in RADMM (Li et al., 2022) and ManIAL (Deng et al., 2024). This paper can be seen as an accelerated version of RADMM, incorporating the Barzilai-Borwein step size and extrapolation for improved performance.

**Weaknesses:**

The paper is written in a way that compiles all the technical material with complex notations, making it difficult for readers to follow. It would be beneficial if the author could summarize the results more effectively and provide insights and discussions. For instance, highlighting what specifically contributes to the complexity improvement compared to RADMM would enhance clarity. Additionally, I have several detailed concerns:

1. The setting of the paper requires $h$ to be proximal-friendly. In Table 1, when comparing this method to those that do not require proximal operations (e.g., Li et al., 2021), it would be helpful to make a clear statement regarding this distinction before presenting the complexity comparison. This is important because, without proximal information, classical smoothing techniques typically incur an additional worst-case complexity of $\mathcal{O}(\epsilon^{-1})$, which should not be ignored.

2. The author claims that the proposed OADMM can achieve the oracle complexity of $\mathcal{O}(\epsilon^{-3})$, but it is unclear what kind of explicit oracle is being used. If it relies on the classical first-order oracle, some additional costs of the proposed algorithm should be discussed. For example, the subproblem of OADMM-EP restricts it to the manifold (even though some manifolds have closed-form solutions). Additionally, the line search step in OADMM-RR can be very costly, and computing its exact cost is challenging.

3. In the numerical experiments section, the current function $f$ in the given sparse PCA example does not appear to satisfy the $L_f$-smooth assumption. It would be better to do a further computation combining with the properties of the orthogonal constraints to make a equivalent formulation satisfying assumptions fully supported by the theoretical analysis.

4. The paper could be more mathematically rigorous in several places. For example: - Line 038: The author seems to intend to refer to $\mathcal{A}$ rather than $\mathcal{A}(X)$ as not being a mapping. - Line 041: $\mathcal{f}(X)$ is incorrectly referred to as a function, which occurs repeatedly throughout the paper.

**Questions:**

Is the approach extendable to general submanifolds? The current approach seems to be applicable only to compact manifolds. For instance, in the primal update, it is challenging to control boundedness, which implies that the lower boundedness of the potential function may not be guaranteed. Could you comment on this?

---

### Official Review · Reviewer_AL9y · 2024-11-04

**Soundness:** 2
**Presentation:** 2
**Contribution:** 3
**Rating:** 5
**Confidence:** 4

**Summary:**

This paper proposes an ADMM-type algorithm to solve a class of nonconvex nonsmooth composite problems under orthogonality constraints. The iteration complexity of the proposed OADMM surpasses that of the ADMM by Li et al. (2022) and aligns with the best-known complexity found in studies such as Beck and Rosset (2023). Numerical results are provided to evaluate the efficiency of the proposed method.

**Strengths:**

The complexity of the ADMM proposed in this paper is superior to that of the existing Riemannian ADMM. Additionally, this paper addresses a more general formulation.

**Weaknesses:**

Overall, I find the topic and results intriguing. However, the writing style could be improved for a conference paper. The authors attempt to incorporate extrapolation or adaptive step sizes in the proposed algorithms, yet these techniques are not implemented in the numerical experiments.  If they had focused simply on ADMM and make some comments, the paper might be more accessible.

Moreover, the proposed ADMM is similar to the existing Riemannian ADMM and applies the adaptive technique from Beck and Rosset (2023) to update the smoothing parameter, $\mu$. The authors should emphasize this point in the paper and clarify the key factor contributing to the improved complexity.

**Questions:**

- The paper discusses the convergence rate of the proposed method, though this might be somewhat standard, as it has been addressed in prior work on nonconvex and nonsmooth ADMM. Moreover, based on the K\L\ framework, when the parameter $\tilde \sigma \in (0, 1/2]$, the convergence rate is linear. Why, then, is the range $(1/4, 1/2]$ in this paper?

-  The authors claim that they can use a larger dual step size, $\sigma \in [1,2)$; however, they do not clarify how this value is chosen. The same issue arises for the parameter $\theta$.

- Although the authors note that OADMM-RR uses the BB step size, only a simple constant step size $b^t \equiv 1$ is employed in the numerical experiments.

- Several parameters in the proposed algorithm require careful tuning. For instance, the authors select $\xi$ from a range---are the results in Figures 1 and 2 based on the optimal $\xi$?

- The performance of OADMM-RR and RADMM-II is very close. Could RADMM-II achieve better results with careful selection of $\beta^t$?

---

### Meta-Review · Area_Chair_NaSv · 2024-12-14

**Metareview:**

This paper considers ADMM for nonsmooth composite optimization with orthogonality constraints. The authors claim that they obtained improved complexity result comparing with existing work. However, all reviewers found that the paper is not well-written. There are technical concepts not defined or not clearly explained, making the paper hard to read and the results hard to verify. The authors did not provide a rebuttal to clarify the issues raised.

**Additional Comments On Reviewer Discussion:**

The author didn't provide a rebuttal.

---

### Decision · Program_Chairs · 2025-01-22

Reject